# Synaptic connectivity to L2/3 of primary visual cortex measured by two-photon optogenetic stimulation

Travis A Hage*, Alice Bosma-Moody, Christopher A Baker, Megan B Kratz, Luke Campagnola, Tim Jarsky, Hongkui Zeng, Gabe J Murphy

Allen Institute for Brain Science, Seattle, United States

**Abstract** Understanding cortical microcircuits requires thorough measurement of physiological properties of synaptic connections formed within and between diverse subclasses of neurons. Towards this goal, we combined spatially precise optogenetic stimulation with multicellular recording to deeply characterize intralaminar and translaminar monosynaptic connections to supragranular (L2/3) neurons in the mouse visual cortex. The reliability and specificity of multiphoton optogenetic stimulation were measured across multiple Cre lines, and measurements of connectivity were verified by comparison to paired recordings and targeted patching of optically identified presynaptic cells. With a focus on translaminar pathways, excitatory and inhibitory synaptic connections from genetically defined presynaptic populations were characterized by their relative abundance, spatial profiles, strength, and short-term dynamics. Consistent with the canonical cortical microcircuit, layer 4 excitatory neurons and interneurons within L2/3 represented the most common sources of input to L2/3 pyramidal cells. More surprisingly, we also observed strong excitatory connections from layer 5 intratelencephalic neurons and potent translaminar inhibition from multiple interneuron subclasses. The hybrid approach revealed convergence to and divergence from excitatory and inhibitory neurons within and across cortical layers. Divergent excitatory connections often spanned hundreds of microns of horizontal space. In contrast, divergent inhibitory connections were more frequently measured from postsynaptic targets near each other.

*For correspondence:
travish@alleninstitute.org

Competing interest: The authors declare that no competing interests exist.

## Editor's evaluation

Hage and colleagues used channelrhodopsin-mediated 2 photon stimulation combined with genetic labeling to survey the synaptic connections from layers 4, 5 and 6 onto layer 2/3 pyramidal neurons in the adult mouse primary visual cortex (V1). This work not only confirms prior knowledge regarding synaptic connectivity made through paired intracellular recordings, but also provides important new parameters and constraints about these connections. The results will contribute to our understanding of the cortical information processing.

## Introduction

The receptive field properties of visual cortical neurons are governed by long-range synaptic connections formed across brain areas and local inputs formed between cells within a cortical area. Lack of detailed information about both inter-areal and local connections limits mechanistic understanding of how sensory information is propagated and transformed. So, too, does the absence of detailed information about the patterns of convergence of synaptic inputs to neurons in the recipient layers and divergence of outputs from the projecting layers.

Positioned between the thalamic input layer 4 and output layer 5, L2/3 pyramidal cells (L2/3 PCs) represent a critical internal node in the canonical pathway. Although L2/3 neurons in primary visual cortex (V1) are known to receive direct input from dorsal lateral geniculate nucleus (dLGN) (*Ji et al., 2016*; *Morgenstern et al., 2016*; *D'Souza et al., 2019*) and from higher visual areas (*D'Souza et al., 2016*; *Young et al., 2021*), monosynaptic rabies tracing suggests the majority of presynaptic inputs arise from neurons within V1 (*Liu et al., 2013*; *Wertz et al., 2015*). These local inputs include well-established recurrent excitatory connections between L2/3 PCs, horizontal inhibition from nearby interneurons, and ascending inputs from L4 excitatory neurons that directly receive sensory information from thalamus.

Local ascending projections from neurons in layers 5 and 6 to L2/3 have been described in anatomical studies of sensory cortex and are thought to represent feedback from later stages of cortical processing (*Burkhalter, 1989*; *Yoshioka et al., 1994*; *Harris et al., 2019*). Experiments using laser scanning photostimulation combined with glutamate uncaging in visual and somatosensory cortex of juvenile rodents have identified connections from excitatory neurons in L5 to L2/3 PCs (*Dantzker and Callaway, 2000*; *Bureau et al., 2004*; *Shepherd et al., 2005*; *Shepherd and Svoboda, 2005*; *Yoshimura et al., 2005*; *Hooks et al., 2011*; *Xu et al., 2016*). However, connections from L5 to L2/3 have been scarce or absent when probed via simultaneous electrophysiological recordings in rodent cortex (*Thomson and Bannister, 1998*; *Reyes and Sakmann, 1999*; *Thomson et al., 2002*; *Lefort et al., 2009*; *Jiang et al., 2015*). The L5 to L2/3 connection is most often observed from L5a and therefore thought to originate from intratelencephalic (IT)-type L5 pyramidal neurons (*Dantzker and Callaway, 2000*; *Bureau et al., 2004*; *Shepherd et al., 2005*; *Shepherd and Svoboda, 2005*; *Yoshimura et al., 2005*; *Hooks et al., 2011*; *Xu et al., 2016*). It was recently demonstrated that feedback projections from anteromedial or lateromedial visual areas preferentially target IT-type L5 neurons in V1 to generate loops between higher and lower visual areas (*Young et al., 2021*), raising the possibility that local ascending projections from L5 could transmit behavioral or brain-state-related information to superficial layers. A direct comparison of the translaminar excitation from layers 4–6 to L2/3 in adult rodent V1 has not yet been made and will be necessary to understand the contributions of feedforward- and feedback inputs to the response properties of L2/3 neurons.

Likewise, direct comparisons of horizontal and vertical inhibition are needed to understand the role of local inhibition in shaping neuronal responses. Intralaminar inhibition from interneurons to nearby PCs has been described as abundant and nonselective when measured using two-photon glutamate uncaging (*Packer and Yuste, 2011*; *Fino et al., 2013*; *Karnani et al., 2014*). Such horizontal inhibition from somatostatin-expressing (Sst) interneurons shapes the receptive field properties of nearby L2/3 PCs in mouse visual cortex (*Adesnik et al., 2012*). Several patterns of vertical inhibition (mediated by translaminar inhibitory connections) have been described for Sst interneurons (*Kapfer et al., 2007*; *Silberberg and Markram, 2007*; *Jiang et al., 2015*; *Anastasiades et al., 2016*; *Naka et al., 2019*) and parvalbumin-expressing (Pvalb) interneurons (*Olsen et al., 2012*; *Bortone et al., 2014*). Furthermore, subtypes of Sst interneurons may be differentially integrated into distinct layer-specific subnetworks (*Muñoz et al., 2017*; *Naka et al., 2019*). While intra- and inter-laminar inhibitory inputs to L2/3 of V1 have been compared using one-photon photostimulation of GABAergic interneurons via either caged glutamate (*Xu et al., 2016*) or channelrhodopsin-2 (ChR2) expression (*Kätzel et al., 2011*), potentially unique roles of subclasses of GABAergic interneurons may be revealed by photostimulation techniques with increased genetic and spatial specificity.

To this end, two-photon excitation targeted to individual opsin-expressing neurons within genetically defined subclasses of cells can provide improved genetic and spatial precision. Two-photon optogenetic stimulation capable of inducing action potentials (APs) was first demonstrated using rapid galvo scanning of cultured dissociated neurons expressing ChR2 (*Rickgauer and Tank, 2009*). Newly discovered and engineered opsin variants were subsequently combined with rapid scanning and patterned illumination methods to demonstrate feasibility of two-photon optogenetic stimulation in acute brain slices and in vivo (*Andrasfalvy et al., 2010*; *Papagiakoumou et al., 2010*; *Prakash et al., 2012*; *Packer et al., 2012*; *Packer et al., 2015*; *Chaigneau et al., 2016*; *Ronzitti et al., 2017*; *Pégard et al., 2017*; *Shemesh et al., 2017*; *Forli et al., 2018*; *Mardinly et al., 2018*; *Yang et al., 2018*; *Marshel et al., 2019*). In recent years, these methods have been combined with electrophysiological recordings to measure synaptic connectivity, with individual studies focused on recurrent

excitatory connectivity within cortical layers (*Izquierdo-Serra et al., 2018*; *Seeman et al., 2018*) or distinct connectivity of subtypes of Sst interneurons (*Naka et al., 2019*).

In this study, we examined the applicability of two-photon optogenetics to a probe a diverse array of potential synaptic connections by measuring the reliability and specificity of two-photon optogenetic stimulation using seven different Cre lines with either transgenic or adeno-associated virus (AAV)-mediated expression of the opsin ChrimsonR (*Klapoetke et al., 2014*). We then combined multiphoton stimulation with multicellular patch-clamp recording (up to four neurons) to measure the spatial distribution and functional properties of intralaminar and translaminar synaptic connections to neurons in L2/3 of primary visual cortex in the young adult mouse (722 total connections identified from 10,720 probed). Measures of within-layer connectivity were validated by comparing two-photon optogenetics data to independently collected, gold standard multicellular recordings from the same cortical region and developmental stage. Additionally, we performed targeted patching and validation of optically identified connections in a subset of experiments.

Consistent with the canonical microcircuit, L4 excitatory neurons (labeled by Rorb-Cre or Scnn1a-Cre) and inhibitory interneurons within L2/3 displayed the highest rates of connectivity onto L2/3 PCs, while recurrent excitatory connectivity was sparse. Furthermore, we observed translaminar connections from L5 IT-type excitatory neurons (labeled by Rorb-Cre or Tlx3-Cre), and vertical inhibition from Pvalb and Sst interneurons in L4 and L5. Multicellular recordings allowed us to identify individual presynaptic cells with divergent connections onto multiple postsynaptic targets in L2/3. Diverging outputs from inhibitory interneurons were most often confined to postsynaptic cells within 100 µm of each other. By contrast, translaminar connections from individual excitatory neurons in L4 and L5 often diverged to target distant postsynaptic cells separated by hundreds of microns. Many classes of synaptic connections displayed broad and skewed distributions in the amplitudes of excitatory and inhibitory postsynaptic potentials (EPSPs and IPSPs) – with a small minority of connections generating very strong responses. Differences in the strength, kinetics, and short-term dynamics of synaptic responses were found across presynaptic and postsynaptic subclasses of cells; however, the amplitudes of less common synaptic connections were often similar to the more canonical connection paths when probed at the single-cell level.

## Results

We measured intra- and translaminar synaptic connectivity to L2/3 in primary visual cortex of young adult mouse using two-photon stimulation of ChrimsonR-expressing neurons. We used layer-specific Cre lines – Penk-Cre (L2/3 IT), Scnn1a-Cre (L4 IT), Rorb-Cre (L4 and L5 IT), Tlx3-Cre (L5 IT), and Ntsr1-Cre (L6 coritcothalamic [CT]) to stimulate excitatory neurons. Pvalb-Cre and Sst-Cre were used to stimulate subclasses of inhibitory interneurons in multiple layers. Photostimulation of possible presynaptic cells has the potential to greatly increase the throughput of synaptic physiology experiments, but the approach does not provide the verification and record of presynaptic APs for all tested connections that one obtains from multicellular patch-clamp recording. Additionally, while current injection into a patch-clamped cell reliably yields single-cell precision, photostimulation has the potential to activate nearby photosensitive neurons (*Anastasiades et al., 2018*). Therefore, we began our study by carefully measuring the sensitivity and specificity of two-photon stimulation across all Cre lines used in subsequent mapping experiments.

### Characterization of two-photon stimulation of ChrimsonR-expressing neurons

A number of methods have been developed for two-photon stimulation of neurons, with varying trade-offs in terms of reliability, spatial specificity, ability to stimulate multiple neurons, and complexity (*Emiliani et al., 2015*). We pursued galvo-based rapid spiral scanning of ChrimsonR-expressing neurons (*Rickgauer and Tank, 2009*; *Prakash et al., 2012*; *Packer et al., 2012*; *Klapoetke et al., 2014*) as a method to stimulate presynaptic cells (*Figure 1A*). We tested the reliability and specificity of two-photon stimulation using two methods of ChrimsonR expression. We first utilized the Ai167 TIGRE2.0 line, which we previously found generated robust one-photon-evoked photocurrents (*Daigle et al., 2018*). The second method improved the spatial specificity of photostimulation by targeting trafficking of ChrimsonR to the soma and proximal dendrites of neurons (*Baker et al.,*

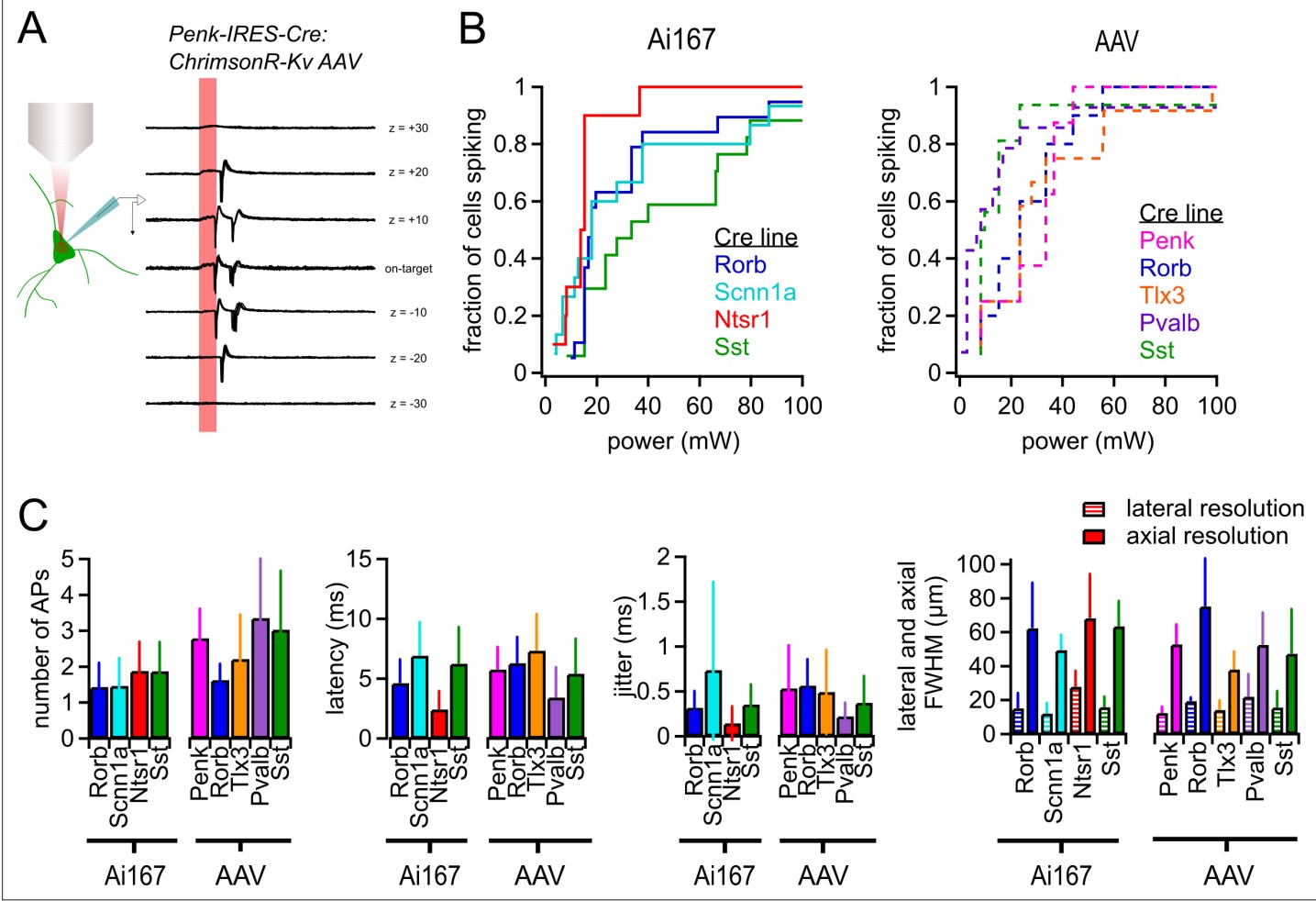

**Figure 1.** Characterization of two-photon-evoked spiking. (**A**) Examples of action potentials (APs) evoked by two-photon stimulation of a Penk-Cre neuron with indicated axial offsets. Each offset includes 10 overlaid sweeps. Stimulus duration was 10 ms (red shaded bar). (**B**) Cumulative probability of light-evoked spiking versus photostimulus power for the indicated Cre lines crossed to Ai167 or injected with AAV carrying soma-targeted ChrimsonR. (**C**) The average number of APs evoked per photostimulus, latency, jitter, and spatial resolution of photostimulation (measured using the same power used for mapping experiments) for all Cre line and opsin expression strategies used in this study. Error bars represent standard deviation across cells. Minimum power, latency, and jitter data were collected from a total of 123 neurons (8–19 neurons per Cre line-expression method combination). Lateral resolution was measured from a total of 66 neurons (5–12 per Cre line-expression method combination. Axial resolution was measured from a total of 78 neurons (6–13 per Cre line-expression method combination)).

The online version of this article includes the following source data and figure supplement(s) for figure 1:

**Source data 1.** Raw values for plots in *Figure 1*.

**Figure supplement 1.** Characterization of spiking evoked by two-photon stimulation.

**Figure supplement 1—source data 1.** Raw values for plots in *Figure 1—figure supplement 1*.

**Figure supplement 2.** Lateral resolution of spiking evoked by two-photon stimulation.

**Figure supplement 3.** Axial resolution of spiking evoked by two-photon stimulation.

2016). Specifically, we utilized a combinatorial approach in which we performed stereotaxic injections AAV providing Cre-dependent expression of soma-targeted-ChrimsonR (flex-ChrimsonR-EYFP-KV2.1) into primary visual cortex of Cre-positive mice.

To measure the sensitivity and specificity of two-photon stimulation, we used loose-seal cell-attached recordings to minimally disturb intrinsic cell properties (*Figure 1A*; 10 overlaid sweeps at each stimulus position). Across all ChrimsonR-expressing cells, the minimum power necessary to reliably generate one or more AP per stimulus ranged from 0.5 to 94 mW (10 ms photostimulus duration, measured from 8 to 19 neurons for each Cre line/expression method combination; *Figure 1B*).

Although the distributions vary between Cre lines, we observed spiking in 85–100% of targeted cells in all Cre lines when using an 85 mW photostimulus. AAV-transfected neurons in the Pvalb and Sst lines were very sensitive to photostimulation – half of the sampled cells reliably generated APs in response to 8 mW photostimuli (*Figure 1B*). Based on these results, we used 85 mW as our primary photostimulation power for probing synaptic connectivity in experiments using Ai167 and excitatory Cre lines transfected with AAV. To avoid off-target activation of the more photosensitive AAV-transfected interneurons, we used a lower stimulus intensity of 35 mW, which reliably activated 86% (Pvalb) and 94% (Sst) of ChrimsonR-expressing interneurons tested.

We measured the number of light-evoked APs, the latency to the first AP, and the associated temporal jitter (standard deviation of the latency) as a function of stimulus intensity (*Figure 1—figure supplement 1*). The mean and standard deviation of these values at the powers used for mapping experiments are presented in *Figure 1C*. Latency and jitter decreased with higher stimulus intensities, although the effect on jitter was less dramatic as the majority of cells displayed sub-millisecond jitter even with low-power stimuli. Some cells generated multiple APs per photostimulus (*Figure 1A and C*, *Figure 1—figure supplement 1*). Similarly, we observed instances of multiple light-evoked postsynaptic potentials (PSPs) in subsequent mapping experiments and utilized exponential deconvolution to aid in identifying such cases (see 'Materials and methods').

Finally, we characterized the spatial resolution of photostimulation. The probability of light-evoked spiking decreased rapidly with lateral offset of the photostimulus target (full-width at half-maximum [FWHM] of Gaussian curve fit: 12–23 μm; *Figure 1C*, *Figure 1—figure supplement 2*), whereas greater axial offset was necessary to observe the same decrease in light-evoked spiking (FWHM: 38–75 μm; *Figure 1C*, *Figure 1—figure supplement 3*). This asymmetry in effective excitation volume is consistent with previous reports (*Rickgauer and Tank, 2009*; *Prakash et al., 2012*; *Packer et al., 2012*; *Izquierdo-Serra et al., 2018*) and suggests that unwanted off-target activation of cells will be most likely when two or more Cre-labeled cells are in near axial alignment. Both the latency and jitter of light-evoked spiking increased with either lateral or axial offset of the photostimulus location (*Figure 1—figure supplement 2*, *Figure 1—figure supplement 3*). As a result, while nearly all responsive neurons displayed AP firing with short latency (<15 ms) and low jitter (<2 ms) following on-target stimulation (143/144 cells, 99%, aggregated across Cre lines), only 30% of photoresponses measured with 10 μm of lateral offset met the same criteria (127/423 photostimulus locations). Such precise firing was nearly absent with 20 μm of lateral offset (9/396 photostimlus locations; 2.3%). The data further suggest that off-target photostimulation is unlikely to produce reliable, short-latency synaptic responses.

## Measurements of cell densities and estimation of off-target activation

In addition to the spatial resolution of individual opsin-expressing neurons, the likelihood of generating off-target spiking will be influenced by the density of photosensitive cells. Across Cre lines and expression methods used in this study, the average densities of labeled cells ranged from ~2000 to 22,000 cells/mm$^3$ (*Figure 2A–C*). As expected, labeling with interneuron Cre lines was relatively sparse in comparison to excitatory Cre lines (*Figure 2A–C*). For each Cre line, the density of cells labeled using AAV was slightly lower or approximately equal to the corresponding density measured in Ai14 (tdTomato reporter) mice (*Figure 2B*; on average, density with AAV was 84% of densities measured in Ai14). In contrast, cell densities measured from Ai167 mice suggest a minority of cells within a targeted subclass were labeled (*Figure 2C*; on average, density with Ai167 was 25% of densities measured in Ai9 or Ai14). Incomplete or sparse labeling has been observed with other combinations of Cre lines and some TIGRE1.0 or TIGRE2.0 reporter lines (*Madisen et al., 2015*; *Daigle et al., 2018*; *Bounds et al., 2021*). Although the sparse labeling by Ai167 may be serendipitous to the extent that it limits off-target photostimulation, the more thorough labeling and inclusion of a soma-targeting motif with AAV-mediated expression motivated us to use it as our primary strategy in this study. Data collected from individual Cre lines using either method of expression (Rorb and Sst) are directly compared in subsequent sections (Figure 4B and C, Figure 6B and C, Figure 7—figure supplement 3). When targeting excitatory subclasses, labeled cells were observed in the expected cortical layers (Figures 3-5), and the intrinsic electrophysiological features of labeled Pvalb and Sst interneurons resembled previous reports (Figure 3—figure supplement 1). In total, the data suggest that the targeted Cre line-defined subclasses were appropriately labeled. Notably, in all experiments,

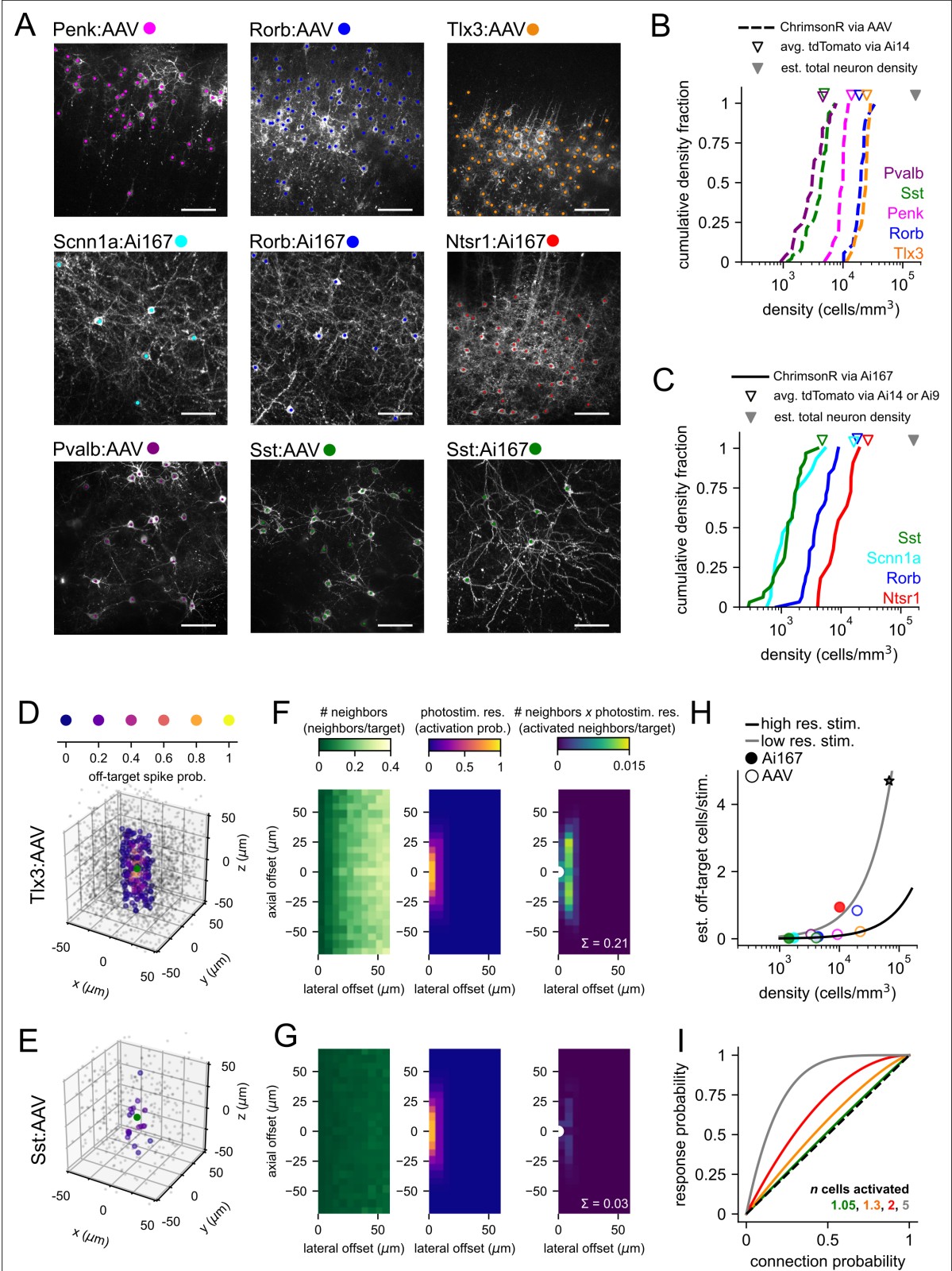

**Figure 2.** Labeled cell densities and estimates of off-target activation. (**A**) Example images for all Cre lines and expression method combinations used in this study. Two-photon z-stacks were collected at the conclusion of mapping experiments. Each image shown is a maximum intensity projection of a 40 μm subset of the axial/z dimension. Scale bars = 60 μm. The positions of labeled cells were manually annotated in three dimensions (colored dots, z location not represented here). (**B**) Cumulative distributions of labeled cell densities measured across multiple experiments using adeno-

*Figure 2 continued on next page*

*Figure 2 continued*

associated virus (AAV) (dashed lines). Distributions are color-coded by Cre line as in panel (**A**). Open triangles represent the mean density of labeled cells measured from Ai14 adult mice crossed to the same Cre lines. Filled gray triangle represents the average reported total neuron density in mouse visual areas (*Keller et al., 2018*). (**C**) Same as panel (**B**) for experiments using Ai167-mediated ChrimsonR expression. (**D**) For 100 randomly selected neurons in the Tlx3:AAV dataset (overlaid green circles), the relative location of all identified neighboring cells s plotted and color-coded by estimated off-target activation probability (note warmer colors near the origin). Neighbors with an off-target activation probability < 0.01 are represented by smaller gray circles. (**E**) Same as panel (**D**) for 100 starter neurons from the Sst:AAV dataset (green circles). (**F**) Estimation of off-target activation for Tlx3:AAV experiments. Left panel: heatmap of the number of neighbors per cell at combined lateral and axial offsets. With increasing lateral offsets, each bin corresponds to a concentric disc of increasing volume (*Figure 2—figure supplement 1*), and therefore, an increasing number of neighbors are observed. Middle panel: heatmap of the photostimulus resolution. Right panel: heatmap of the number of activated off-target neighbors per targeted cell estimated as the product of the number of neighbors and the probability of off-target activation. The sum across all distances (0.21; white text) estimates the average number of off-target cells activated by photostimulation in Tlx3:AAV experiments and is plotted as an orange circle in panel (**H**). White semi-circle represents the location of the targeted neuron. (**G**) Same as panel (**F**) for Sst:AAV experiments. The same color scales were used in each vertically aligned panel. (**H**) The estimated number of activated off-target cells per stimulus versus labeled cell density for all Cre line and expression method combinations used in this study. Continuous lines represent estimates of off-target activation across a range of cell densities using two different photostimulus resolutions (chosen resolutions correspond to the range of resolutions observed across Cre lines/expression methods; black line: lateral full-width at half-maximum [FWHM] = 12 µm, axial FWHM = 35 µm; gray line: lateral FWHM = 24 µm, axial FWHM = 66 µm; see also *Figure 2—figure supplement 1*). Gray star represents the reported estimate of the total number of cells activated per photostimulus using regularly spaced two-photon photolysis of caged glutamate in rat visual cortex (*Matsuzaki et al., 2008*). (**I**) Predicted fraction of photostimuli generating a synaptic response following stimulation of multiple cells (n) with a common connection probability (Equation 1) plotted against connection probability. Solid colored lines represent different values of *n*. Dashed black line represents unity.

The online version of this article includes the following source data and figure supplement(s) for figure 2:

**Source data 1.** Raw values for plots in *Figure 2*.

**Figure supplement 1.** Estimation of off-target photostimulation.

the photosensitive cells constitute a minority of all neurons as estimates of total neuron densities in visual cortex of the mouse range from 92,000 to 214,000 cells/mm$^3$ (*Figure 2B and C*; *Cragg, 1967*; *Heumann et al., 1977*; *Rockel et al., 1980*; *Keller et al., 2018*).

To translate our measurements of cell densities to estimates of off-target activation for each Cre line and expression method, we first calculated all cell-to-cell distances in the lateral (x and y) and axial (z) dimensions. Using these distances and the measurements of spatial resolution described above (*Figure 1*), we estimated the probability of off-target activation for each neighbor. Spatial offsets and corresponding activation probabilities are illustrated for 100 randomly selected starter cells and all identified neighbors from a densely labeled driver line (Tlx3-Cre; *Figure 2D*) and a relatively sparse line (Sst-Cre; *Figure 2E*) – both transfected with AAV. For quantification across all experimental conditions, two-dimensional histograms measuring the number of neighbors at combined lateral and axial offsets were generated and an edge correction was applied to account for possible off-target cells outside of the imaged volumes (see 'Materials and methods'). The resulting heatmaps (*Figure 2F and G*, left panels) represent the average number of labeled neighbors per targeted cell. Next, measurements of the spatial resolution of photostimulation (*Figure 1—figure supplement 2*, *Figure 1—figure supplement 3*) were used to estimate the probability of off-target activation at the same lateral and axial offsets (*Figure 2F and G*, middle panels). The product of these two arrays predicts the number of off-target activated neighbors at each distance (*Figure 2F and G*, right panels). Finally, the sum across all elements estimates the total number of activated off-target cells per targeted cell (0.21 off-target cells/stimulus for Tlx3:AAV and 0.03 off-target cells/stimulus for Sst:AAV). These estimates were generated for the nine experimental conditions used in this study (*Figure 2H*, colored markers) and for homogeneously distributed photosensitive neurons (*Figure 2H*, *Figure 2—figure supplement 1*) at densities up to 164,000 cells/mm$^3$ (the average reported density of all neurons in mouse visual cortex; *Keller et al., 2018*).

The estimates outlined above predict that unwanted off-target activation is uncommon with sparse interneuron Cre lines (estimated off-target cells/stimulus < 0.13; *Figure 2H*), but considerably higher when using more densely labeled excitatory Cre lines (up to 0.94 off-target cells/stimulus for Ntsr1-Cre combined with Ai167; *Figure 2H*, filled red circle). Applying photostimulation with similar spatial resolution to experiments in which a larger fraction of neurons are photosensitive (due to either broader opsin expression or use of caged glutamate) will likely activate multiple cells per target (*Figure 2H*, line plots). In close agreement with these estimates, two-photon photolysis of caged glutamate was

previously estimated to activate ~4.7 excitatory neurons at each photostimulus location (based on an estimated PC density of 68,000 cells/mm$^3$ in rat visual cortex; *Matsuzaki et al., 2008*; *Figure 2H*, gray star). Therefore, restriction of photosensitivity using subclass-specific Cre lines can dramatically limit the extent of unwanted off-target activation. Additionally, synaptic responses resulting from off-target activation are expected to originate from the subclass of cells labeled by the utilized Cre line.

Finally, we considered what effect off-target activation may have on estimates of synaptic connectivity inferred from photostimulus responses. The probability of generating an off-target synaptic response will depend on both the total number of cells activated and the connection probability of those cells. If we assume that off-target presynaptic cells exhibit similar connection probabilities to the targeted cells (of a common Cre line and in a similar anatomical location), we can estimate the probability of at least one photostimulated cell producing a synaptic response (P$_{response}$) as (*Equation 1*):

$$P_{response} = 1 - (1 - P_{connection})^n \tag{1}$$

where P$_{connection}$ represents the per cell connection probability and $n$ represents the total number of cells activated per photostimulus (including targeted and untargeted). For values of $n$ predicted for the experimental conditions within this study (<2 cells/photostimulus), Equation 1 predicts that P$_{response}$ will be proportional to P$_{connection}$ over nearly all probabilities from 0 to 1 (*Figure 2I*, colored lines). If $n$ is increased to five activated neurons (approximating a two-photon stimulus in which the majority of neurons are photosensitive), P$_{response}$ becomes saturated as P$_{connection}$ approaches 0.5 (*Figure 2I*, gray line). In this scenario, it may be difficult to distinguish abundant connectivity from ubiquitous connectivity; however, the measured likelihood of responses can potentially reveal patterns within sparsely connected circuits. It is also important to consider that as the number of off-target cells increases the potential for polysynaptic responses and/or complex responses from a heterogenous population of cells will likely increase as well. In total, these analyses predict that our two-photon optogenetic stimulation strategy will robustly correspond to underlying synaptic connectivity, even as modest off-target activation and some misassignment of connectivity to specific presynaptic cells may occur. We sought to verify these predictions in our initial mapping experiments and test putative connections identified by photostimulation using targeted patching.

## Intralaminar connectivity measured by two-photon stimulation resembles connectivity measured by paired recordings

Given the caveats associated with photostimulation, we began our study of synaptic connectivity to L2/3 of V1 by examining intralaminar connections so that we could compare data generated using two-photon optogenetics to analogous measurements from multicellular recordings (*Seeman et al., 2018*; *Campagnola et al., 2021*). L2/3 excitatory neurons were genetically targeted using the Penk-Cre line (*Daigle et al., 2018*; *Tasic et al., 2018*) with AAV-mediated ChrimsonR expression. An example experiment in which four cells in L2/3 were patched is shown in *Figure 3A*. Postsynaptic cells A–C displayed electrophysiological and morphological features consistent with excitatory PCs, while cell D displayed high-frequency AP firing and other intrinsic features typical of Pvalb-expressing fast-spiking interneurons (FSIs) (*Figure 3A*). Throughout the study, interneurons were assigned to one of three subclasses – FSIs (putative Pvalb), putative VIP, and putative Sst interneurons – based on unsupervised clustering using 14 intrinsic electrophysiological features (*Figure 3—figure supplement 1*). Subclass assignments were validated by genetic labeling of a subset of Pvalb and Sst interneurons and by comparison to genetically labeled interneurons in the Allen Institute Cell Types Database (*Gouwens et al., 2019*; *Figure 3—figure supplement 1*). Example traces from three photostimulated Penk-labeled neurons (*Figure 3A*) highlight a connection to a L2/3 PC (photostimulus 1 to cell B) and two convergent connections onto the FSI (photostimuli 2 and 3 to cell D).

Data compiled across experiments are presented in *Figure 3B* by plotting each probed connection according to the distance of the presynaptic neuron from pia and the horizontal offset between pre- and postsynaptic neurons. Consistent with previous reports of sparse recurrent excitatory connectivity in L2/3 of mouse V1 (*Holmgren et al., 2003*; *Jiang et al., 2015*; *Seeman et al., 2018*; *Izquierdo-Serra et al., 2018*), a small fraction of photostimulated Penk neurons generated an excitatory response in L2/3 PCs (5.8%, 12/205 connections probed at intersomatic distances < 100 μm). The connection probability decreased significantly at larger intersomatic distances (*Figure 3C*; chi-squared test statistic = 11.8, p=0.0027, largest distance of connected cells = 258 μm).

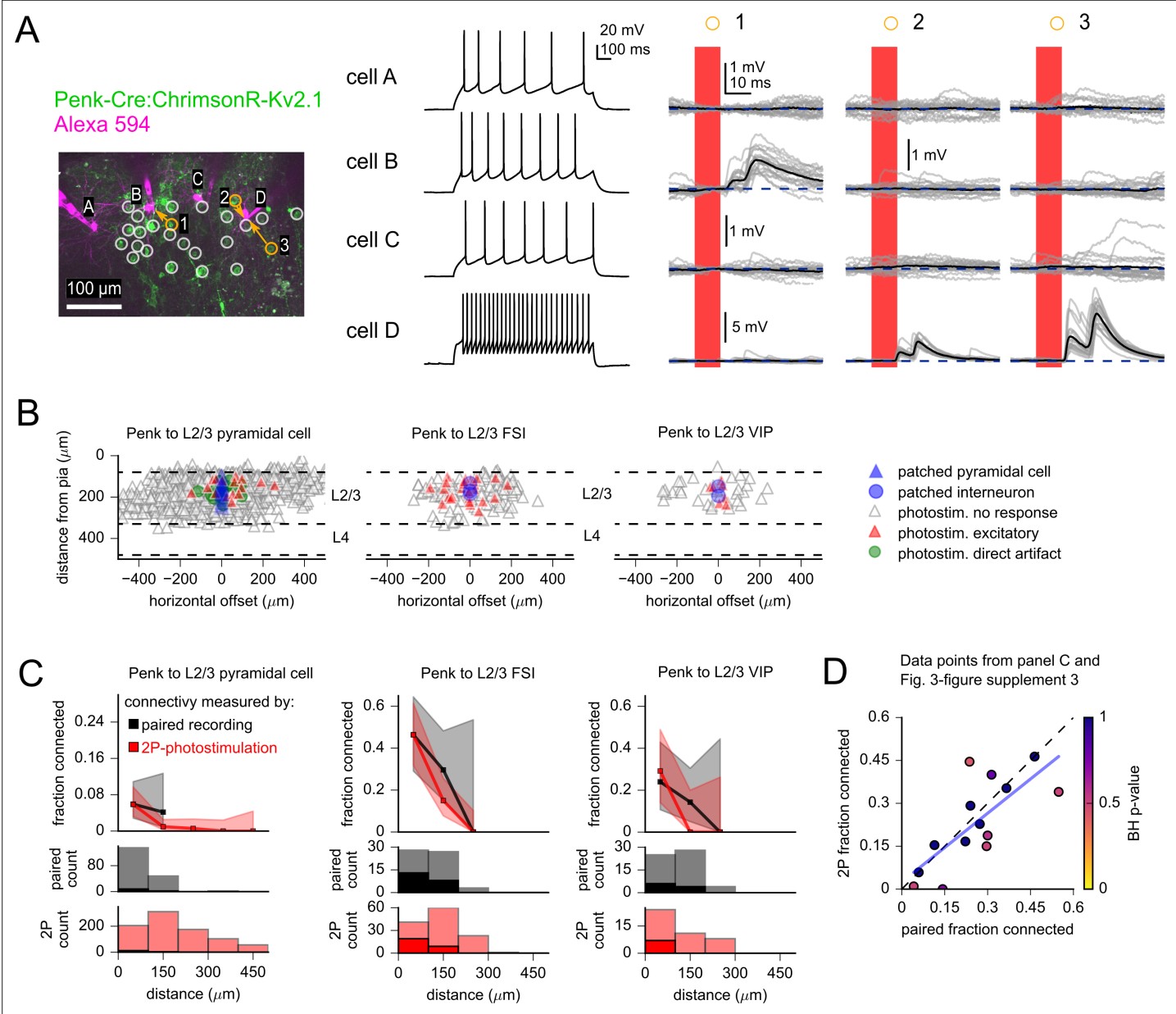

**Figure 3.** Measurement of intralaminar excitatory connectivity by two-photon optogenetic stimulation. (**A**) Example experiment measuring connectivity from L2/3 excitatory neurons labeled by the Penk-Cre line to three L2/3 pyramidal cells (PCs) (cells A–C) and one fast-spiking interneuron (FSI) (cell D). Left: flattened z-stack of patched neurons filled with Alexa 594 (magenta) and EYFP signal from soma-targeted ChrimsonR-expressing neurons (green). Yellow circles: photostimulated cells with responses plotted in right panels. White circles: photostimulated cells with responses not shown. Middle: voltage responses to 1 s current injection, +40 pA relative to rheobase. Right: photoresponses to indicated locations. Photostimulation occurred during the red bar, black traces are an average of individual voltage responses (gray traces). The y-axis scale was increased for cell D to accommodate the large excitatory postsynaptic potentials (EPSPs). (**B**) Summary maps of Penk-labeled excitatory inputs to the indicated subclass of postsynaptic cell (n = 895 connections probed to L2/3 PCs, 125 to L2/3 FSIs, and 43 to L2/3 putative VIP interneurons). Locations of neurons are plotted according to their distance from the pia and the horizontal distance between cells with the following color scheme: blue, postsynaptic cells (n = 25 PCs, three FSIs, two putative VIP interneurons); gray, photostimulated neurons that did not evoke a synaptic response; red, photostimulated cells that produced an excitatory response; green, photostimulated cells that produced a direct stimulus artifact due to expression of ChrimsonR in a subset of the patched L2/3 PCs (19 photostimulus locations). Dashed lines represent approximate layer boundaries. Positive horizontal offsets correspond to presynaptic cells posterior to the postsynaptic cell. (**C**) Connection probability measured by two-photon optogenetic stimulation (red) or paired recording (black). Distance refers to total Euclidean distance in three dimensions. Shading represents 95% confidence intervals. Histograms on bottom rows indicate number of connections probed (faint colors) and identified (saturated colors) using each technique. (**D**) Connection probability measured by two-photon stimulation plotted against corresponding measures made by paired recordings. Each point represents a category of connection defined by presynaptic

*Figure 3 continued on next page*

*Figure 3 continued*

and postsynaptic cell subclass and intersomatic distance. See *Figure 3—figure supplement 3* and *Table 1* for additional statistics. Color of markers indicates multiple-hypothesis-corrected p-values generated by performing a Fisher's exact test for each dataset, followed by a Benjamini–Hochberg procedure with a false discovery rate set to 0.25. A high false discovery rate was used to avoid type II errors that could discount differences between the two techniques. Dashed line represents unity. Blue line represents a linear regression between the connection probabilities measured by each method (slope = 0.80 ± 0.21, y-intercept = 0.03 ± 0.06).

The online version of this article includes the following source data and figure supplement(s) for figure 3:

**Source data 1.** Raw values for plots in *Figure 3*.

**Figure supplement 1.** Classification of recorded interneurons using intrinsic electrophysiological features.

**Figure supplement 1—source data 1.** Raw values for plots in *Figure 3—figure supplement 1*.

**Figure supplement 2.** Estimation of interneuron labeling by the Penk-Cre driver line.

**Figure supplement 3.** Comparison of intralaminar inhibition measured using two-photon optogenetics to paired recordings.

To determine if cell-class preferences were present in the outputs of photostimulated L2/3 excitatory neurons, we compared the frequency of observed excitatory responses in L2/3 interneurons to the frequency recorded in L2/3 PCs (*Figure 3B*). The throughput of two-photon stimulation allowed us to probe 168 potential intralaminar excitatory connections from a small number of interneuron recordings (three recordings from FSIs and two recordings from putative VIP interneurons). In contrast to the sparse excitation of L2/3 PCs, over 40% of photostimulated Penk neurons generated EPSPs in nearby FSIs (*Figure 3C*; intersomatic distance < 100 µm: 46.3%, 19/41 connections probed; Fisher's exact p=1e-9 compared to L2/3 PCs). Intralaminar excitatory connectivity to putative VIP interneurons was intermediate in likelihood (intersomatic distance < 100 µm: 29.2%, 7/24 connections probed; Fisher's exact p=1e-3 compared to L2/3 PCs). The data demonstrate cell-class specificity in the intralaminar outputs of L2/3 excitatory neurons labeled by Penk-Cre in adult mouse V1.

When comparing optogenetic data to studies using paired recordings or glutamate uncaging, it is important to note that use of transgenic driver lines may label cells that are not representative of a broader population – such as excitatory neurons defined by cortical layer. Of further note with the Penk-Cre driver line, transcriptomic analysis of dissociated cells shows that reporter-labeled cells include VIP interneurons in addition to L2/3 excitatory cells (*Tasic et al., 2018*). However, the fraction of reporter-labeled cells expressing the GABAergic marker gene *Gad1* appears low (~12%) when examined in situ (*Figure 3—figure supplement 2*; see 'Animals and stereotaxic injections' for additional information). The presence of a subpopulation of interneurons within the photostimulated cells will result in a proportional underestimate of excitatory connectivity; however, relative differences across postsynaptic subclasses are expected to be preserved (as the identity of photostimulated cells will not depend on the subclass of the patched cells). Accordingly, the high connectivity of L2/3 excitatory neurons to nearby FSIs is consistent with previous reports (*Holmgren et al., 2003*; *Kapfer et al., 2007*; *Hofer et al., 2011*; *Avermann et al., 2012*; *Pala and Petersen, 2015*), suggesting that Penk-labeled L2/3 excitatory neurons resemble other L2/3 excitatory neurons in terms of outgoing intralaminar connectivity.

We previously observed congruous measurements of recurrent connectivity among L5 IT neurons using either paired recordings or two-photon photostimulation of ReaChR-expressing cells (*Seeman et al., 2018*). To similarly validate the updated photostimulation paradigm used here, we compared measures of synaptic connectivity made by two-photon stimulation to analogous measurements from multicellular recordings of adult mouse V1 in the Allen Institute Synaptic Physiology dataset (*Campagnola et al., 2021*). Consistent with photostimulation data, paired recordings showed that the connectivity from L2/3 PCs to nearby Pvalb and VIP interneurons is dramatically higher than the connectivity between L2/3 PCs (*Figure 3C*). Furthermore, directly comparing connection probabilities measured by each technique at matched intersomatic distances did not reveal significant differences (*Table 1*), illustrating the accuracy of connectivity as measured by two-photon optogenetic stimulation.

We expanded the comparative analysis of paired recordings and two-photon photostimulation to include intralaminar inhibition from Pvalb and Sst interneurons to L2/3 PCs and L2/3 FSIs (*Figure 3—figure supplement 3*). Spatial patterns and properties of inhibitory connections measured by two-photon photostimulation will be later discussed in greater depth (see below). The greatest dissimilarity between the two techniques was found in the measures of L2/3 Sst to L2/3 PC connectivity within

**Table 1.** Comparison of synaptic connectivity measured by two-photon optogenetics to paired recordings.

Distance refers to Euclidean distance measured in three dimensions. Fisher's exact p-values are reported before correcting for multiple comparisons. Benjamini–Hochberg p-values were calculated from Fisher's exact p-values with a false discovery rate set to 0.25.

| Connection class | Distance (μm) | Paired recording connections (found/probed) | 2P opto connections (found/probed) | Paired recording connection probability | 2P opto connection probability | Fisher's exact p-value | Benjamini–Hochberg p-value |
|---|---|---|---|---|---|---|---|
| | 0–100 | 8/135 | 12/205 | 0.06 | 0.06 | 1 | 1 |
| Exc. to exc. | 100–200 | 2/48 | 3/312 | 0.04 | 0.01 | 0.13 | 0.51 |
| | 0–100 | 13/28 | 19/41 | 0.46 | 0.46 | 1 | 1 |
| Exc. to Pvalb | 100–200 | 8/27 | 9/60 | 0.3 | 0.15 | 0.15 | 0.51 |
| | 0–100 | 6/25 | 7/24 | 0.24 | 0.29 | 0.75 | 1 |
| Exc. to VIP | 100–200 | 4/28 | 0/11 | 0.14 | 0 | 0.31 | 0.72 |
| | 0–100 | 9/38 | 45/101 | 0.24 | 0.45 | 0.03 | 0.44 |
| Sst to exc. | 100–200 | 6/22 | 38/167 | 0.27 | 0.23 | 0.6 | 1 |
| | 0–100 | 8/36 | 2/12 | 0.22 | 0.17 | 1 | 1 |
| Sst to Pvalb | 100–200 | 4/35 | 4/26 | 0.11 | 0.15 | 0.71 | 1 |
| | 0–100 | 17/31 | 17/50 | 0.55 | 0.34 | 0.1 | 0.51 |
| Pvalb to exc. | 100–200 | 9/30 | 18/96 | 0.3 | 0.19 | 0.21 | 0.58 |
| | 0–100 | 30/82 | 6/17 | 0.37 | 0.35 | 1 | 1 |
| Pvalb to Pvalb | 100–200 | 32/102 | 14/35 | 0.31 | 0.4 | 0.41 | 0.82 |

100 μm of intersomatic distance (p=0.03 Fisher's exact test, uncorrected for multiple comparisons; *Table 1*). However, differences in connectivity were not statistically significant following correction for multiple comparisons (all p-values>0.44 following Benjamini–Hochberg correction; *Table 1*, *Figure 3—figure supplement 3*). As an additional test of coherence between the two methods, we plotted connection probabilities measured by two-photon optogenetic stimulation against probabilities measured by paired recordings for each class of connection and intersomatic distance range examined using both methods (*Figure 3D*). Connection probabilities measured by paired recordings and two-photon optogenetics displayed a strong and significant positive correlation (Pearson's $R$ = 0.74, p=2.3e-3). In total, the comparison of data from two-photon photostimulation with gold standard paired recordings demonstrates that two-photon optogenetic stimulation can be a powerful and accurate tool to measure synaptic connectivity in acute brain slices.

## Validation of optically identified connections

Encouraged by the alignment of our optical measurements of intralaminar connectivity with data from paired recordings, we turned our attention to translaminar excitatory connectivity onto L2/3 PCs. As there is less gold standard data on across-layer connectivity, in a subset of experiments, we sought to validate translaminar excitatory connections by targeting patch-clamp recordings to optically identified, presynaptic cells in L4 or L5 (ChrimsonR expression driven by either Rorb-Cre or Tlx3-Cre). In the example shown in *Figure 4A*, photostimulation of a L4 Rorb neuron reliably generated large EPSPs (>1 mV) in a L2/3 PC (*Figure 4Ai*). A second pipette was then used to patch the photostimulated L4 neuron, and presynaptic APs were generated via current injection (*Figure 4Aii*). Spike-aligned EPSPs measured in the postsynaptic L2/3 PC closely resembled the previously recorded, photo-evoked responses. Additional photostimulated Rorb cells in this experiment (*Figure 4Aiii*) included another optically identified, weaker connection from a L4 Rorb neuron (photostimulus 4).

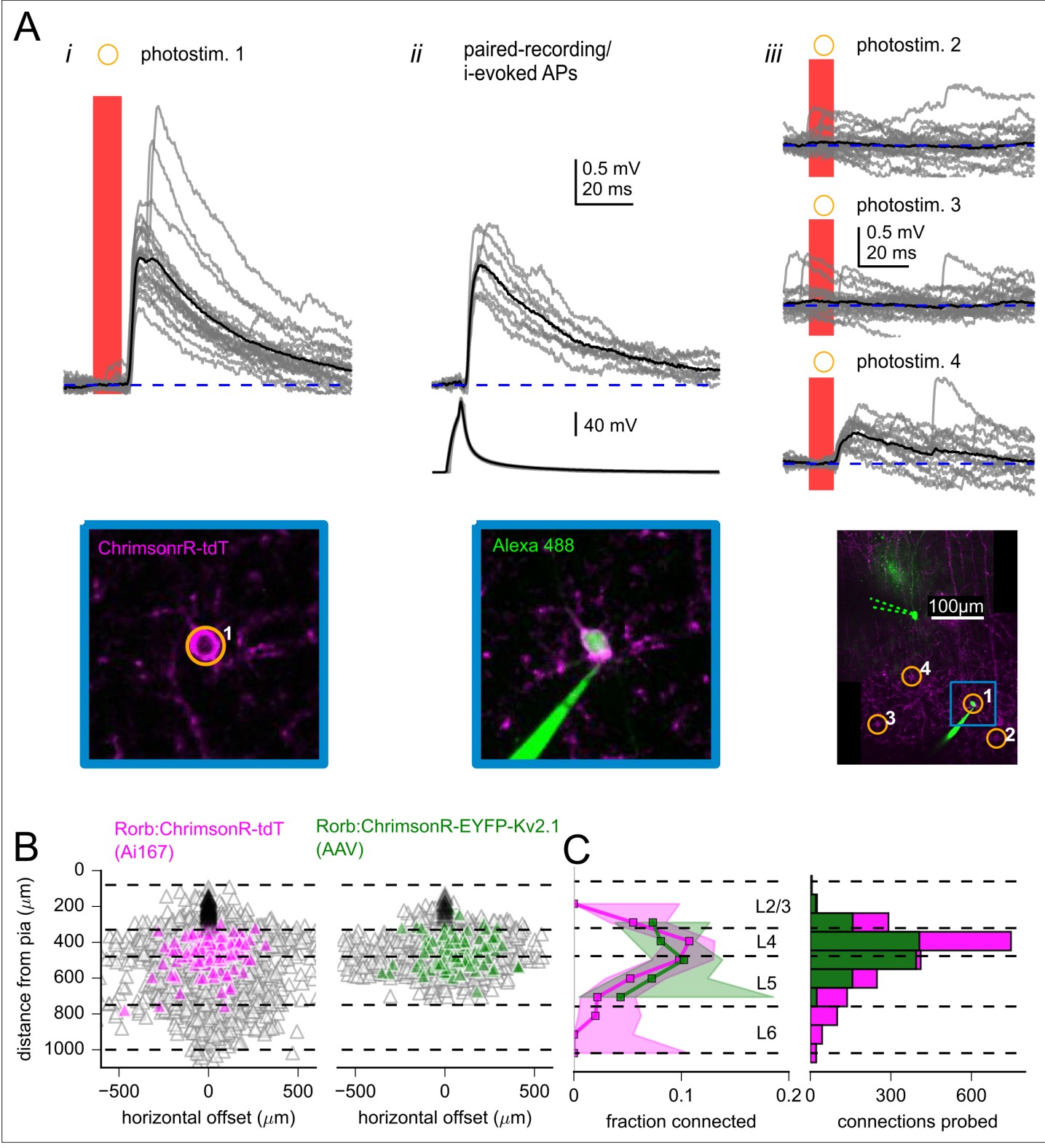

**Figure 4.** Measurement of excitatory connectivity from Rorb-labeled neurons to L2/3 pyramidal cells (PCs). (**A**) Example experiment measuring connectivity to a L2/3 PC from excitatory neurons labeled using Rorb-Cre:Ai167. Panel i: synaptic responses evoked by photostimulation of a Rorb-labeled cell (photostim. 1) Image of ChrimsonR-tdT signal (bottom) was captured immediately prior to photostimulation. Panel ii: the cell targeted by photostimulus 1 was subsequently patched and synaptic responses (top) were evoked by suprathreshold current injection into the Rorb-neuron. Image of the Rorb-labeled cell filled with Alexa 488, captured at the end of the experiment (bottom). Panel iii: additional photostimulus responses including

*Figure 4 continued on next page*

*Figure 4 continued*

a smaller excitatory synaptic response (photostim. 4). Overview images of photostimulated and patched cells collected at the end of the experiment (bottom). Dashed green lines represent original location of the postsynaptic recording pipette (removed before collection of the z-stack). Blue inset corresponds to location of images from panels i and ii. In all panels, black traces are an average of individual sweeps (gray traces). (**B**) Summary maps for all experiments with Rorb-Cre mice crossed to Ai167 (left) or injected with adeno-associated virus (AAV) encoding soma-targeted ChrimsonR (right; n = 2,269 connections probed using Rorb:Ai167, 1157 connections probed using Rorb:AAV). Locations of neurons are plotted with the following color scheme: black, postsynaptic L2/3 PCs (n = 79 recordings using Rorb:Ai167, 28 recordings using Rorb:AAV); gray, photostimulated neurons that did not evoke a synaptic response; magenta/green, photostimulated cells that produced an EPSP using Ai167 or AAV-mediated ChrimsonR expression. Dashed lines represent approximate layer boundaries. Positive horizontal distances correspond to presynaptic cells posterior to the postsynaptic cell. (**C**, left) The fraction of Rorb-labeled neurons connected to L2/3 PCs plotted against the distance of the presynaptic cell from the pia. Measurement was limited to cells within 300 μm horizontal distance. Shading indicates 95% confidence intervals. The fraction of cells connected, and associated confidence intervals, were not drawn if fewer than 20 connections were probed within a distance bin. Right: histogram of number of connections probed within each distance bin using either Ai167 or AAV, within 300 μm horizontal distance.

The online version of this article includes the following source data and figure supplement(s) for figure 4:

**Source data 1.** Raw values for plots in *Figure 4*.

**Figure supplement 1.** Verification of optically identified connections.

Across experiments examining translaminar excitatory connectivity with Rorb and Tlx3-Cre lines, we tested 12 optically identified presynaptic neurons by subsequent patching. We observed spike-evoked EPSPs for 11 of the putative excitatory connections – corresponding to a positive predictive value of 92% (binomial 95% confidence interval = 67–99%; *Figure 4—figure supplement 1*). We performed similar experiments to test optically identified inhibitory connections using the Pvalb and Sst-Cre lines. All 22 of these putative connections were verified by patch-clamp recording (binomial confidence interval = 89–100%). The amplitudes of postsynaptic responses measured by the two methods were strongly correlated (*Figure 4—figure supplement 1D*), illustrating that optically evoked responses provide an accurate measure of synaptic strength and providing further confidence in the specificity of our photostimulation protocol.

The observation of a false positive among optically identified connections (*Figure 4—figure supplement 1A*) demonstrates how connections in our mapping experiments may be viewed as 'putative connections' as photoresponses may have resulted from activation of a connected, Cre-labeled, ChrimsonR-expressing neuron other than the targeted cell. This is consistent with our characterization of the spatial resolution of our photostimulus protocol, which exceeds the size of a single soma, particularly in the axial dimension (*Figure 1C*). In combination with statistical comparisons of two-photon mapping data to paired recordings (*Figure 3C and D*), the high positive predictive values measured for excitatory and inhibitory connections argue that connectivity measured in our mapping experiments is representative of the underlying connectivity.

## Translaminar excitatory connectivity measured using two-photon optogenetics

The excitatory synaptic connection from L4 to L2/3 represents the initial feedforward projection within the canonical cortical microcircuit (immediately following input from the thalamus to L4 cells). To probe physiological features of this connection, we utilized Rorb-Cre mice with either AAV-mediated expression of soma-targeted ChrimsonR or transgenic expression of ChrimsonR via the Ai167 mouse line. Both expression methods resulted in labeling of neurons within L4, as well as relatively sparse expression in L2/3, L5, and L6 (*Figure 4B*). However, compared to using the transgenic line, labeling by AAV was more confined to L4 and the superficial region of L5 (*Figure 4B*, right panel). The differences in expression patterns could result from broader expression of Rorb over the course of development, as well as possible tropism of the virus (*Haery et al., 2019*). The spatial specificity of two-photon stimulation provided the ability to sequentially stimulate individual cells to test for differences in connectivity according to presynaptic cortical layers and opsin expression methods. Optically identified connections were observed from Rorb-labeled neurons within L2/3, L4, and L5; however, connections were most often found from Rorb neurons in L4 and slightly below the L4/L5 border (*Figure 4B*). Nearly all identified presynaptic partners in L4 were within 300 μm of horizontal distance from the L2/3 PC (*Figure 4B*). Within this horizontal distance, connection probabilities of L4 Rorb cells labeled by Ai167 or AAV were similar (*Figure 4B*). As an additional comparison, we

measured excitatory connectivity from L4 using Scnn1a-Cre mice (labeled with Ai167) and observed similar connection probabilities across all three groups (chi-squared p=0.18; Rorb:Ai167: 10.7%, 110 connections/1022 probed, Rorb:AAV: 7.8%, 38 connections/489 probed, Scnn1a:Ai167: 9.1%, 12 connections/132 probed).

Excitatory projections from L5 to L2/3 have been described as tightly focused and weak (*Thomson and Bannister, 1998*; *Reyes and Sakmann, 1999*; *Thomson and Bannister, 2003*; *Lefort et al., 2009*; *Jiang et al., 2015*) relative to inputs from L4 (*Douglas and Martin, 2004*). In this context, we were surprised to find many connections to L2/3 PCs from Rorb neurons in L5 using both transgenic and AAV-mediated ChrimsonR expression (*Figure 4B*). Furthermore, connection probability from Rorb neurons located in L5 was only one-third lower than the connectivity of Rorb neurons in L4 (*Figure 4B*; combining Ai167 and AAV data within 300 μm horizontal distance, Rorb L4: 9.8%, 148 connections/1511 probed, Rorb L5: 6.9%, 71 connections /1028 probed, 6.9%, Fisher's exact p=0.01).

Many of the L5 Rorb connections were found near the L4/L5 border (*Figure 4B*), leading us to ask if these connections arose from 'misplaced' L4 projection neurons. However, transcriptomic characterization of Rorb-Cre-labeled cells suggests that the gene expression patterns of at least a subset of these neurons resemble IT-type L5 neurons (*Tasic et al., 2016*; *Tasic et al., 2018*). Additionally, the axonal projections of Rorb neurons extend outside of primary visual cortex to higher visual areas and the striatum (*Harris et al., 2019*). Therefore, it is possible that Rorb-Cre expression in L5 revealed local projections from L5 IT neurons (that are primarily associated with inter-areal projections). To more deliberately measure the local projections of L5 IT neurons, we used the Tlx3-Cre line (*Kim et al., 2015*) to drive soma-targeted ChrimsonR expression. *Figure 5A* shows an example experiment in which combining multicellular recording with photostimulation revealed divergence from a Tlx3-labeled neuron to two L2/3 PCs (photostimulus 2). Across experiments, we observed Tlx3 to L2/3 PC connectivity from a slightly lower but statistically similar fraction of tested connections compared to stimulation of L5 Rorb neurons (*Figure 5B*; Tlx3 all data: 5.0%, 86 connections/1742 probed; L5 Tlx3 within 300 μm horizontal distance: 5.8%, 61 connections/1052 probed; Fisher's exact comparing L5 Rorb to L5 Tlx3, within 300 μm horizontal distance p=0.32). In total, L5 to L2/3 PC connectivity was assayed via transgenic ChrimsonR expression driven by Rorb-Cre, as well as soma-targeted ChrimsonR expression driven by either Rorb-Cre or Tlx3-Cre with AAV. All three experiments suggest that L5 IT-type neurons are a significant source of excitatory input to L2/3 PCs in young adult mouse V1.

Excitatory L6 corticothalamic (CT) neurons display limited axonal projections to superficial cortical layers (*Kim et al., 2014*; *Bortone et al., 2014*), and stimulation of L6 by glutamate uncaging suggests that excitatory synaptic input from L6 to L2/3 is weak (*Xu et al., 2016*). We assayed connectivity from L6 CT neurons to L2/3 PCs using two-photon stimulation of ChrimsonR-expressing cells in the Ntsr1-Cre line (*Vélez-Fort et al., 2014*). Consistent with the weak axonal projections to L2/3 (*Kim et al., 2014*; *Bortone et al., 2014*), we found a single excitatory connection from 1145 possible connections tested (*Figure 5B*). In further support of low connectivity from Ntsr1-labeled neurons relative to Rorb- and Tlx3-labeled neurons, responses to one-photon stimulation were dramatically smaller in Ntsr1 experiments than Rorb or Tlx3 experiments (Ntsr1: 0.12 ± 0.17 mV; Rorb: 9.09 ± 7.13 mV; Tlx3: 9.43 ± 6.21 mV; *Figure 5—figure supplement 1*).

To determine if translaminar excitatory connections display subclass-specific targeting, we measured connectivity from Rorb and Tlx3-labeled cells in L4 and L5 to L2/3 FSIs. In contrast to intralaminar excitatory connections, which displayed a strong bias for FSIs over PCs (*Figure 3B*), translaminar connection probabilities from excitatory neurons labeled by Rorb and Tlx3 were similar for L2/3 PCs and L2/3 FSIs (Rorb: 7.3%, 24 connections/331 probed, Tlx3: 7.1%, 7 connections/99 probed, all measures for connections within 300 μm horizontal distance; *Figure 5C and D*). The data suggest that L2/3 PCs and L2/3 FSIs receive a combination of intralaminar and translaminar excitatory input; however, L2/3 FSIs receive a larger fraction of their excitatory input from intralaminar neighbors.

Excitatory connections to L2/3 were observed over a wide range of horizontal offsets, including some translaminar connections with over 300 μm of horizontal distance between cells (*Figures 4B and 5B*). Excitatory connectivity from excitatory Cre lines to L2/3 PCs displayed a significant dependence on the horizontal distance between pairs (*Figure 5D*; Penk: p=0.013, n = 895, L4 Rorb: p=0.002, n = 1712; Tlx3: p=0.003, n = 1724, chi-squared test, measured in 100 μm bins up to 600 μm). We were unable to measure the horizontal dependence of excitatory connectivity to L2/3 FSIs as completely due to a smaller number of recordings and fewer connections tested (*Figure 5D*). Future experiments

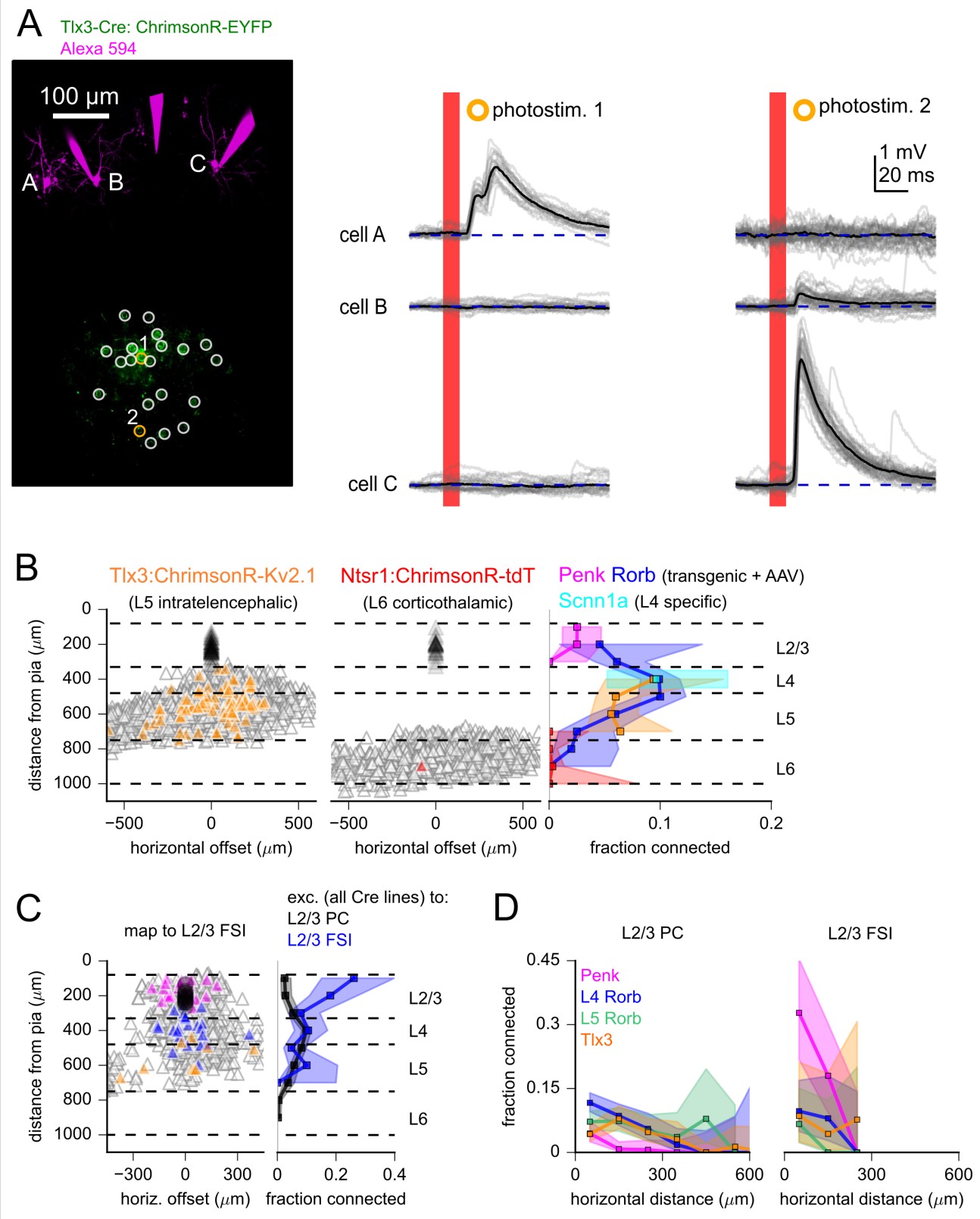

**Figure 5.** Measurement of translaminar excitatory connectivity to L2/3. (**A**) Example experiment measuring connectivity from Tlx3-labeled excitatory neurons to three L2/3 pyramidal cells (PCs) (cells A–C). Left: flattened z-stack of recorded L2/3 PCs filled with Alexa 594 (magenta) and EYFP signal from soma-targeted ChrimsonR-expressing neurons (green). Yellow circles: photostimulated cells with responses plotted in the adjacent panels. White circles: photostimulated cells with responses not shown. Right: responses to photostimulation. Photostimulus occurred during the red bar, black lines

*Figure 5 continued on next page*

*Figure 5 continued*

are an average of individual sweeps (gray). (**B**) Summary maps for experiments with Tlx3-Cre (left, n = 1734 connections probed) and Ntsr1-Cre (center, n = 1145 connections probed) mice. Locations of neurons are plotted according to their distance from the pia and the horizontal distance between cells with the following color scheme: blue, postsynaptic L2/3 PCs (Tlx3 n = 46, Ntrs1 n = 23); gray, photostimulated neurons that did not evoke a synaptic response; orange/red, stimulated cells that produced an EPSP using Tlx3-Cre or Ntsr1-Cre. Dashed lines represent approximate layer boundaries. Positive horizontal distances correspond to presynaptic cells posterior to the postsynaptic cell. Right: the measured connection probability plotted against the distance of the presynaptic cell from the pia to L2/3 PCs for the indicated Cre lines. (**C**, left) Summary map for experiments measuring excitatory inputs to L2/3 fast-spiking interneurons (FSIs) using either Penk (magenta), Rorb (blue), or Tlx3 (orange) Cre lines to drive opsin expression (n = 648 connections probed, 63 connections found, from 19 postsynaptic recordings). Right: the connection rate of excitatory neurons to L2/3 PCs (black) and L2/3 FSIs (blue). Data across excitatory Cre lines were pooled in this plot. Measurement of connection rate versus pia distance was limited to cells < 300 µm horizontal distance (panels **B** and **C**). (**D**) Connection rate plotted against the absolute value of the horizontal distance between pre- and postsynaptic neurons for the indicated Cre lines to either L2/3 PCs (left) or L2/3 FSIs (right). Shading indicates 95% confidence intervals in panels (**B**) – (**D**). The connection probability and associated confidence intervals were not drawn if fewer than 20 connections were probed within a distance bin.

The online version of this article includes the following source data and figure supplement(s) for figure 5:

**Source data 1.** Raw values for plots in *Figure 5*.

**Figure supplement 1.** One-photon-evoked synaptic responses.

**Figure supplement 2.** Evaluation of excitatory connectivity.

**Figure supplement 3.** The timing of two-photon-evoked excitatory responses is consistent with monosynaptic connectivity.

may benefit from genetic labeling of Pvalb and other interneuron subclasses using a complementary recombinase to allow targeted patching of less common postsynaptic targets.

Human annotation of photo-evoked connections took place over the course of the study, and it was impractical to blind the annotator to experimental parameters such as the Cre line or the cell class of the postsynaptic neurons. We questioned to what extent unintended annotator bias or variability in human attention could have influenced the results. Therefore, we employed two complementary, quantitative methods to measure connectivity (see 'Analysis of connectivity' for details). The first used classification by a support vector machine (SVM) trained on a randomly selected subset of the annotated data (*Figure 5—figure supplement 2*). The second approach used measurement of the peak photo-evoked voltage response relative to background variance (z-score) and was completely independent of human annotation. For the majority of tested connections, both methods aligned with human annotation. While discrepancies were observed for a small number of individual connections, the spatial distribution and cell-class preferences of excitatory connectivity were highly similar at the population level (*Figure 5—figure supplement 2*). The quantitative approaches served to increase our confidence in the patterns of connectivity reported in this study, and in a small number of cases helped to identify errors in human annotation (detailed in 'Materials and methods'). We anticipate that further development will allow more automated annotation of mapping data and real-time analysis during data acquisition that could facilitate collection of additional photostimulus responses for putative connections.

Given that two-photon stimulation may activate multiple ChrimsonR-expressing cells, we wondered if some of the observed excitatory synaptic responses could be polysynaptic; that is, the result of AP firing in an intermediate cell that is connected to the patched cell rather than a direct monosynaptic connection from the targeted presynaptic cell. Multiple lines of evidence lead us to believe that this is not common in our dataset. First, experiments in which optically identified presynaptic partners were subsequently patched suggest that the majority of observed synaptic responses are the result of spiking in the targeted cell (*Figure 4A*, *Figure 4—figure supplement 1*). Additionally, the latencies of optically evoked EPSPs are consistent with our measurements of light-evoked spiking and display low jitter (*Figure 5—figure supplement 3*). Finally, two-photon-evoked responses in patched cells rarely exceeded 1 mV or approached the threshold for AP generation (described further in Figure 7). While one-photon-evoked synaptic responses were larger (due to simultaneous activation of many cells in a field of view), these relatively strong stimuli only rarely generated APs in PCs that were not ChrimsonR-positive (i.e., directly photosensitive; *Figure 5—figure supplement 1*). However, the possibility remains that intermediate cells that mediate polysynaptic connections reside in other areas of the slice or were otherwise not sampled among the cells patched in our experiments.

## Translaminar and within-layer inhibition by Pvalb and Sst neurons

How, if at all, does the pattern of vertical inhibitory input to L2/3 PCs resemble that of vertical excitatory input to the same cells? The diverse axonal projection patterns of interneurons and reports of translaminar synaptic inhibition (*Kapfer et al., 2007*; *Silberberg and Markram, 2007*; *Nigro et al., 2018*; *Gouwens et al., 2019*; *Naka et al., 2019*; *Gouwens et al., 2020*) inspired us to measure Pvalb and Sst-mediated inhibition to L2/3 PCs and compare connection probabilities and strengths between intralaminar and translaminar inhibition. An example experiment utilizing AAV-mediated expression of soma-targeted-ChrimsonR shows a high rate of connectivity from Sst neurons to nearby L2/3 PCs (*Figure 6A*). Consistent with previous reports (*Fino and Yuste, 2011*; *Packer and Yuste, 2011*), experiments using AAV-mediated ChrimsonR expression in either Pvalb or Sst interneurons demonstrated a high probability of within-layer inhibitory connectivity that had a steep dependence on the horizontal distance between cells (*Figure 6B and C*). L2/3 PCs were innervated by Pvalb and Sst interneurons within L2/3 at similar rates (<200 µm horizontal distance, Pvalb:AAV: 22.4%, 35 connections/156 probed, Sst:AAV: 28.4%, 60 connections/211 probed, p=0.23, Fisher's exact). In addition to within-layer inhibition, translaminar inhibition of L2/3 PCs was observed from both Pvalb and Sst interneurons in L4 and L5 (*Figure 6B and C*). While a few L5 Pvalb to L2/3 PC connections were observed, connections from L5 Sst cells were significantly more common (<200 µm horizontal distance, Pvalb:AAV: 0.8%, 2 connections/259 probed, Sst:AAV: 6.0%, 16 connections/252 probed, p=1.1e-3, Fisher's exact). The vertical and horizontal distributions of Pvalb and Sst interneuron connections were similar when connectivity was determined by either visual annotation of photoresponses, machine categorization, or measurement of maximum voltage responses relative to background variance (*Figure 6—figure supplement 1*). Thus, inhibition of L2/3 PCs by Pvalb and Sst interneurons originates largely, but not exclusively, from interneurons within the same layer.

We wondered to what extent our results might be influenced by the use of viral transfection that may exhibit tropism for specific cell types, toxicity from viral load, or side effects from the transfection procedure. In contrast, many previous studies of synaptic connectivity have utilized transgenic labeling of interneuron subclasses that may present other caveats, such as the developmental history of Cre expression. To explore potential differences between labeling strategies, we measured Sst to L2/3 PC connectivity using the Ai167 transgenic line. Similar to experiments using Sst-Cre with AAV, we observed dense intralaminar inhibition in L2/3 as well as translaminar connections from labeled neurons in L4 and L5 (*Figure 6B and C*). Notably, there were fewer labeled cells in L4 when using Ai167, leading to fewer connections probed (*Figure 6B*); however, the fractions of photostimulated L4 Sst cells that generated a synaptic response were statistically similar (Sst:Ai167: 10.5%, 2 connections/19 probed; Sst:AAV: 20.6%, 51 connections/248 probed; p=0.38, Fisher's exact, all measures within 200 µm horizontal distance). Therefore, transgenic or AAV-driven ChrimsonR expression using Sst-Cre resulted in anatomically overlapping but unique distributions of labeled cells. However, Sst-labeled cells in common anatomical locations displayed similar connectivity to L2/3 PCs in experiments using either Ai167 or AAV-mediated ChrimsonR expression.

Unlike excitatory connections, which often spanned hundreds of microns in the horizontal (anterior-posterior) dimension (*Figure 5D*), inhibitory connections to L2/3 PCs were more spatially confined (*Figure 6B and C*). The inhibitory connection probability displayed a steep decline with horizontal distance for both Pvalb and Sst connections (*Figure 6C*). A recent study using rabies virus to reveal the spatial distribution of presynaptic ensembles to L2/3 PCs demonstrated a broader distribution of excitatory inputs relative to inhibitory inputs (*Rossi et al., 2020*). This pattern is further recapitulated in our data when we combine connectivity measured across excitatory and inhibitory Cre lines (*Figure 6—figure supplement 2*). Therefore, both anatomical and physiological surveys reveal differences in the spatial distribution of excitatory and inhibitory connections to L2/3 PCs.

Do translaminar inhibitory projections to L2/3 specifically target PCs, or alternatively, do translaminar inhibitory projections also form connections onto L2/3 interneurons? To address this, we photostimulated potential intralaminar and translaminar inhibitory inputs from Pvalb and Sst-labeled interneurons to L2/3 FSIs (postsynaptic FSIs were also genetically labeled in Pvalb-Cre experiments). Inhibitory connections to L2/3 FSIs were observed from presynaptic Sst- and Pvalb- interneurons in layers 2/3, 4, and 5 (*Figure 6D and E*). Therefore, translaminar inhibition to L2/3 is not specific to PCs and cross-layer inhibitory connections will also shape the activity of FSIs. In total, experiments

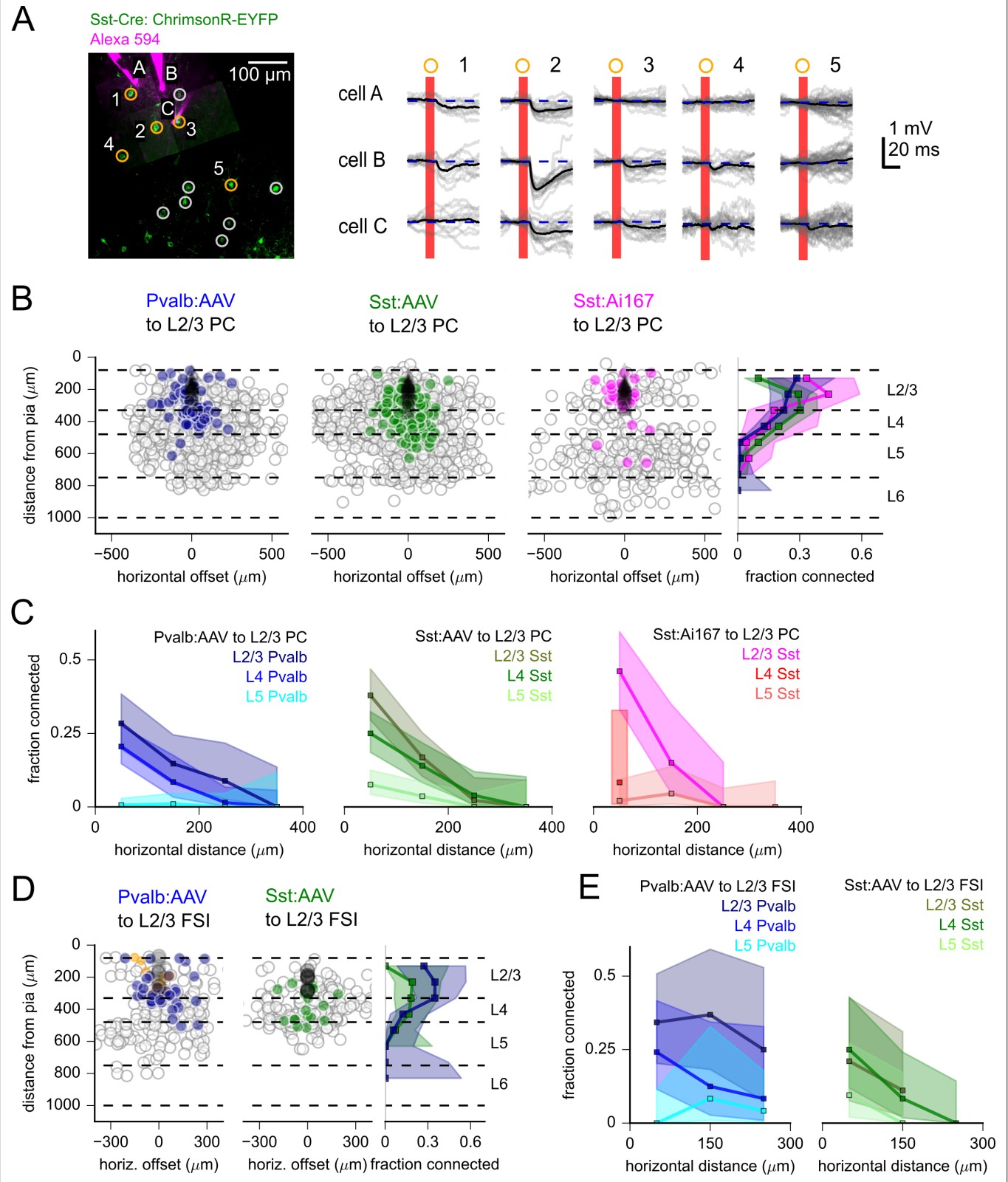

**Figure 6.** Measurement of inhibitory connectivity to L2/3 pyramidal cells (PCs). (**A**) Example experiment measuring connectivity from Sst neurons to three L2/3 PCs (cells A–C). Left: flattened z-stack of recorded L2/3 PCs filled with Alexa 594 (magenta) and EYFP signal from soma-targeted ChrimsonR-expressing neurons (green). Yellow circles: photostimulated cells with responses plotted in right panels. Responses following stimulation of cells in white circles are not shown. Right: responses to photostimulation of the indicated cells. Postsynaptic cells were depolarized to –55 mV with automated bias

*Figure 6 continued on next page*

*Figure 6 continued*

current to increase the driving force of inhibitory currents. Photostimulation occurred during the red bar, black lines are an average of individual sweeps (gray). (**B**) Summary maps of inhibitory connections to L2/3 PCs for experiments with Pvalb:AAV-injected (902 connections probed, 33 postsynaptic recordings), Sst:AAV-injected (1052 connections probed, 49 postsynaptic recordings), and Sst:Ai167 mice (360 connections probed, 39 postsynaptic recordings). Locations of neurons are plotted according to their distance from the pia and the horizontal distance between cells with the following color scheme: black, postsynaptic L2/3 PCs; gray, photostimulated neurons that did not evoke a synaptic response in the recorded cell; blue/green/magenta, stimulated cells that produced an IPSP using Pvalb-Cre:AAV/Sst-Cre:AAV/Sst:Ai167-mediated ChrimsonR expression. Dashed lines represent approximate layer boundaries. Positive horizontal offsets correspond to presynaptic cells posterior to the postsynaptic cell. Right: measured connection probability plotted against the distance of the presynaptic cell from the pia for the three datasets. Measurement of connection probability versus pia distance was limited to cells < 200 µm horizontal distance. (**C**) Connection probability versus horizontal distance between pre- and postsynaptic cells for each dataset, separated by the cortical layer of the presynaptic cell. (**D**) Summary maps of inhibitory connections to L2/3 fast-spiking interneurons (FSIs) (Pvalb: 201 connections probed, 12 postsynaptic recordings; Sst: 166 connections probed, 7 postsynaptic recordings) following the same scheme as panel (**B**), with the addition of orange markers to indicated stimulus locations that produced a direct stimulus artifact due to expression of ChrimsonR in the patched, Pvalb-labeled neurons (13 photostimulus locations). Shading in panels (**B**) – (**F**) represents 95% confidence intervals. The fraction of cells connected and associated confidence intervals were not drawn if fewer than 10 connections were probed within a distance bin.

The online version of this article includes the following source data and figure supplement(s) for figure 6:

**Source data 1.** Raw values for plots in *Figure 6*.

**Figure supplement 1.** Evaluation of inhibitory connectivity.

**Figure supplement 2.** Spatial distribution of excitatory and inhibitory connectivity to L2/3 pyramidal cell (PCs).

measuring inhibitory connections via two-photon optogenetics demonstrate that translaminar inhibition is a feature of multiple pre- and postsynaptic subclasses in adult mouse V1.

## Features of synaptic connections vary by pre- and postsynaptic subclass

As shown in example experiments in *Figures 2–5*, amplitudes of light-evoked synaptic responses varied dramatically within and across subclasses of connections (*Figure 7A*). For most classes of connections, the distribution of PSP amplitudes displayed a mean larger than the median due to a small minority of strong connections (*Table 2*). Skewed distributions such as these have been described in previous large-scale surveys of synaptic properties made using paired recordings (*Sayer et al., 1990*; *Markram et al., 1997*; *Feldmeyer et al., 2002*; *Song et al., 2005*; *Lefort et al., 2009*; *Cossell et al., 2015*). In further agreement with previous studies, we observed low trial-to-trial variability in the amplitudes of strong connections, and an inverse correlation between the coefficient of variation (CV) and PSP amplitude (*Figure 7B*). It has been suggested that these strong and reliable connections, while uncommon, can powerfully influence network activity (*Song et al., 2005*; *Lefort et al., 2009*).

Is the strength of a synaptic connection influenced by the subclass of the pre- and/or postsynaptic cell? To address this question, we compared PSP amplitudes measured in L2/3 PCs and L2/3 FSIs for each Cre-labeled presynaptic subclass (*Figure 7A*, *Figure 7—figure supplement 2*). In some cases, the strength of outgoing connections from a single Cre-defined presynaptic population differed according to the postsynaptic subclass. For example, L2/3 excitatory neurons labeled by the Penk-Cre line generated dramatically larger EPSPs in L2/3 FSIs than in L2/3 PCs (*Figure 7A*, *Figure 7—figure supplement 2A*; p=2.1e-4, post hoc Dunn's). In other cases, the properties of connections to a postsynaptic class differed according to the presynaptic subclass. For example, IPSPs measured from Pvalb interneurons to L2/3 PCs were stronger (*Figure 7A*), and displayed faster rise times (*Figure 7C*), than IPSPs measured from Sst interneurons to L2/3 PCs (amplitude: p=1.5e-6, rise time: p=5.6e-4, post hoc Dunn's; *Figure 7—figure supplement 2B and D*). These observations are consistent with previously observed biases in subcellular compartments targeted by each interneuron subclass (*Tremblay et al., 2016*). Namely, given that Pvalb interneurons target the soma and proximal processes of postsynaptic cells, Pvalb-mediated IPSPs will experience less dendritic filtering than IPSPs from Sst interneurons that target more distal dendritic regions. However, other mechanisms such as differences in postsynaptic receptor composition and transmitter release mechanisms may also contribute to the observed differences in PSP properties.

Experiments using Rorb-Cre and Sst-Cre with either Ai167 or viral expression of ChrimsonR led to overlapping but unique anatomical distributions of labeled cells; however, measures of connectivity at cortical depths labeled by both methods were not statistically different (*Figure 4*, *Figure 6B and*

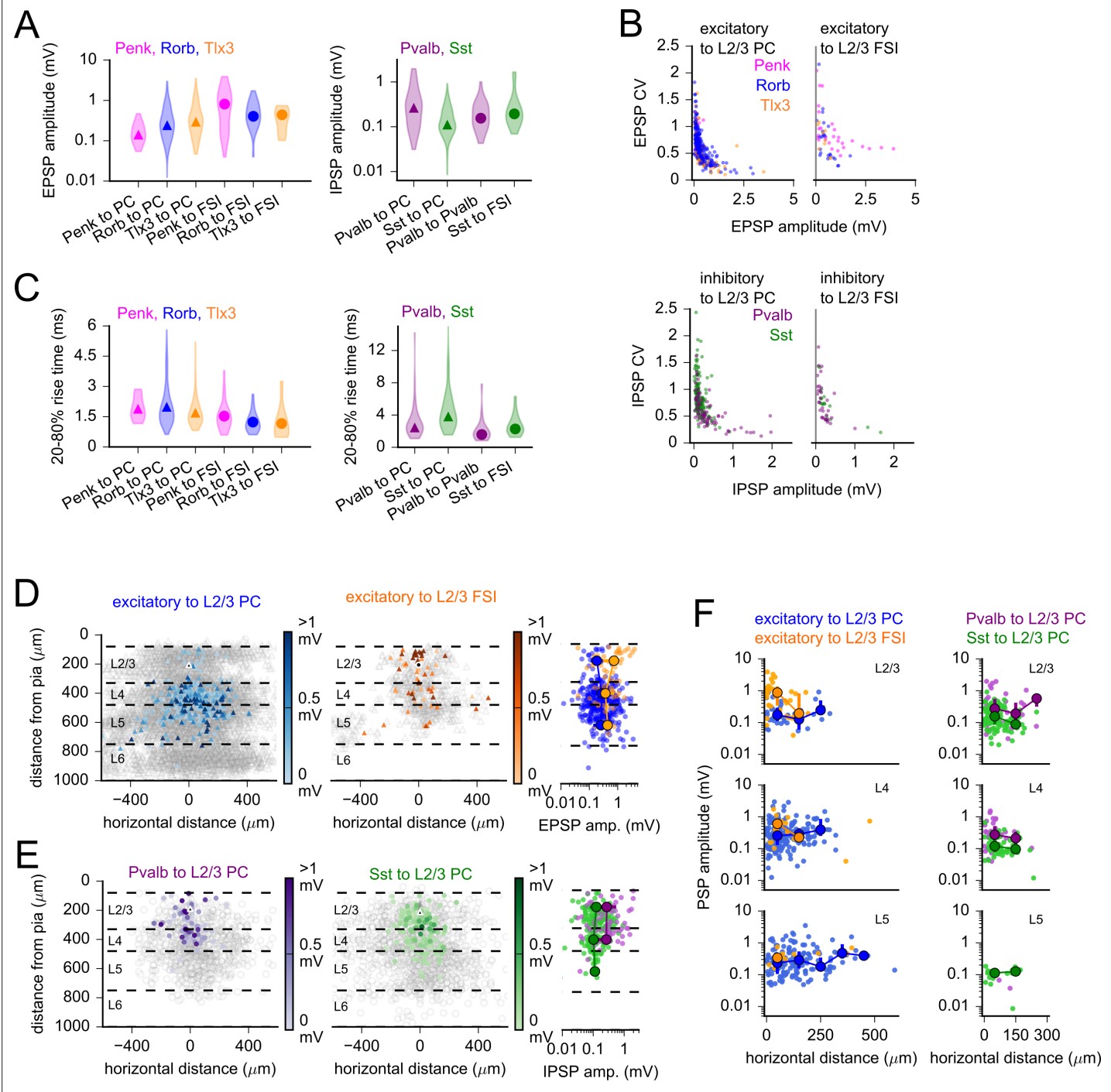

**Figure 7.** Postsynaptic potential (PSP) properties across subclasses and space. (**A**) Violin plots of PSP amplitude distributions for excitatory (left) and inhibitory (right) Cre lines. Markers represent median values. (**B**) Coefficient of variation (CV) plotted against mean PSP amplitude for all connections of indicated subclass. Excitatory to L2/3 pyramidal cell (PC): Spearman's $R = –0.76$, p=5e-67; excitatory to L2/3 fast-spiking interneuron (FSI): $R = –0.58$, p=1.2e-6; inhibitory to L2/3 PC: $R = –0.65$, p=7.2e-30; inhibitory to L2/3 FSI: $R = –0.79$, p=2.2e-12. (**C**) Violin plots of PSP rise time distributions for excitatory (left) and inhibitory (right) Cre lines. Markers represent median values. (**D**, left) Heatmaps of measured EPSP amplitudes plotted according to horizontal distance between cells and the distance of the presynaptic neuron from the pia. Dashed lines represent approximate layer boundaries. Black markers represent the average location of all postsynaptic cells. Right: EPSP amplitudes plotted against the distance of the presynaptic neuron from the pia. Larger markers represent median values for data binned by layer (markers are not drawn if fewer than three connections were measured within a distance bin). Error bars represent interquartile ranges (25–75% of data in each distance bin). Some error bars are smaller than the markers. (**E**) Same as panel (**D**) for the labeled inhibitory connection classes. (**F**) PSP amplitudes versus horizontal distance between cells for the indicated subclasses of

*Figure 7 continued on next page*

*Figure 7 continued*

excitatory (left) and inhibitory (right) connections. Plots are separated by the layer of origin of the presynaptic cell (see labels in top-right corners of each plot). Larger markers represent median and interquartile ranges as described for panel (**D**). Bin width is 100 µm.

The online version of this article includes the following source data and figure supplement(s) for figure 7:

**Source data 1.** Raw values for plots in *Figure 7*.

**Figure supplement 1.** Strategy for measuring first response peak from photoresponses with multiple postsynaptic potentials (PSPs).

**Figure supplement 2.** Statistical comparisons of postsynaptic potential (PSP) properties.

**Figure supplement 3.** Comparison of postsynaptic potential (PSP) properties by opsin expression method.

*C*). Could comparison of PSP properties reveal differences between the populations of cells labeled by each expression strategy that measures of connectivity did not? We compared the amplitudes, rise times, and CV of synaptic responses measured using Ai167 and AAV-mediated ChrimsonR expression for both Rorb-labeled and Sst-labeled presynaptic neurons (*Figure 7—figure supplement 3*). We did not observe any significant differences, but note that inhibitory connections measured using Sst:Ai167 did trend towards larger amplitude responses compared to those measured using Sst:AAV (*Figure 7—figure supplement 3B*). This may be indicative of higher off-target labeling of Pvalb interneurons in transgenic Sst:Ai167 animals due to the expression of *Sst* in early development (*Hu et al., 2013*). However, we observed greater differences between Cre lines than between expression strategies (*Figure 7—figure supplement 3*). Based on these observations, we pooled data acquired using Ai167 and AAV for comparisons between presynaptic subclasses.

To what extent might the relative locations of pre- and postsynaptic cells influence connection strength? We examined synaptic connection strength as a function of the horizontal distance between connected cells and the distance of the presynaptic neuron from the pia (*Figure 7D–F*). Within Cre lines, we observed little to no correlation between PSP amplitude and the horizontal distance between cells (*Table 3*). Differences in excitatory connection amplitudes according to the cortical layer of the presynaptic cell largely recapitulated differences observed when measuring differences across excitatory Cre lines (*Figure 7D*). For L2/3 PCs, EPSP amplitudes were larger for translaminar connections than for intralaminar connections (*Figure 7D*, blue markers; p=0.017, Mann–Whitney *U*). Excitatory inputs to L2/3 FSIs displayed the opposite pattern – intralaminar excitatory connections were stronger than translaminar excitatory connections (*Figure 7D*, orange markers; p=0.035, Mann–Whitney *U*). For both Pvalb and Sst interneurons, the strengths of inhibitory connections were similar when grouped according to presynaptic layer (*Figure 7E*, Pvalb: p=0.10, Sst: p=0.38, Kruskal–Wallis).

**Table 2.** Properties of two-photon-evoked PSPs.

| Connection class | Amp. mean (mV) | Amp. median (mV) | Amp. sd (mV) | Amp. skew | n | CV | 20–80% rise (ms) |
|---|---|---|---|---|---|---|---|
| Penk to L2/3 PC | 0.17 | 0.15 | 0.1 | 1.3 | 16 | 0.8 | 1.92 |
| Rorb to L2/3 PC | 0.38 | 0.25 | 0.42 | 3.02 | 245 | 0.63 | 2.01 |
| Tlx3 to L2/3 PC | 0.46 | 0.30 | 0.51 | 3.2 | 85 | 0.54 | 1.72 |
| Ntsr1 to L2/3 PC | 0.09 | 0.09 | | | 1 | 1.09 | 1.19 |
| Penk to L2/3 FSI | 1.00 | 0.81 | 0.99 | 1.41 | 28 | 0.9 | 1.52 |
| Rorb to L2/3 FSI | 0.55 | 0.40 | 0.38 | 1.3 | 24 | 0.75 | 1.23 |
| Tlx3 to L2/3 FSI | 0.4 | 0.44 | 0.22 | 0.18 | 9 | 0.75 | 1.16 |
| Pvalb to L2/3 PC | 0.41 | 0.27 | 0.45 | 1.97 | 77 | 0.58 | 3.04 |
| Sst to L2/3 PC | 0.16 | 0.11 | 0.13 | 2.35 | 162 | 0.85 | 4.51 |
| Pvalb to L2/3 FSI | 0.21 | 0.16 | 0.18 | 2.61 | 34 | 0.86 | 2.03 |
| Sst to L2/3 FSI | 0.34 | 0.19 | 0.42 | 2.38 | 18 | 0.75 | 2.73 |

PSP: postsynaptic potential; PC: pyramidal cell; FSI: fast-spiking interneuron.

**Table 3.** Distance dependence of EPSP and IPSP amplitudes.

The strength and significance of correlations between PSP amplitudes and horizontal distance or the presynaptic neuron's distance from the pia. Benjamini–Hochberg (BH)-corrected p-values were calculated using a false discovery rate of 0.1.

| Connection class | Horizontal Spearman's R | Horizontal Spearman's p-value | Horizontal BH-corrected p-value | Presyn. pia Spearman's R | Presyn. pia Spearman's p-value | Presyn. pia BH-corrected p-value | n |
|---|---|---|---|---|---|---|---|
| Penk to L2/3 PC | –0.53 | 0.04 | 0.12 | –0.51 | 0.05 | 0.23 | 16 |
| Rorb to L2/3 PC | 0.08 | 0.21 | 0.27 | 0.03 | 0.65 | 0.90 | 244 |
| Tlx3 to L2/3 PC | 0.18 | 0.09 | 0.17 | –0.01 | 0.91 | 0.91 | 85 |
| Penk to L2/3 FSI | –0.35 | 0.07 | 0.17 | –0.44 | 0.02 | 0.18 | 28 |
| Rorb to L2/3 FSI | –0.27 | 0.19 | 0.27 | –0.05 | 0.80 | 0.90 | 24 |
| Tlx3 to L2/3 FSI | 0.72 | 0.03 | 0.12 | 0.57 | 0.11 | 0.35 | 9 |
| Pvalb to L2/3 PC | 0.08 | 0.51 | 0.57 | 0.17 | 0.14 | 0.35 | 77 |
| Sst to L2/3 PC | 0.19 | 0.02 | 0.12 | 0.04 | 0.58 | 0.90 | 162 |
| Pvalb to L2/3 FSI | –0.07 | 0.68 | 0.68 | 0.04 | 0.81 | 0.90 | 34 |
| Sst to L2/3 FSI | 0.40 | 0.10 | 0.17 | 0.13 | 0.62 | 0.90 | 18 |

PSP: postsynaptic potential; PC: pyramidal cell; FSI: fast-spiking interneuron.

Therefore, while translaminar vertical inhibition from Pvalb and Sst neurons may be less common than within-layer horizontal inhibition (*Figure 6*), the unitary connection strengths are similar.

## Differences in synapse dynamics across cell classes and layers

Synaptic transmission is dynamic, and its efficacy may be influenced by the recent activity of either the presynaptic or postsynaptic cell. Changes in synaptic strength over the course of milliseconds to seconds (short-term plasticity [STP]) provide diverse computational functions to synapses (*Abbott and Regehr, 2004*). The direction, magnitude, and time course of STP may vary according to the presynaptic and postsynaptic cell identities (*Reyes et al., 1998*; *Blackman et al., 2013*; *Markram et al., 2015*; *Lefort and Petersen, 2017*). To look for differences in STP among intralaminar and translaminar connections in our dataset, we focused on a subset of photoresponses that displayed two or more distinct PSPs (*Figure 8A*, additional examples in *Figures 3A and 5A*). We measured the paired-pulse ratio (PPR) for sweeps in which the estimated interspike interval was between 5 and 10 ms (see 'Materials and methods'). Most connections from Penk neurons to L2/3 PCs and FSIs displayed paired-pulse facilitation (PPR > 1) or were approximately linear (*Figure 8B*). In contrast, most translaminar excitatory connections (from Rorb and Tlx3-labeled cells) displayed paired-pulse depression; however, roughly one-quarter of translaminar excitatory connections to L2/3 PCs displayed facilitation (*Figure 8B*). Heterogeneity in the direction and magnitude of STP of translaminar connections was previously observed in paired recordings made in barrel cortex of young mice (*Lefort and Petersen, 2017*). Facilitation of synaptic responses measured from recurrent connections in L2/3 has previously been reported in both barrel and visual cortex (*Lefort and Petersen, 2017*; *Seeman et al., 2018*) – consistent with our findings using optogenetic stimulation of Penk-Cre-labeled cells.

We also measured the short-term dynamics of intralaminar and translaminar inhibition from Pvalb and Sst interneurons (*Figure 8C*). Interestingly, while the majority of Pvalb to L2/3 PC connections

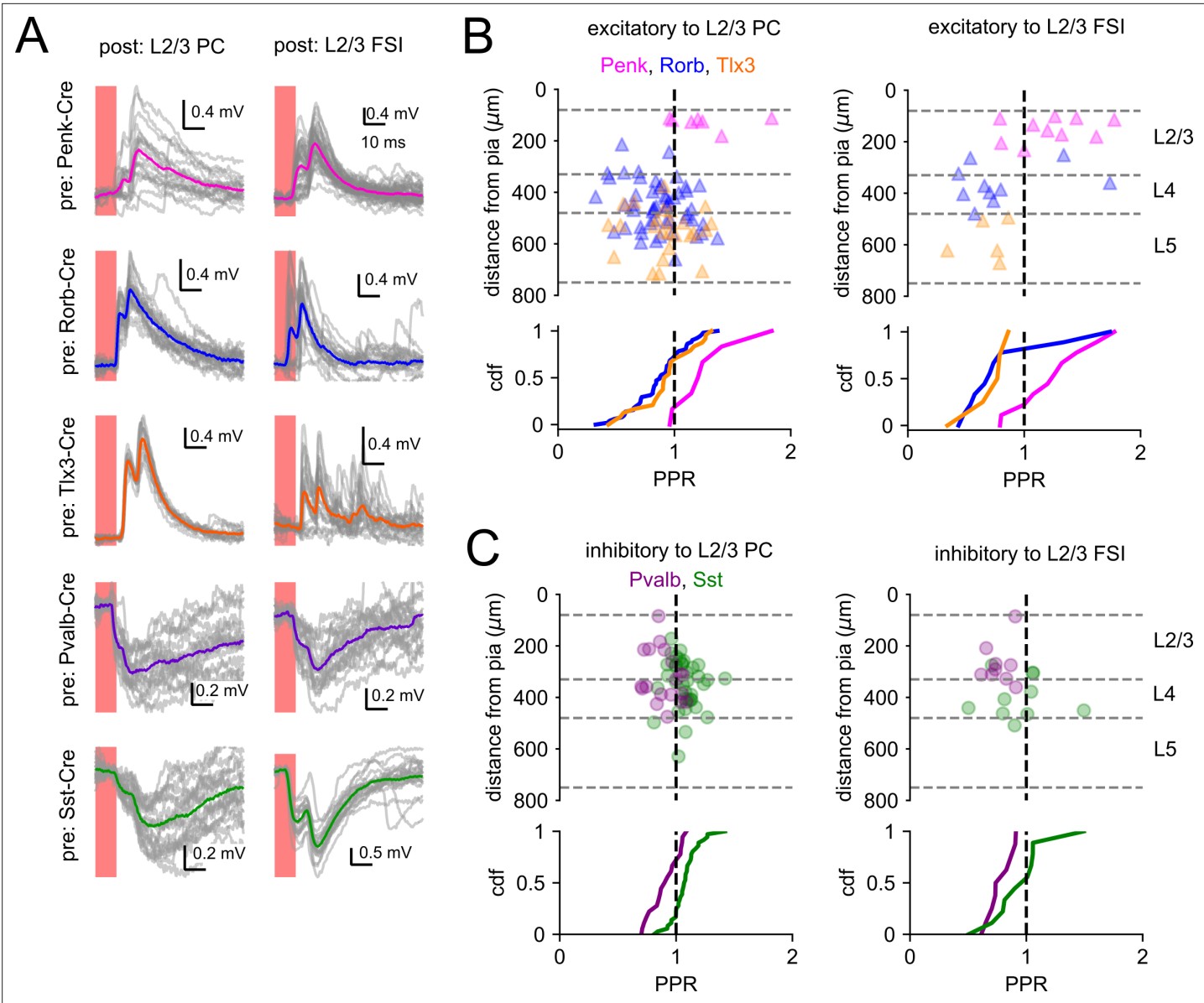

**Figure 8.** Short-term plasticity of optically evoked synaptic responses. (**A**) Examples of light-evoked synaptic responses measured from indicated presynaptic Cre lines (rows) and postsynaptic subclasses (columns). Horizontal scale bar is set to 10 ms in all panels. Vertical scale was adjusted to accommodate the amplitude of each photoresponse. Individual photoresponses are plotted in gray, and average responses are represented by colored traces. (**B**) Top: paired-pulse ratio (PPR) of excitatory connections plotted according to the distance of the presynaptic cell from the pia. The colors of markers indicate the presynaptic Cre line. The vertical dashed line corresponds to a PPR = 1. The horizontal dashed lines represent approximate layer boundaries. Bottom: cumulative distribution function (CDF) of PPR values measured for each Cre line. (**C**) PPR values measured from inhibitory synapses as in panel (**B**).

The online version of this article includes the following source data and figure supplement(s) for figure 8:

**Source data 1.** Raw values for plots in *Figure 8*.

**Figure supplement 1.** Statistical comparisons of paired-pulse ratios (PPRs).

displayed paired-pulse depression, Sst to L2/3 PC connections were approximately linear or facilitated. This pattern was observed for both intralaminar and translaminar inhibition from the two interneuron subclasses and is consistent with previously reported differences in IPSP dynamics according to presynaptic interneuron subclass (*Beierlein et al., 2003*). The PPRs measured for Sst to L2/3 FSI connections were significantly smaller than the PPRs measured for Sst to L2/3 PC connections (p=0.028, post hoc Dunn's; *Figure 8C*, *Figure 8—figure supplement 1*), illustrating an example of

target-specific synaptic plasticity (*Reyes et al., 1998*; *Blackman et al., 2013*). We acknowledge that these data present a limited view of short-term dynamics that could be more fully characterized by paired recordings of the pre- and postsynaptic cells. We did not attempt to apply trains of photo-stimuli as standard practice in our mapping experiments due to the variability we observed in the number and timing of light-evoked APs (*Figure 1*). We are optimistic that high-conductance opsins with faster kinetics will permit deeper optical interrogation of STP at unitary connections in the near future (*Mardinly et al., 2018*).

## Locations and strengths of connections sharing post- or presynaptic neurons

Two-photon stimulation allowed us to probe up to 75 potential presynaptic partners of a recorded postsynaptic cell – many more than would have been feasible by patch clamping alone – and we observed 96 examples in which multiple presynaptic neurons (up to 12 connections per cell) converged onto individual L2/3 PCs (*Figure 9A*, also see examples in Figures 2A, 3A, and 4A). We compared the observed number of convergence motifs (two presynaptic cells form a connection with one post-synaptic cell) to a simulated random network with experimentally measured distance dependence (*Figure 9—figure supplement 1*). Out of 10 possible convergence patterns (five presynaptic Cre lines to either L2/3 PCs or L2/3 FSIs), 2 were significantly more abundant than expected by chance – Rorb to L2/3 PC, and Tlx3 to L2/3 PC. In both cases, overrepresentation of convergence motifs was fairly subtle (30–35% more abundant than expected by random connectivity), but highly significant (p<0.001) due to the large number of possible two-to-one motifs examined (Rorb: n = 61,911; Tlx3: n = 33,714).

To examine the spatial patterns of convergence, we considered all potential connections that would complete a convergence motif (schematic in *Figure 9B*). To do so, we examined all unique ordered pairs of cells that were photostimulated while recording from a given postsynaptic cell. We then selected all examples in which the first cell (cell *i*) made a connection to the postsynaptic cell. From this subset, we calculated the fraction of instances in which the second presynaptic cell (cell *j*) completed the convergence motif by also connecting to the postsynaptic cell. This value represents the connection probability from cell *j*, given that cell *i* is connected to the target cell (abbreviated as prob.j|i in *Figure 9*). We measured the probability of completed convergence as a function of the horizontal distance between presynaptic cells (*Figure 9B*) and the horizontal distance between the postsynaptic cell and the potential presynaptic cell (cell *j*; *Figure 9C*). Similar to overall connectivity patterns, the probability of convergence was dependent on the horizontal distance between presynaptic pairs and the postsynaptic target. While this reinforces the observation that connectivity is most common between vertically aligned cells, we also observed instances in which presynaptic neurons separated by hundreds of microns converged onto a common L2/3 PC (*Figure 9A and B*).

Measurements of recurrent intralaminar connectivity between PCs show that strong connections are frequently observed between bidirectionally connected pairs, and the strengths of connections sharing a presynaptic or postsynaptic neuron are correlated, suggesting that strong connections are clustered to a subset of cells (*Song et al., 2005*; *Cossell et al., 2015*). To determine if this was true of connections formed on L2/3 PCs from the presynaptic Cre lines utilized in this study, we analyzed the relationship between the strengths of connections that converged onto either L2/3 PCs or L2/3 FSIs (*Figure 9—figure supplement 2*). Connections that shared a postsynaptic target displayed either weak or statistically insignificant correlations in PSP amplitudes. Therefore, while the skewed distributions of PSP amplitudes suggest a disproportionate influence of rare and strong synaptic connections (*Figure 7A*), we do not find evidence that strong synaptic connections (of the classes examined here) are biased to particular target cells.

After finding that excitatory and inhibitory inputs converge across hundreds of microns to individual targets in L2/3, we next asked if the spatial patterns of divergence from individual presynaptic neurons to multiple target cells in L2/3 are similarly broad. Across all Cre lines, we identified 82 photostimulated cells that diverged to 2–4 simultaneously recorded cells in L2/3 (*Figure 10A*, also see examples in *Figures 5A and 6A*). To measure spatial patterns of divergence, we identified all potential connections that would complete a divergence motif. This analysis follows the same logic as that used for convergence. We examined all ordered pairs of postsynaptic neurons that share a photostimulated, presynaptic partner (*Figure 10B*). We then selected all examples in which the first

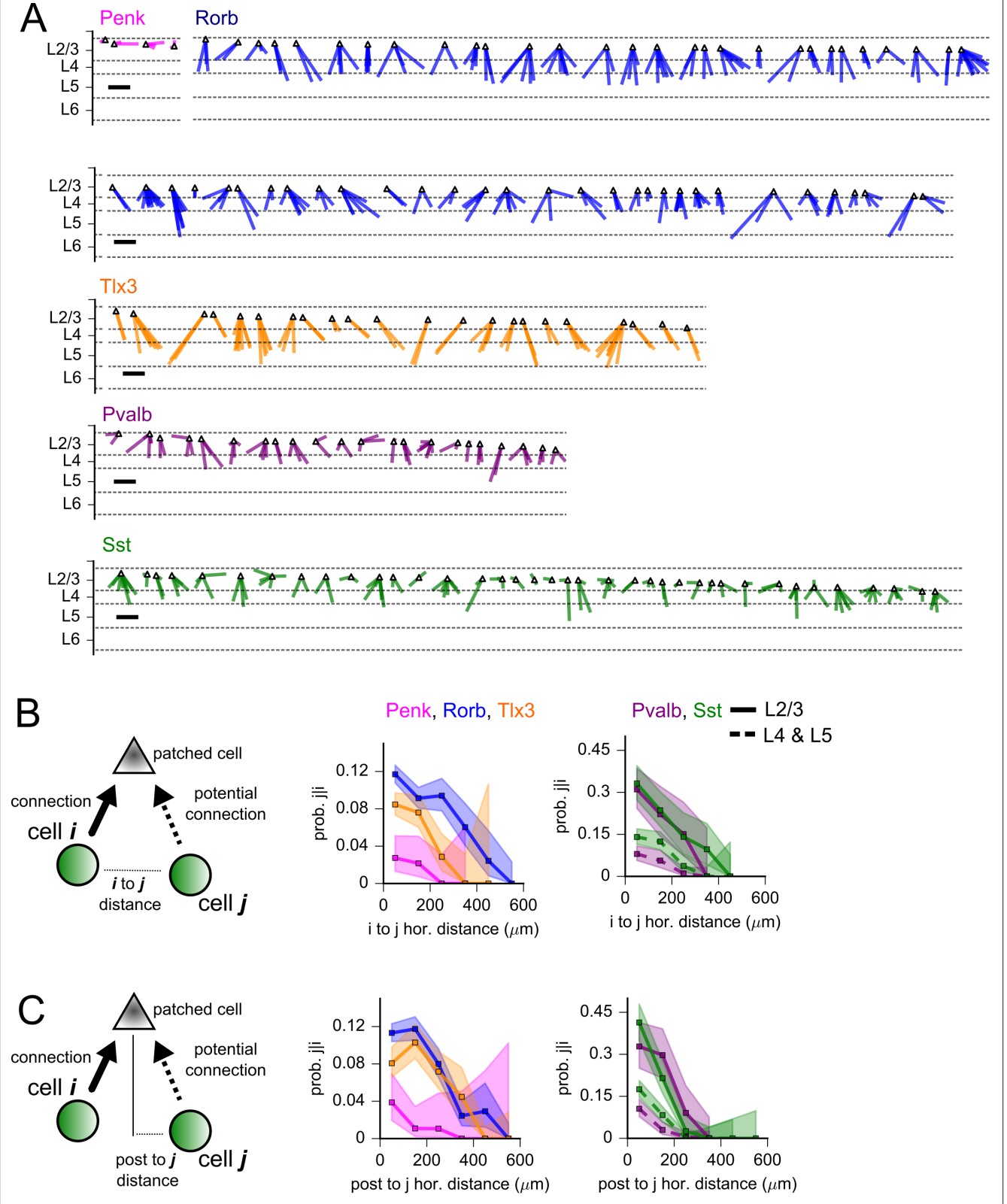

**Figure 9.** Convergence of synaptic inputs. (**A**) All identified instances of convergent inputs to L2/3 pyramidal cells (PCs) from Penk (magenta), Rorb (blue), Tlx3 (orange), Pvalb (purple), or Sst (green)-labeled neurons. The same colors are used to indicate presynaptic Cre lines throughout the figure. Lines are drawn from identified presynaptic partners to a common point representing the L2/3 PC (triangle). Cases of convergence are ordered according to the distance of the postsynaptic cell from the pia. Dashed lines represent approximate layer boundaries. Scale bar = 200 μm. (**B**, left)

*Figure 9 continued on next page*

*Figure 9 continued*

Cartoon of completed convergence analysis. We identified all ordered pairs of photostimulated neurons (cell *i* and cell *j*, green) that shared a common potential postsynaptic partner (patched cell, gray). We further required that the first photostimulated cell (cell *i*) formed a connection to the postsynaptic cell (solid arrow). From these potential convergence motifs, we measured the probability that the second photostimulated cell (cell *j*) also forms a connection with the recorded postsynaptic cell. Middle: for excitatory Cre lines, the probability that cell *j* completes a convergence motif (Pj|i) plotted against the horizontal distance between photostimulated cells. Right: same as middle panel for inhibitory Cre lines. Data from Pvalb and Sst were further split according to the layer of the presynaptic interneurons (solid lines represent intralaminar connections, dashed lines represent translaminar connections). (**C**, left) Cartoon of completed convergence as in panel (**B**). Middle/right: Pj|i plotted against the horizontal distance between presynaptic cell *j* and the postsynaptic cell. Shading represents 95% confidence intervals. The fraction of cells connected and associated confidence intervals were not drawn if fewer than 20 connections were probed within a distance bin.

The online version of this article includes the following source data and figure supplement(s) for figure 9:

**Source data 1.** Raw values for plots in *Figure 9*.

**Figure supplement 1.** Comparison of the number of observed convergence motifs to a random network.

**Figure supplement 2.** Scatter plots of incoming connections strengths for all pairs of cells sharing a postsynaptic partner for each Cre line and postsynaptic cell class.

cell (cell *i*) received a connection from the presynaptic neuron. From this subset, we calculated the fraction of pairs in which the divergence motif was completed; that is, the presynaptic cell also formed a connection to cell *j*. This value represents the connection probability to cell *j* given that cell *i* receives a connection from the presynaptic cell (abbreviated as prob. j|i in *Figure 10*).

We first explored how the probability of divergence depends on the distance between the postsynaptic neurons. We expected this probability to decrease with greater intersomatic distance as it would be less likely that distant postsynaptic cells both reside within a common presynaptic axonal arbor. The process of in vitro slicing could amplify this pattern if disparate regions of an axon are more likely to become severed than adjacent regions. It was therefore surprising to find that the probability of divergence did not vary with total intersomatic distance for excitatory Cre lines (*Figure 10B*; Fisher's exact test; Penk: p=0.29, n = 3/68; Rorb: p=0.43, n = 71/280; Tlx3: p=0.22, n = 16/103; n reported as divergence motifs/motifs in which cell *i* received a connection, across all distances). The data illustrate that single excitatory axons contact L2/3 neurons separated by hundreds of microns and argue that excitatory neurons display target specificity beyond spatial proximity. In contrast to excitatory neurons, both Pvalb and Sst interneurons displayed a decrease in the likelihood of completed divergence as the distance between postsynaptic targets increased (*Figure 10B*; Fisher's exact test; Pvalb: p=0.0014, n = 56/118; Sst: p=0.046, n = 45/114; n reported as divergence motifs/motifs in which cell *i* received a connection, across all distances). Consistent with dense local innervation from inhibitory interneurons (*Karnani et al., 2014*), postsynaptic cells near each other were likely to share common input from Pvalb and Sst interneurons (Pj|i > 0.5 at intersomatic distances < 100 μm). The relatively high, but distance-dependent, probability of inhibitory divergence suggests that the outgoing connectivity of interneurons is less selective than the output of excitatory neurons. However, the data additionally illustrate that interneurons can connect to widely distributed targets as we observed examples in which postsynaptic cells more than 200 μm apart shared divergent inhibitory input from the same interneuron.

We next examined how divergence varies according to the layer of the presynaptic neuron (*Figure 10C*). For Penk and Rorb-labeled neurons within L2/3, the probability of completed divergence was between 4% and 5% (*Figure 10C*) – similar to the connection probability measured across all tested intralaminar inputs to L2/3 PCs (*Figures 3B and 5B*). Accordingly, the number of observed divergence motifs measured using the Penk-Cre line was consistent with a random network (*Figure 10—figure supplement 1*). By contrast, for translaminar excitatory connections (from Rorb or Tlx3-labeled cells), the probability of completed divergence was two- to threefold higher than the corresponding connection probabilities measured across all tested connections (*Figure 10C*). Additionally, the numbers of observed divergence motifs from Rorb and Tlx3 neurons were dramatically and significantly greater than predicted by a random network (L4 Rorb: p<0.001, L5 Rorb: p=0.004; L5 Tlx3: p=0.007; *Figure 10—figure supplement 1*). Divergence from inhibitory interneurons was also overrepresented, and this was especially true for translaminar inhibition from both Pvalb and Sst interneurons (*Figure 10C*, *Figure 10—figure supplement 1*). Data from both excitatory and inhibitory Cre lines suggest the existence of subsets of photostimulated neurons in L4 and L5 that mediate

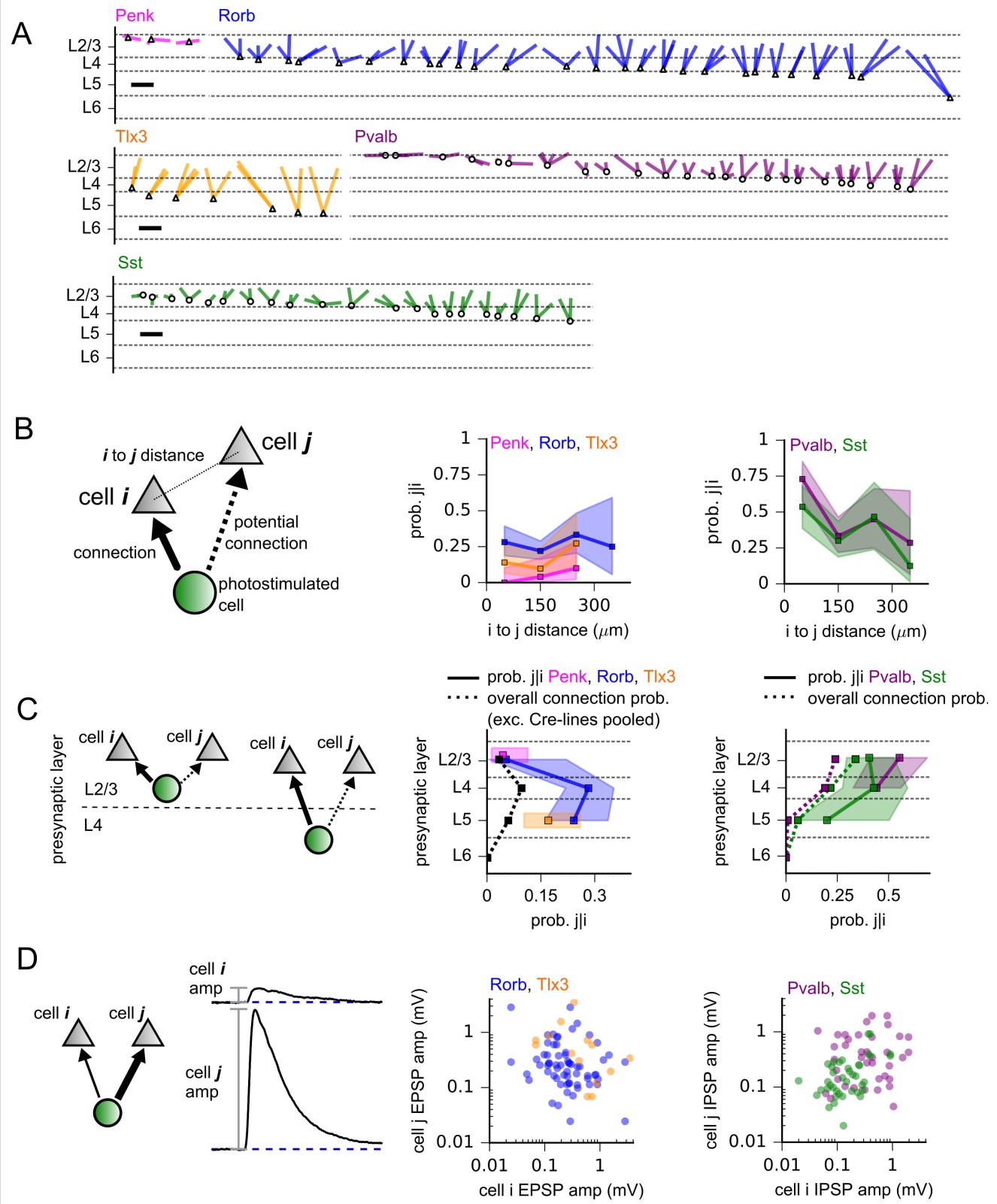

**Figure 10.** Divergence of synaptic outputs. (**A**) All identified instances of divergent output to multiple neurons in L2/3 from Penk (magenta), Rorb (blue), Tlx3 (orange), Pvalb (purple), and Sst (green)-labeled presynaptic cells. The same colors are used to indicate Cre lines throughout the figure. Lines are drawn from presynaptic neurons (noted by triangles and circles for excitatory and inhibitory Cre lines, respectively) to 2–4 points representing the recipient cells. Cases of divergence are ordered according to the distance of the presynaptic cell from the pia. Scale bars = 200 μm. (**B**, left) Cartoon of

*Figure 10 continued on next page*

Figure 10 continued

completed divergence measurement. We first considered all unique ordered pairs of patch-clamped postsynaptic neurons (gray) that shared a possible presynaptic cell assayed by photostimulation (green). We then subset the data for all pairs in which the first cell (cell *i*) received a connection from the presynaptic cell (bold arrow). From this subset, we measured the fraction of pairs in which the second postsynaptic cell (cell *j*) also received a connection from the photostimulated cell. Middle: The probability of completed divergence (prob. j|i) plotted against the total intersomatic distance (Euclidean distance in three dimensions) between postsynaptic cells for excitatory Cre lines. Experiments performed using Penk or Tlx3 Cre lines did not include recordings of postsynaptic cells separated by more than 300 μm. Right: same as the middle panel for inhibitory Cre lines. (**C**) Prob. j|i was calculated as in panel (**B**), except data were grouped by the cortical layer of the presynaptic neuron. Dotted lines in the middle and right panels represent total connection probabilities measured across all tested connections to L2/3 pyramidal cells (PCs). To facilitate comparison between divergence and total connection probabilities, divergence was only calculated for pairs in which cell *j* was a L2/3 PC. Shading represents 95% confidence intervals. (**D**) Scatter plots of outgoing connection strengths for all pairs of L2/3 PCs sharing a presynaptic partner for each Cre line.

The online version of this article includes the following source data and figure supplement(s) for figure 10:

**Source data 1.** Raw values for plots in *Figure 10*.

**Figure supplement 1.** Comparison of the number of observed divergence motifs to a random network.

translaminar synaptic connections to L2/3. By extension, these subsets of cells in L4 and L5 likely target a greater number of neurons within L2/3 than predicted by the overall measured connection probabilities.

Finally, we examined the relationships between the strengths of divergent connections made by individual presynaptic cells onto multiple L2/3 PCs (*Figure 10D*). We did not observe significant correlations when examining PSP amplitudes from single Rorb, Tlx3, or Pvalb neurons onto multiple L2/3 PCs (Rorb: $R = -0.25$, p=0.17, n = 32; Tlx3: $R = -0.46$, p=0.30, n = 7; Pvalb: $R = 0.07$, p=0.77, n = 19, Pearson's $R$ in log-space). However, we found a moderate positive correlation in the amplitudes of synaptic responses measured from diverging Sst interneurons (*Figure 10D*; Sst: $R = 0.52$, p=0.016, n = 21). In total, the data demonstrate that the effective strength of connections generated by a single presynaptic neuron onto its targets can vary substantially, but also provide evidence that some Sst neurons more strongly inhibit their local network than others.

## Discussion
### Differences in connectivity, strength, and dynamics across and within subclasses

The combination of multicellular patch-clamp recordings and multiphoton stimulation was used to probe more than 10,000 neuron pairs and characterize hundreds of synaptic connections in adult mouse V1 with genetic and spatial specificity. The data reveal aspects of synaptic connectivity, strength, and dynamics that align with specific presynaptic and/or postsynaptic cell subclasses and may suggest unique roles for these connections in cortical computations. For example, intralaminar excitatory connections from Penk-labeled neurons in L2/3 displayed a strong bias for FSIs over PCs in terms of connectivity (*Figure 3B*) and unitary connection strength (*Figure 7B*). In contrast, translaminar excitatory connections from Rorb- and Tlx3-neurons in layers 4 and 5 displayed similar connection probabilities for L2/3 PCs and L2/3 FSIs (*Figure 5D*). The intrinsic electrophysiological properties and rapid kinetics of their outgoing inhibitory connections make FSIs/Pvalb interneurons well suited to mediate feedforward inhibition (FFI; *Muñoz et al., 2017*). Our measurements of excitatory inputs to L2/3 FSIs and L2/3 PCs suggest that FFI may be engaged following excitatory neuron activity within layers 2/3, 4, or 5, but will be most robustly recruited by activity within L2/3 due to the abundance and strength of intralaminar excitatory inputs to L2/3 FSIs. Strong intralaminar FFI could promote sparse coding by L2/3 PCs and contribute to feature competition among L2/3 PCs, wherein the activity of one L2/3 PC reduces the activity of similarly tuned neighbors (*Chettih and Harvey, 2019*).

Differences in the strength, kinetics, and dynamics of inhibitory signaling likely impart diverse computational functions to inhibitory interneuron subclasses (*Abbott and Regehr, 2004*; *Silver, 2010*). Inhibitory responses from Pvalb interneurons to L2/3 PCs were on average stronger, faster, and displayed greater paired-pulse depression than responses from Sst interneurons to L2/3 PCs (*Figure 7A and C*). These differences are generally consistent with previous reports (*Beierlein et al., 2003*; *Pfeffer et al., 2013*), but our study is unique in the extent to which both intralaminar and translaminar inhibition were characterized at the single-cell level. As differences between Pvalb and

Sst connection properties were found for both intralaminar and translaminar connections (*Figure 7E and C*), our data suggest that horizontal and vertical and inhibition from an interneuron subclass may serve similar computational functions.

In addition to the significant differences in connection probability and synaptic properties described above, the data reveal broad diversity within subclasses and overlapping distributions across subclasses (Figures 6 and 7). Furthermore, we recorded widely varying amplitudes of synaptic responses converging to individual postsynaptic neurons (*Figure 9—figure supplement 2*) and diverging from individual presynaptic neurons (*Figure 10C*). Therefore, the properties of synaptic connections are influenced, but are far from fully determined, by presynaptic and postsynaptic cell identity.

## Spatial patterns of connectivity and visual coding

There is a retinotopic map of visual space that is conserved across layers in V1 – arguing for strong vertical synaptic connections (*Dräger, 1975*; *Mangini and Pearlman, 1980*; *Wagor et al., 1980*; *Schuett et al., 2002*; *Bonin et al., 2011*). Rodent V1 lacks the smoothly varying orientation columns observed in higher mammals (*Grinvald et al., 1986*; *Hübener et al., 1997*; *Ts'o et al., 1990*); instead, neurons with disparate orientation tuning are intermingled in a 'salt and pepper' fashion (*Ohki et al., 2005*). However, it is reported that nearby cells display more similar tuning than those farther away from each other (*Ringach et al., 2016*) and that local biases in orientation tuning reveal a global map in orientation tuning that spans V1 and neighboring visual areas (*Fahey et al., 2019*). Connections with little horizontal offset may preserve or stabilize functional maps, while connections between more horizontally distributed neurons may promote their reorganization. In our study, the highest rate of excitatory neuron connectivity to L2/3 PCs was observed from L4 Rorb-labeled neurons with 0–100 μm of horizontal offset (*Figure 5D*) – a likely means of preserving the retinotopic map established by inputs from dLGN. However, the falloff in connection rate with horizontal distance was gradual for all classes of translaminar excitatory inputs to L2/3 PCs measured in this study (*Figure 5D*, *Figure 6—figure supplement 2*). Additionally, there was little to no correlation between intersomatic horizontal distances and EPSP amplitudes (*Figure 7D*, *Table 3*). These findings, and the convergence and divergence of connections across hundreds of microns of horizontal space (Figures 8 and 9), illustrate putative circuit mechanisms by which translaminar excitatory connections can promote both the maintenance and redistribution of sensory information across layers.

We found that excitatory input from L5 IT neurons (labeled by either Rorb or Tlx3-Cre) to L2/3 PCs rivals excitation from L4 IT neurons (labeled by either Rorb or Scnn1a-Cre) in terms of connection probability (*Figures 5B and 4D*), unitary connection strength (*Figure 7*), as well as convergence and divergence (Figures 9 and 10). The connection from L5 IT neurons to L2/3 PCs is interesting in the context of in vivo studies demonstrating a strong influence of brain state and active behavior on the activity of neurons in primary visual cortex (*Niell and Stryker, 2010*; *Keller et al., 2012*; *Neske et al., 2019*; *Stringer et al., 2019*). If L5 IT neurons receive long-range input encoding behavioral variables from higher cortical areas, local projections from L5 IT neurons could provide a mechanism to distribute this information within V1 so that it can be integrated with synaptic connections providing sensory information (e.g., thalamic input or connections ascending from L4).

Why were excitatory connections from L5 to L2/3 previously described as sparse or weak (*Thomson and Bannister, 1998*; *Reyes and Sakmann, 1999*; *Thomson et al., 2002*; *Lefort et al., 2009*; *Jiang et al., 2015*)? The age range of animals used in this study is older than most, and connectivity in younger animals may be more specifically driven by sensory input. It is also possible that the use of Tlx3 and Rorb-Cre allowed us to more thoroughly identify and measure these somewhat rare connections that may be specific to IT-type L5 neurons rather than the primarily subcortical-projecting extratelencephalic (ET)-type L5 neurons. This difference may be especially valuable in visual cortex where L5 excitatory projection types are not as distinctly separated into sublayers compared to somatosensory cortex (*Groh et al., 2010*; *Kim et al., 2015*). More generally, imparting photosensitivity to genetically defined subclasses of neurons can provide additional insight into the source of photostimulus responses not achieved with one-photon or two-photon glutamate uncaging. However, connections arising from unlabeled populations of cells cannot be tested. Glutamate uncaging may allow less biased measures of connectivity due to the ubiquitous expression of glutamate receptors, but the higher density of photosensitive neurons may result in greater off-target activation and subsequent

false positives, even with spatially refined two-photon stimulation (*Figure 2D–I*). One such study estimated that roughly five neurons were activated at each photostimulus location (*Matsuzaki et al., 2008*), though effective resolution will depend on many experimental factors.

## Limitations of photostimulation

When assaying connectivity by photostimulation, we do not have record of presynaptic APs and instead rely on separate experiments to characterize the reliability and precision of two-photon-evoked spiking (*Figure 1*). We found our photostimulation protocol to be effective across several Cre lines. However, we observed differences in photosensitivity between Cre lines and expression methods (*Figure 1B*). We endeavored to account for these differences by calibrating photostimulation power according to the Cre line used; however, due to variability between cells within a Cre line, we estimate that up to 15% of photostimulated neurons did not generate APs (*Figure 1B*). Additionally, while the timing of two-photon-evoked spiking in response to repeated photostimulation was highly consistent for most ChrimsonR-expressing cells (jitter < 1 ms; *Figure 1C*), even low variability in AP timing may preclude detection of weak and/or unreliable connections. These sources of false negatives are strongly mitigated in paired recordings where variability in timing, or failure to evoke presynaptic APs by current injection, can be corrected using AP detection and alignment of postsynaptic responses.

The spatial resolution of our photostimulation protocol approached, but did not reach, the dimensions of an individual neuron (*Figure 1D*). Therefore, photostimulation can generate spiking in off-target presynaptic cells that also express the opsin, leading to false positives when measuring connectivity. Experiments in which we subsequently patched optically identified presynaptic cells (*Figure 4*, *Figure 4—figure supplement 1*) demonstrated that this occurs, but suggested that it was rare under our experimental conditions (11 verified excitatory connections in 12 tested, 22 verified inhibitory connections in 22 tested). Given the potential for both false positives and false negatives to arise from photostimulation, it was critical to compare optical measures of connectivity to measures made using paired recordings. We did so for 14 different categories of synaptic connections determined by presynaptic and postsynaptic cell identities and intersomatic distance (*Figure 3C*, *Figure 3—figure supplement 2*, *Table 1*). Connection probabilities estimated by each method were not statistically different. Furthermore, connection probabilities measured using two-photon stimulation displayed a strong and significant correlation with corresponding measures made by multicellular recordings (*Figure 3D*). Altogether, the results argue that while photostimulation carries sources of error not present in paired recordings, measurements of synaptic connectivity made at the population level accurately represent the underlying connectivity.

## Limitations of slice physiology

A source of false negatives inherent to acute slice preparations is the severing of neuronal processes. This subject has been explored in other studies, and in this context, connection probabilities measured in vitro may be thought of as the lower bounds for connection probabilities present in vivo (*Stepanyants et al., 2009*; *Levy and Reyes, 2012*). In this study, slices were cut at a slightly oblique angle to preserve neuronal processes, and we targeted patching and photostimulation to cells that were >40 µm below the surface of the slice (both cells were >40 µm deep for 81.9% of connections probed). We observed connections from L5 IT-type neurons (labeled by Rorb or Tlx3-Cre) to distant L2/3 neurons with 300–600 µm of horizontal offset – corresponding to total intersomatic distances of 485–744 µm (11 connections/506 probed, 2.2%). Considering this, we believe the near absence of connections observed over similar intersomatic distances (e.g., from Sst neurons with >200 µm of horizontal offset or from Ntsr1 neurons) indicates selectivity of the connections and is not strictly a result of acute slice preparation. However, we acknowledge that intersomatic distance is not equivalent to the distance an axon travels to form a synaptic contact (e.g., synapses formed on the distal dendrites of a nearby postsynaptic cell), and some presynaptic projections may be more prone to cutting than others due to tortuous axon trajectories.

The effect of a synaptic connection on the postsynaptic membrane potential is influenced by multiple complex biological mechanisms including presynaptic transmitter release, postsynaptic receptor activation, and the intrinsic electrophysiological properties of the cell. Features measured in the acute slice preparation may differ in vivo due to differences in ionic gradients and the dynamic

influence of neuromodulators. Furthermore, the effect of a distal synaptic connection on its local postsynaptic compartment may be drastically different than the response measured by a somatic recording, and such connections may go undetected. To this end, recording of postsynaptic responses in either voltage- or current-clamp each comes with inherent advantages and disadvantages (*Williams and Mitchell, 2008*; *Barth et al., 2016*; *Jiang et al., 2016*). Although voltage-clamp recording is generally thought to increase the ability to detect small responses (especially when combined with pharmacology to partially isolate synaptic currents), it presents challenges in the interpretation of results due to space-clamp errors, and strong somatic bias has been shown to persist under nominally ideal recording conditions (*Spruston et al., 1993*; *Williams and Mitchell, 2008*; *Beaulieu-Laroche and Harnett, 2018*). Given these limitations and the insight into cell subclass provided by the intrinsic features of patched cells (*Figure 3—figure supplement 1*), we opted to conduct this survey of synaptic connectivity and functional properties using current-clamp recordings, as have many previous studies (*Sayer et al., 1990*; *Markram et al., 1997*; *Feldmeyer et al., 2002*; *Song et al., 2005*; *Lefort et al., 2009*; *Cossell et al., 2015*; *Jiang et al., 2015*).

Due to the diversity of postsynaptic morphologies and the target preferences of interneuron subclasses for specific subcellular compartments (*Tremblay et al., 2016*), the propensity to miss distal synaptic contacts is unlikely to be uniform across cell classes. In the context of results reported in this study, differences in the amplitude and kinetics of Pvalb and Sst-mediated connections are consistent with Sst neurons targeting more distal dendritic locations (*Figure 7A and C*). This may predict a greater likelihood of missed connections from Sst interneurons compared to Pvalb interneurons. Large-scale anatomical studies of cortical microcircuitry are beginning to provide unprecedented insight on this topic (*Turner et al., 2020*), and it will be of great value to examine the correspondence between anatomical and physiological measures of connectivity and function across cortical cell types.

This study illustrates that combining multicellular recording with multiphoton optogenetic stimulation can efficiently characterize synaptic connectivity from genetically defined populations of neurons. Future studies can readily employ the Cre lines validated in this study with electrophysiological recording in other layers or cortical areas. Validation of additional Cre lines will allow more subclasses of connections to be probed. Furthermore, there is the potential to test for differences in synaptic connectivity between transcriptomically defined cell types within Cre line-defined subclasses by following two-photon circuit characterization with post hoc measurement of gene expression using multiplex fluorescence in situ hybridization (*Nicovich et al., 2019*). Future work utilizing a variety of complementary electrophysiological and optical methods to characterize local and long-range synaptic signaling between categories of neurons promises to further our understanding of cell types and their unique roles in circuit function and behavior.

# Materials and methods

## Key resources table

| Reagent type (species) or resource | Designation | Source or reference | Identifiers | Additional information |
|---|---|---|---|---|
| Genetic reagent (*Mus musculus*) | B6.FVB(Cg)-Tg(Ntsr1-cre)GN220Gsat/Mmucd | MMRRC | RRID:MMRRC_030648-UCD | |
| Genetic reagent (*M. musculus*) | B6;129S-*Penk*$^{tm2(cre)Hze}$/J | The Jackson Laboratory | RRID:IMSR_JAX:025112 | |
| Genetic reagent (*M. musculus*) | B6;129P2-*Pvalb*$^{tm1(cre)Arbr}$/J | The Jackson Laboratory | RRID:IMSR_JAX:008069 | |
| Genetic reagent (*M. musculus*) | B6;129S-*Rorb*$^{tm1.1(cre)Hze}$/J | The Jackson Laboratory | RRID:IMSR_JAX:023526 | |
| Genetic reagent (*M. musculus*) | B6;C3-Tg(Scnn1a-cre)3Aibs/J | The Jackson Laboratory | RRID:IMSR_JAX:009613 | |
| Genetic reagent (*M. musculus*) | B6J.Cg-*Sst*$^{tm2.1(cre)Zjh}$/MwarJ | The Jackson Laboratory | RRID:IMSR_JAX:028864 | |
| Genetic reagent (*M. musculus*) | B6.FVB(Cg)-Tg(Tlx3-cre)PL56Gsat/Mmucd | MMRRC | RRID:MMRRC_041158-UCD | |

*Continued on next page*

*Continued*

| Reagent type (species) or resource | Designation | Source or reference | Identifiers | Additional information |
|---|---|---|---|---|
| Genetic reagent (*M. musculus*) | Ai167(TIT2L-ChrimsonR-tdT-ICL-tTA2) | Available from Allen Institute | | |
| Recombinant DNA reagent | flex-ChrimsonR-EYFP-Kv | Addgene | RRID:Addgene_135319 | |
| Software, algorithm | ACQ4 | http://www.acq4.org/, *Campagnola et al., 2014* | RRID:SCR_016444, | |
| Software, algorithm | Igor Pro | WaveMetrics | RRID:SCR_000325 | |
| Software, algorithm | MIES: Software for Electrophysiology functions | Allen Institute | RRID:SCR_016443 | |

## Animals and stereotaxic injections

Mice were housed and sacrificed according to the Institutional Animal Care and Use Committee-approved protocols. Cre driver lines used in this study include Ntsr1-Cre_GN220, Penk-IRES2-Cre-neo, Pvalb-IRES-Cre, Rorb-IRES2-Cre, Scnn1a-Tg3-Cre, Sst-IRES-Cre, and Tlx3-Cre_PL56. Driver names were simplified as Ntsr1, Rorb, Penk, Pvalb, Scnn1a, Sst, and Tlx3 throughout the article. For each driver line, *Cre* and *tdTomato* reporter expression have been characterized by in situ hybridization as part of the Allen Institute Transgenic Characterization dataset (https://connectivity.brain-map.org/transgenic). The transcriptomic identities of labeled cells in each driver line have been previously described (*Tasic et al., 2018*).

Transcriptomic characterization of the Penk-Cre line shows labeling of L2/3 IT neurons, L6 IT neurons, and multiple types of VIP interneurons (*Tasic et al., 2018*). Photostimulation of interneurons in this line will result in an underestimate of excitatory connectivity, though the magnitude of this effect will depend on the proportion of interneurons within the labeled population. Such proportions are difficult to infer from dissociated tissue; therefore, we examined double fluorescence in situ hybridization (dFISH) images staining for tdTomato and *Gad1* mRNA in a Penk:Ai14 mouse (available within the Allen Institute Transgenic Characterization dataset). *Gad1* fluorescence was measured using ImageJ in regions of interest (ROIs) defined by automatically detected particles. Specifically, binary images were generated using a threshold automatically determined by the 'IsoData' method. Particles were then detected using the 'Analyze Particles' interface with a minimum size corresponding to 50 μm² and a minimum circularity of 0.3. This process to define ROIs was independently applied to the tdTomato- and *Gad1*-associated channels of each image. Fluorescence intensity in the *Gad1* channel was measured from the original images within both sets of ROIs. tdTomato-defined ROIs were used to measure *Gad1* fluorescence within tdTomato-labeled cells. Fluorescence intensities measured in *Gad1*-defined ROIs were used to set the minimum fluorescence value required to classify tdTomato-labeled cells as co-labeled (*Figure 3—figure supplement 2*).

In images from a Penk:Ai14 mouse, we observed colocalization of *Gad1* fluorescence signal in approximately 12% of tdTomato-labeled cells within L2/3 of visual cortex (26/211 cells; *Figure 3—figure supplement 2*), suggesting a small minority of cortical Penk-Cre-labeled neurons are GABAergic. In comparison, *Gad1* labeling was observed in the majority of tdTomato-labeled cells in two interneuron Cre lines (VIP:Ai14: 191/225 cells, 85%; Pvalb:Ai14: 208/243 cells, 86%), demonstrating the relatively high sensitivity of the co-labeling criteria. Transcriptomic characterization of the other excitatory Cre lines used in this study does not suggest significant labeling of interneurons (*Tasic et al., 2018*). Consistent with this, only 2.9% of Rorb:Ai14-labeled neurons (13/455 cells) displayed significant *Gad1* signal, likely resulting from close apposition of excitatory and inhibitory cells (see example image in *Figure 3—figure supplement 2*). In total, the analyses suggest that less than 20% of targeted Penk-Cre-labeled neurons were interneurons. Reducing the number of connections probed by this factor has little effect on the relative differences in excitatory connectivity measured between postsynaptic subclasses or across intersomatic distances (*Figure 3*). Additionally, photostimulation data remain closely aligned with data from paired recordings (*Table 1*) following a reduction in the number of excitatory connections tested via photostimulation. Inhibitory synaptic responses were not observed in mapping experiments using the Penk-Cre line. This is consistent with previous reports that VIP

interneurons largely avoid synapsing with L2/3 PCs (*Lee et al., 2013*; *Pfeffer et al., 2013*; *Pi et al., 2013*), and our estimate that <20% of Penk-labeled neurons were interneurons (resulting in a relatively small number of inhibitory connections tested).

Transgenic expression of ChrimsonR was achieved by crossing the indicated Cre line to the effector line Ai167(TIT2L-ChrimsonR-tdT-ICL-tTA2) (*Daigle et al., 2018*). Expression of soma-targeted ChrimsonR was achieved by stereotaxic injection of AAV serotype one carrying ChrimsonR-EYFP fused to a $K_v 2.1$ somatic localization motif ('flex-ChrimsonR-EYFP-kv,' Addgene plasmid #135319, titer ~4 × $10^{12}$ vg/ml). Injections were made into right visual cortex of mice aged between P26 and P53. Injection coordinates were 3.8 mm posterior from bregma and 3.0 mm lateral from midline. Two injections of 500 nL each were targeted 300 and 600 µm below the surface of the pia with the goal of transfecting cells in all cortical layers. Experiments were performed 21–28 days after injection.

## Slice preparation

Brain slices were prepared from young adult mice (40–126 days old, mean ± SD = 58.0 ± 11.2) of either sex. Mice were anesthetized with isoflurane then transcardially perfused with oxygenated, cold NMDG-slicing solution containing (in mM) 98 HCl, 96 N-methyl-D-glucamine (NMDG), 2.5 KCl, 25 D-glucose, 25 $NaHCO_3$, 17.5 4-(2-hydroxyethyl)–1-piperazineethanesulfonic acid (HEPES), 12 N-acetylcysteine, 10 $MgSO_4$, 5 Na-L-ascorbate, 3 myo-inositol, 3 Na pyruvate, 2 thiourea, 1.25 $NaH_2PO_4$ $H_2O$, 0.5 $CaCl_2$, and 0.01 taurine. Acute parasagittal slices (350 µm thick) were cut with either a Compresstome (Precisionary Instruments) or a Leica VT1200S at an angle of 17° relative to the sagittal plane to preserve PC apical dendrites (*Seeman et al., 2018*). Slices were stored at 34°C in NMDG-slicing solution for 10 min, after which slices were maintained at room temperature in holding solution containing (in mM) 94 NaCl, 25 D-glucose, 25 $NaHCO_3$, 14 HEPES, 12.3 N-acetylcysteine, 5 Na-L-ascorbate, 3 myo-inositol, 3 Na pyruvate, 2.5 KCl, 2 $CaCl_2$, 2 $MgSO_4$, 2 thiourea, 1.25 $NaH_2PO_4$ $H_2O$, and 0.01 taurine.

## Electrophysiological recordings

At least 1 hr after slicing, slices were transferred to recording chambers with constant perfusion of aCSF containing (in mM) 126 NaCl, 18 $NaHCO_3$, 12.5 D-glucose, 3 KCl, 2 $CaCl_2$, 2 $MgSO_4$, 1.25 $NaH_2PO_4$, and 0.16 Na L-ascorbate. pH was measured between 7.2 and 7.3, and osmolarity was between 290 and 300 mOsm. We opted to use a high divalent aCSF in this study for two reasons. 2 mM $Ca^{2+}$ was used with the intent of increasing the probability of presynaptic release, thereby decreasing the likelihood that a connection was missed as a result of a low probability of release. 2 mM $Mg^{2+}$ was used to limit the activation of NMDA receptors that could promote sustained depolarization of cells and increase the likelihood of polysynaptic responses. aCSF was bubbled with carbogen and maintained at 32–34°C with an inline heater. Slices were visualized under infrared illumination using either a ×10/0.2 NA or ×40/1.0 NA objective (Zeiss) and a digital camera (Hamamatsu; ORCA Flash4.0). Early experiments used HCImageLive (Hamamatsu) to visualize the slice; however, the majority of experiments used ACQ4 software (acq4.org; *Campagnola et al., 2014*) to acquire images, control and monitor pipette locations, and place and record photostimulus locations in a common software interface. Patch-clamp recordings were made from 1 to 4 headstages that were amplified and low-pass filtered at 10 kHz using Multiclamp 700B amplifiers (Molecular Devices). Signals were digitized at 25–100 kHz using ITC-18 digitizers (Heka). Data was acquired using Multi-channel Igor Electrophysiology Suite (MIES; https://github.com/AllenInstitute/MIES, *Braun et al., 2022*), within Igor Pro (WaveMetrics).

Patch electrodes were pulled from thick-walled filamented borosilicate glass (Sutter Instruments) with a DMZ Zeitz-Puller. Tip resistance was 3–6 MΩ. Pipette recording solution contained (in mM) 130 K-gluconate, 10 HEPES, 0.3 ethylene glycol-bis(β-aminoethyl ether)-N,N,N',N'-tetraacetic acid (EGTA), 3 KCl, 0.23 $Na_2$GTP, 6.35 $Na_2$ phosphocreatine, 3.4 Mg-ATP, 13.4 biocytin. 25 µM of either Alexa 594 or Alexa 488 was added for experiments using ChrimsonR labeled with enhanced yellow fluorescent protein (EYFP) or tdTomato, respectively. Osmolarity was between 280 and 290 mOsm. pH was adjusted to 7.2–7.3 using KOH. Reported voltages are uncorrected for a liquid junction potential of 9 mV between aCSF and the K-gluconate-based internal.

For mapping experiments, 1–4 cells in L2/3 were targeted for patching (23–140 µm below the surface of the slice, mean = 70 µm, median = 66 µm). For the majority of experiments, we measured synaptic responses evoked by full-field one-photon stimulation before conducting two-photon

**Table 4.** Intrinsic physiological properties of recorded neurons.
Measured intrinsic values for L2/3 PCs and L2/3 interneurons. Data are presented as mean ± SD.

| Cell class | L2/3 PC | L2/3 FSI | L2/3 VIP | L2/3 Sst |
|---|---|---|---|---|
| $V_{rest}$ (mV) | –72.8 ± 6.7 | –68.2 ± 7.3 | –62.6 ± 10.5 | –61.8 ± 4.6 |
| Input resistance (MΩ) | 90.3 ± 31.5 | 92.4 ± 33.8 | 174 ± 64.3 | 199 ± 57.0 |
| Tau (ms) | 14.4 ± 4.3 | 6.5 ± 1.8 | 9.9 ± 4.3 | 16.8 ± 4.3 |
| Capacitance (pF) | 171 ± 58 | 80 ± 44 | 56 ± 15 | 87 ± 20 |
| Sag | 0.032 ± 0.027 | 0.062 ± 0.042 | 0.117 ± 0.068 | 0.16 ± 0.096 |
| Rheobase (pA) | 171 ± 75 | 268 ± 119 | 126 ± 77 | 129 ± 52 |
| FWHM (ms) | 1.37 ± 0.31 | 0.62 ± 0.17 | 1.25 ± 0.29 | 0.79 ± 0.14 |
| Upstroke-downstroke ratio | 4.30 ± 0.68 | 1.72 ± 0.23 | 3.36 ± 0.58 | 2.10 ± 0.36 |
| AP peak (mV) | 40.6 ± 8.0 | 19.5 ± 7.9 | 27.5 ± 12.7 | 24.0 ± 13.4 |
| AP trough (mV) | –48.4 ± 4.9 | –53.0 ± 5.16 | –51.0 ± 2.8 | –52.7 ± 5.2 |
| AP height (mV) | 89.0 ± 10.1 | 72.5 ± 10.2 | 78.5 ± 13.1 | 76.7 ± 17.0 |
| Average firing rate (spikes/s) | 7.78 ± 2.36 | 30.9 ± 10.8 | 15.5 ± 11.3 | 15.6 ± 6.6 |
| Adaptation | 0.090 ± 0.079 | 0.008 ± 0.021 | 0.044 ± 0.083 | 0.081 ± 0.090 |
| f-i slope (spikes/s/pA) | 0.12 ± 0.06 | 0.66 ± 0.19 | 0.20 ± 0.11 | 0.29 ± 0.11 |

PC: pyramidal cell; AP: action potential; FWHM: full-width at half-maximum.

stimulation of individual cells (*Figure 5—figure supplement 1*). When measuring either one-photon or two-photon-evoked synaptic responses, automated bias current injection was used to maintain the membrane potential near –70 mV or –55 mV for mapping of excitatory or inhibitory inputs, respectively. Recordings were terminated if necessary holding current exceeded ±300 pA or if access resistance increased above 35 MΩ.

PC identity was confirmed by examination of cell morphology in a two-photon z-stack acquired at the end of each experiment (typically 15–90 min after obtaining whole-cell access). Z-stacks were examined for dendritic spines and a prominent apical dendrite. To further assist in our classification of the postsynaptic cells, intrinsic properties were characterized by a family of 1 s square current injections from –130 pA to 250 to pA, in 20 pA steps (*Table 4*, *Figure 3—figure supplement 1*). A subset of cells required current injections > 250 pA to evoke APs (additional current injections applied with +20 pA intervals). Responses to current injection were measured from a baseline membrane potential of –70 mV (adjusted by automated bias current injection). Intrinsic parameters were measured as described by *Gouwens et al., 2019* with the exception that the smallest hyperpolarizing current injection (–10 pA) was not used in our estimation of the membrane time constant. Exponential fits to the weakest hyperpolarizing current injection were often inaccurate due to background synaptic activity present in mapping experiments. AP properties (FWHM, upstroke-downstroke ratio, peak, trough, and height) were measured from the first AP evoked at rheobase. The average firing rate and adaptation index were measured from current injections 40 pA above rheobase. A small number of cells initially identified as PCs displayed electrophysiological characteristics typical of interneurons (high input resistance, narrow AP width, and/or small upstroke:downstroke ratio) and were reclassified as interneurons. Interneurons were further classified as FSIs (putative Pvalb), putative VIP, or putative Sst interneurons based on unsupervised clustering of intrinsic features (*Figure 3—figure supplement 1*).

## Photostimulation

Experiments were performed using Prairie Ultima two-photon laser scanning microscopes (Bruker Corp). A tunable pulsed Ti:sapphire laser (Chameleon Ultra, Coherent) was used for imaging, and a fixed wavelength (1070 nm) pulsed laser (Fidelity 2, Coherent) was used for photostimulation. Locations of putative presynaptic neurons were manually identified from two-photon reference images

acquired in PrairieView software. The experimenter placed a target at the center of the soma, around which a spiral path of 5 revolutions and 10 μm outer diameter was generated. Stimulus duration was 10 ms, and each cell was stimulated at least 20 times at 85 mW for all experiments except those using Pvalb-Cre and Sst-Cre neurons transfected with AAV. ChrimsonR-expressing neurons identified using these Cre lines with AAV-mediated expression were stimulated at 35 mW due to their higher photosensitivity (*Figure 1B*). Photostimulation and electrophysiology recordings were synchronized via a TTL trigger generated by MIES acquisition software. The voltage output from the Prairie system to the photoactivation Pockels cell was recorded in MIES for precise alignment of electrophysiological recordings to the photostimulus. In initial mapping experiments, photostimulation targets were placed by the experimenter within the PrairieView user interface. However, for the majority of data collection, two-photon reference images were first collected in PrairieView and then imported into ACQ4. An experimenter placed targets within ACQ4 software, and photostimulation was controlled via the PrairieLink API. Photostimulation parameters were the same in experiments using PrairieView or ACQ4 to place photostimulus locations. In both cases, the trajectory of the photostimulation galvos was determined by the location of photostimulus targets within the two-photon reference image. The ability to visualize photostimulus locations and two-photon reference images within the same coordinate system in which transmitted light images from the camera were streamed greatly facilitated experiments in which optically identified connections were subsequently patched (*Figure 4*).

For experiments characterizing two-photon-evoked generation of APs (*Figure 1*), loose-seal recordings (20–80 MΩ seal resistance) were made from ChrimsonR-expressing neurons 40–135 μm below the surface of the slice. ChrimsonR-expressing cells were identified by the presence of fluorescence, visualized by brief reflected light illumination via an LED. Prior to two-photon stimulation, we confirmed our ability to detect light-evoked APs by applying a 1 ms one-photon stimulation to the cell via the ×40 objective (1.3–1.5 mW, 590 nm). The minimum power required to drive reliable two-photon-evoked spiking was determined as the stimulus intensity that generated APs on 10 out of 10 trials. Effective optogenetic excitation is predicted to decrease with depth in tissue due to increased scattering of photons (*Yona et al., 2016*). Consistent with this, for some of the Cre line and expression methods examined, the minimum power required to drive reliable firing displayed a weak to moderate positive correlation with cell depth (*Figure 1—figure supplement 1*). However, only one group reached clear statistical significance (Sst:Ai167, Pearson's $R$ = 0.66, p=0.004). We did not attempt to set stimulus power according to cell depth in mapping experiments due to the moderate and variable nature of this relationship. Additional factors including strength of ChrimsonR expression and the intrinsic excitability of cells are likely to contribute to the observed variability.

To characterize the lateral resolution of our photostimulus, we delivered stimuli in a radial grid pattern containing seven spokes, each with three stimuli 10, 20, and 30 μm away from the center of the recorded cell. To measure the axial resolution of our photostimulus, we offset the focus of the objective above and below the recorded cell in 10 μm increments until we observed spiking on 0 out of 10 trials. Spatial resolution was characterized using the same photostimulation powers used for mapping. FWHM was calculated for each cell using from Gaussian fits to spike probability versus the distance of the offset (either lateral or axial).

## Measurements of cell densities and estimation of off-target activation

Z-stacks were collected at the conclusion of each mapping experiment (300 μm × 300 μm images with ~0.6 μm × 0.6 μm pixels, collected in 1 μm z-steps; 89 ± 31 slices per stack, mean ± SD). 217 z-stacks were used to manually identify the number and location of labeled cells. The densities of labeled cells were calculated as the number of cells divided by the imaged volume. The densities of cells in the same Cre lines crossed to the commonly utilized Ai14 mouse line (tdTomato reporter) were measured using images from the Allen Institute Transgenic Characterization webpage (http://connectivity.brain-map.org/transgenic). For each Cre line, three images from experiments measuring tdTomato expression in adult mouse (P55–P97) were selected for analysis and the average densities are represented as inverted triangles in *Figure 2B and C*. For Scnn1a-Tg3-Cre, data from adult Ai14 mice were not available; therefore, images collected from an adult Ai9 mouse were used instead. Links to images utilized and corresponding measurements are provided in *Figure 2—source data 1*. To calculate density using these reference images, the volume was calculated as the product of the surface area of the region containing counted cells and the reported thickness of each slice (25–50 μm).

Our strategy to estimate the extent of off-target activation associated with two-photon stimulation was to associate measurements of relative cell locations denoted when measuring the density of cells to measurements of the spatial resolution of photostimulation (*Figure 1*). We began by calculating the cell-to-cell distances (lateral and axial) from each cell to all other cells in the same imaged volume. We then measured the number of neighbors per cell with combined lateral and axial offsets (*Figure 2F and G*). To avoid discounting photosensitive cells that were not imaged, we normalized the number of neighbors in each bin to the number of seed cells that were a sufficient distance from the edges of the imaged volume to include the corresponding offsets. This edge correction may result in an over-estimate of off-target activation and subsequent false positives in mapping experiments as unimaged cells near the upper boundary of the slice may be prone to axonal severing and therefore unlikely to generate off-target synaptic responses. Unimaged cells below the collected z-stacks may be less likely to be activated due to the scattering of light with increasing depth and resulting need for greater photostimulus intensities (*Figure 1—figure supplement 1*). To avoid underestimating the errors associated with off-target photostimulation, we did not attempt to account for these factors.

For each cell-to-cell distance, we estimated the probability that one cell would be co-activated when the other was targeted. Given the variability observed between Cre lines and expression methods (*Figure 1*), each experimental condition was considered separately using corresponding characterization data. To estimate off-target activation probabilities with combined lateral and axial offsets (which were not directly measured), we first estimated the probability of off-target activation given either the lateral or the axial offset between cells. We subsequently assigned the lower of the two probabilities predicted by the independent offsets. For example, in the Tlx3:AAV dataset, a neighbor with 10 µm lateral offset and 10 µm axial offset would yield independent off-target activation probabilities of 0.18 and 0.83, respectively (estimated from Gaussian fits in *Figure 1—figure supplement 2E*, *Figure 1—figure supplement 3D*). In this case, off-target activation with the combined offsets would be estimated as 0.18. This may be a conservative estimate of our photostimulus resolution as the falloff in off-target activation probability may be more dramatic with combined lateral and axial offsets than with one alone. To illustrate the effective point spread functions predicted by this method, the photostimulus resolutions in *Figure 2—figure supplement 1* were drawn with 1 µm resolution. We also explored making this estimation based on a series of ellipsoids corresponding to off-target activation probabilities ranging from 0.01 to 0.99, with the radius of each axis determined by our characterization data. This strategy produced slightly lower estimates of off-target activation, but similar results in terms of relative estimates generated across Cre lines and expression methods.

## Analysis of connectivity

The individual and averaged responses to photostimulation were quantitatively and visually assessed for evidence of a synaptic response. The features in *Table 5* were measured in a 50 ms post-stimulus window beginning at the start of photostimulation and a 50 ms pre-stimulus window 70–20 ms before photostimulation – when no evoked responses are expected (*Figure 7—figure supplement 1*). The standard deviation of both the average response and the deconvolved response was measured in a baseline window 20 ms before the onset of the photostimulus. The features were used to train a SVM on human annotation of the presence or absence of a synaptic response (using the scikit-learn package

**Table 5.** Features measured for all photostimulus responses.

| Features from average response | Features from deconvolved average response | Features from individual deconvolved responses |
| --- | --- | --- |
| Maximum amplitude of response (peak) | Peak of deconvolved average (deconvolved peak) | Average number of times each trace exceeded 3× deconvolved SD |
| Time of peak | Time of deconvolved peak | SD of number of crossings per sweep |
| Standard deviation (SD) at time of peak | SD of deconvolved trace in baseline window (deconvolved SD) | |
| SD in baseline window | Number of times the deconvolved trace exceeded 3× deconvolved SD | |
| | Number of times the deconvolved trace exceeded 5× deconvolved SD | |

in Python; http://scikit-learn.org/stable/). Experiments examining excitatory or inhibitory responses were considered separately. The excitatory classifier was trained on 3328 randomly selected probed connections and tested against 4994 withheld probed connections (a 40–60% train-test split). The excitatory classifier achieved 99% overall accuracy in classifying the test dataset (*Figure 5—figure supplement 2A*). The inhibitory classifier was trained on 1887 examples and tested on 2831 withheld examples and achieved 97% overall accuracy (*Figure 6—figure supplement 1A*). False negatives missed by either classifier were typically reliable but low-amplitude or slowly rising voltage responses, whereas false positives were often caused by large changes in membrane potential only present on a small number of trials. In a few cases, discrepancies between the SVM and initial human annotations were found to be human errors. In these cases, annotations were corrected, and the analyses were repeated. To determine if differences in human versus machine classification impacted our measurements of connectivity, we ran the classifiers on the full datasets and examined the spatial distribution of connection probabilities (*Figure 5—figure supplement 2B and C*, *Figure 6—figure supplement 1B*). The classifications provided by the SVM displayed similar patterns of connectivity with regard to cortical layer of the presynaptic cell, Cre line, postsynaptic subclass, and horizontal offset between cells.

As an alternative measure of connectivity that was completely independent of human annotation, we generated post-stimulus and pre-stimulus z-scores for each photoresponse by dividing the peak amplitude of the average response by the standard deviation of the average response measured in a 20 ms baseline period (*Figure 7—figure supplement 1*). We then set a threshold equal to the 99th percentile of all pre-stimulus z-scores. If post-stimulus z-scores exceeded that threshold, corresponding cells were classified as connected. Discrepancies between human annotation and this single measure were slightly higher compared to the machine classifier. Despite that, the patterns of connectivity examined by presynaptic layer of origin, horizontal offset, or Cre line were similar for all three methods of connection classification (*Figure 5—figure supplement 2D–F*, *Figure 6—figure supplement 1C and D*).

In a subset of mapping experiments measuring excitatory connectivity from Penk neurons to L2/3 PCs (*Figure 3B*) and in all experiments measuring inhibitory connectivity between Pvalb interneurons (*Figure 6D*), the patched cell expressed ChrimsonR. As a result, a subset of photostimulus locations near the patched cell generated subthreshold depolarization that was an artifact of direct activation of ChrimsonR. Direct stimulus artifacts were distinguished from excitatory connections based on the onset of depolarization, which was coincident with the start of photostimulation. Although the amplitudes of these artifacts were fairly modest (median = 0.23 mV), they could in principle mask small, coincident synaptic connections. Therefore, these photostimulus locations were excluded from calculation of the connection probability.

Stage coordinates of the photostimulated neurons were extracted from the position of the motorized stage and the targeted locations of the photostimulation galvos. Photostimulus locations were then offset to the position of the recorded neuron and rotated according to the orientation of the slice to provide vertical (along the pia to white matter axis) and horizontal (anterior to posterior) offsets between the pre- and postsynaptic neurons. The distance of the postsynaptic neuron from the pia was measured from overlaid transmitted light and fluorescence images captured after the neuron was filled with fluorescent dye. This postsynaptic distance from the pia was added to the vertical offset between probed cells to calculate the distance of the presynaptic neuron from the pia. This does not account for curvature of the pial surface, which was minimal for V1 across the horizontal distances considered here. Representative layer boundaries used in this study are L1 0–80 µm, L2/3 80–330 µm, L4 330–480 µm, L5 480–750 µm, and L6 750–1000 µm. These boundaries were informed by the Allen Reference Atlas and the positions of presynaptic neurons assayed using the layer-specific excitatory Cre lines in this study (*Figure 5*). To measure connection rate as a function of the distance of the presynaptic neuron from the pia and horizontal offset, we used 100 µm bins. Adjustment to bin sizes did not influence findings. We calculated 95% Jeffreys Bayesian confidence intervals based on the number of connections found and probed (presented as shading throughout the article) using statsmodels in Python. Fisher's exact tests using 2 × 2 contingency tables and all chi-squared tests were performed using scipystats in Python. Fisher's exact test using larger contingency tables (3 × 2 or 4 × 2) were performed in RStudio. This approach was taken (in place of chi-squared tests) when measuring the distance dependence of divergence (*Figure 10B*) due to the low number of expected counts.

To test for the overrepresentation of convergence and divergence motifs, we simulated random connectivity of each dataset 1000 times using experimentally measured distance-dependent and subclass-dependent connection probabilities. Distance-dependent connection probabilities were based on horizontal distance between cells and the distance of the presynaptic cell from the pia using 100 μm × 100 μm bins. We counted the number of convergence or divergence motifs in each simulation and compared the resulting distribution to the number of motifs observed in the experimental dataset. p-Values are reported as the fraction of simulations in which the number of motifs was equal to or greater than the observed measurement. In cases where the observed number exceeded all 1000 simulations, the p-value is reported as <0.001.

To measure completed convergence (*Figure 9B and C*), we identified all ordered pairs of probed connections that shared a common postsynaptic target (i.e., two photostimulated cells and one patched cell). We then identified all such pairs in which the first presynaptic cell formed a connection with the postsynaptic cell. From this subset of the data, we calculated the fraction of pairs in which the second presynaptic cell also formed a connection; that is, completed the convergence motif. Measurement of completed divergence (*Figure 10B and C*) followed the same process, except that we started by considering all order pairs of probed connections that shared a common presynaptic cell (one photostimulated cell and two patched cells). Unlike the bootstrap method described above, this analysis made no assumptions of independent connectivity, nor did it employ experimentally determined measures of distance dependence. It provided complementary, descriptive measures that allowed us to examine the spatial profiles of the higher-order motifs within the dataset.

When measuring PSP amplitude correlations across convergent and divergent pairs (*Figure 9—figure supplement 2*, *Figure 10C*), assignment of a connection strength as *i* or *j* would be arbitrary and potentially influence the measurement of Pearson's *R*. Therefore, each pair of connections was plotted twice with a given connection amplitude assigned once as *i* and once as *j* and an *R* score was calculated from these values. To determine the probability of obtaining the measured *R* score (*p*-value), we used the number of unique unordered pairs in the data (one half of the number of pairs plotted). A similar approach was taken to examine recurrent connectivity between L5 pyramidal neurons (*Song et al., 2005*).

## Analysis of PSPs

Voltage recordings were excluded from measures of connection strength if the voltage preceding the photostimulus differed from the target voltage by >5 mV or if the voltage changed by >1 mV in a 50 ms window preceding the photostimulus. The latter was done to avoid the influence of rare, large amplitude spontaneous events. Individual voltage recordings were baseline adjusted by subtracting the median membrane potential in a 20 ms window before photostimulation. An average PSP was generated from the baselined sweeps. The number of two-photon-evoked presynaptic spikes and their timing varies across photostimulated neurons (*Figure 1C*, *Figure 1—figure supplement 1*). Therefore, to identify the timing of the first PSP and measure its amplitude, we used an exponential deconvolution of the average photoresponse to emphasize the timing of underlying synaptic events (*Equation 2*; *Richardson and Silberberg, 2008*). $\tau$ was set to 20 ms.

$$D\left(t\right) = V + \tau \frac{dV}{dt} \tag{2}$$

We identified peaks in the deconvolved trace that occurred 0–50 ms after the start of photostimulation. We set the threshold for peak detection to equal five times the standard deviation of the deconvolved trace prior to photostimulation (measured in a 20 ms pre-stimulus window). If a single-threshold crossing was found, we measured the peak of the average voltage response in a time window from the first threshold crossing to 20 ms later. If multiple peaks were found, the window was refined to start at the first threshold crossing and end at the time of the inter-peak minimum (*Figure 7—figure supplement 1*). The CV of PSP amplitudes was calculated using the peaks of each voltage trace within this time window. Rise time was calculated from the average photoresponse as the time taken to go from 20% to 80% of the peak amplitude. To avoid the influence of spontaneous background activity on this measure, we subset data for responses >0.1 mV in amplitude. Measurements of PSP FWHM and PSP decay were frequently marred by the presence of multiple PSPs and are not reported.

Optically identified connections used in the estimation of the PPR (*Figure 8*) were manually identified as photoresponses with multiple time-locked PSPs observed across trials. An experimenter set two nonoverlapping time windows corresponding to the first and second PSPs. Exponential deconvolution of individual photoresponses was used to account for temporal summation of PSPs (*Equation 2*; *Richardson and Silberberg, 2008*; *Loebel et al., 2009*; *Barri et al., 2016*; *Bird et al., 2016*; *Barros-Zulaica et al., 2019*). For some connections, the value of $\tau$ in Equation 2 was adjusted from a starting value of 20 ms to avoid overshoot or undershoot of the baseline. This was most often a decrease in $\tau$ to 5–10 ms for responses measured in FSIs due to the faster membrane time constants (*Table 4*) and PSP kinetics compared to L2/3 PCs (*Figure 7C*). The peak of each deconvolved photoresponse within the two windows was measured. The PPR is reported as the ratio of the second peak to the first peak. The difference in the timing of the peaks was used to estimate the inter-PSP interval.

To test for differences in connection parameters across groups (*Figure 7—figure supplement 2*, *Figure 8—figure supplement 1*), we first performed a Kruskal–Wallis (KW) test using all excitatory or all inhibitory connections grouped according to presynaptic Cre line and postsynaptic cell class. For comparisons of amplitude, rise time, and PPR, the KW test returned a low p-value, indicating that at least one group had an unequal distribution. Subsequent pairwise comparisons were made by performing a Dunn's multiple comparisons test using the scikit-posthocs Python package. Resulting p-values are reported in each matrix element. Cohen's d was used to measure the effect size of each pairwise comparison and was calculated as the difference in means normalized to the pooled standard deviations for the two groups (*Equation 3*).

$$d = \frac{(mean_2 - mean_1)}{\sqrt{((SD_1^2 + SD_2^2)/2)}} \tag{3}$$

Changes in optically evoked PSP amplitudes over the course of repeated photostimulation were characterized by linear regression of PSP amplitude versus photostimulus number for each connection. On average, PSP amplitude displayed a slight decline or 'rundown' with repeated photostimulation (across all connections, mean = –0.2% change per stimulus, median = –0.3%, std = 1.7%). Data were grouped by presynaptic Cre line and are presented as percent change per stimulus as follows: Penk: median –0.31%, mean 0.04%, std 2.5%; Rorb: median –0.68%, mean –0.53%, std 2.0%; Tlx3: median –0.89%, mean –0.87%, std 1.0%; Pvalb: median –0.15%, mean –0.02%, std 1.3%; Sst: median –0.09%, mean 0.08%, std 1.5%.

## Acknowledgements

We thank the Allen Institute founder, Paul G Allen, for his vision, guidance, and support. We are grateful for the assistance of the Allen Institute for Brain Science Neurosurgery and Behavior, Tissue Processing, and Transgenic Colony Management teams. We thank Lindsey Glickfeld, Alex Hoggarth, Brian Kalmbach, Scott Owen, and Stephanie Seeman for feedback on an earlier version of the manuscript.

## Additional information

### Funding

| Funder | Grant reference number | Author |
| --- | --- | --- |
| Allen Foundation | | Travis A Hage<br>Alice Bosma-Moody<br>Christopher A Baker<br>Megan B Kratz<br>Luke Campagnola<br>Tim Jarsky<br>Hongkui Zeng<br>Gabe J Murphy |

The funders had no role in study design, data collection and interpretation, or the decision to submit the work for publication.

## Author contributions
Travis A Hage, Conceptualization, Data curation, Formal analysis, Investigation, Methodology, Validation, Visualization, Writing - original draft, Writing - review and editing; Alice Bosma-Moody, Conceptualization, Data curation, Formal analysis, Investigation, Methodology, Software, Writing - review and editing; Christopher A Baker, Conceptualization, Investigation, Methodology, Resources, Writing - review and editing; Megan B Kratz, Data curation, Software, Writing - review and editing; Luke Campagnola, Software, Writing - review and editing; Tim Jarsky, Resources, Software, Writing - review and editing; Hongkui Zeng, Resources, Supervision, Writing - review and editing; Gabe J Murphy, Conceptualization, Resources, Supervision, Writing - review and editing

## Author ORCIDs
Travis A Hage ![ORCID] http://orcid.org/0000-0002-6125-2768
Christopher A Baker ![ORCID] http://orcid.org/0000-0002-0604-8449
Tim Jarsky ![ORCID] http://orcid.org/0000-0002-4399-539X
Hongkui Zeng ![ORCID] http://orcid.org/0000-0002-0326-5878

## Ethics
This study was performed in strict accordance with the recommendations in the Guide for the Care and Use of Laboratory Animals of the National Institutes of Health. All of the animals were handled according to approved institutional animal care and use committee (IACUC) protocols (#1807 and #2110) of the Allen Institute.

## Decision letter and Author response
Decision letter https://doi.org/10.7554/eLife.71103.sa1
Author response https://doi.org/10.7554/eLife.71103.sa2

# Additional files

## Supplementary files
• Transparent reporting form

## Data availability
Source data files have been provided for all figures. Code to generate the primary figures within this manuscript is publically available at https://github.com/travis-open/twop_opto_data, (copy archived at swh:1:rev:b71bccd0a5a0d5a70c37d53b4c588d1c00c2a531). This github repository includes a csv file containing quantitative electrophysiological features and metadata for all tested synaptic connections. Neurodata without borders (nwb) files containing original electrophysiological recordings are archived as a Dryad Digital Repository.

The following dataset was generated:

| Author(s) | Year | Dataset title | Dataset URL | Database and Identifier |
|---|---|---|---|---|
| Hage TA | 2021 | two-photon optogenetics mapping in mouse visual cortex | https://datadryad.org/stash/share/1aykN8S3HOGEPhFIkzVFq3pmCVNawzFRTMs5BJxCY9s | Dryad Digital Repository, 10.5061/dryad.zs7h44jbc |

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
