## [Editor Report]

Hage and colleagues used channelrhodopsin-mediated 2 photon stimulation combined with genetic labeling to survey the synaptic connections from layers 4, 5 and 6 onto layer 2/3 pyramidal neurons in the adult mouse primary visual cortex (V1). This work not only confirms prior knowledge regarding synaptic connectivity made through paired intracellular recordings, but also provides important new parameters and constraints about these connections. The results will contribute to our understanding of the cortical information processing.

---

## [Decision Letter]

**Decision letter after peer review:**

[Editors’ note: the authors submitted for reconsideration following the decision after peer review. What follows is the decision letter after the first round of review.]

Thank you for submitting your work entitled "Interlaminar synaptic connectivity in mouse primary visual cortex measured by two-photon optogenetic stimulation" for consideration by *eLife*. Your article has been reviewed by 3 peer reviewers, one of whom is a member of our Board of Reviewing Editors, and the evaluation has been overseen by a Reviewing Editor and a Senior Editor. The reviewers have opted to remain anonymous.

Our decision has been reached after consultation between the reviewers. Based on these discussions and the individual reviews below, we regret to inform you that your work will not be considered for publication in *eLife* at its current form. While the significance and the impact of the work is well recognized by the reviewers, two key control experiments are deemed critical (#1 and #2). If these experiments are complete, we will be happy to look at the new submission. We have summarized the major and minor concerns from the reviewers as well as the appended the full reviews.

In this manuscript, Hage and colleagues applied channelrhodopsin-mediated 2 photon stimulation to measure the synaptic connections from layers 4, 5 and 6 to layer 2/3 pyramidal neurons in the adult mouse primary visual cortex (V1). This work built upon rich literature of the cortical microcircuits of V1. Consistent with previous studies, the authors find the highest connections to layer 2/3 pyramidal neurons are from layer 4 excitatory and layer 2/3 SST neurons. Furthermore, the authors observe a log-normal distribution of connection strengths. The current work is focused on addressing adult mice circuits compared to most of the published work using juvenile mice, and is using two photon to reach single cell stimulation resolution. Because of the single-cell resolution and efficiency of the light stimulation, one main advantage of the current study is to be able to examine the input convergence to L2/3 pyramidal neurons and divergence of L4 *Rorb*+, L5 Tlx3+ and Sst+ neurons to L2/3 pyramidal neurons. Overall, this manuscript showed that 2p photostimulation can be used to map local cortical circuits. Regarding the biological findings, it validates previous work using paired-recording and 1p photostimulation and emphasize on the SST neurons contribution to the local circuits in V1.

Concerns:

1. With the combination of the 2photon stimulation and recent developed ChR2 variants, the reviewers think it is crucial to validate that the observed connections using this method are bonafide synaptic connections and to evaluate false positive and false negative rates with the method. This requires pair recordings in the following two aspects: (i) patching the optically identified putative pre and postsynaptic pair; (ii) using pair recordings to test whether the connective rates between populations of neurons is true at the statistically level. To make this feasible in the defined time frame, this will be best and easiest to do in the layer2/3 to layer 2/3 connections and is also connected with the requested experiment point 2.

2. It is essential to evaluate connections between layer 2/3 pyramidal neurons. This will serve as a benchmark to be compared to not only young adult results known in laye2/3 but also to the rich literature of layer2/3 connectivity in other primary cortices. Given previous findings that local interactions in layer 2/3 may contribute to V1 tuning characteristics, this is also important to the implication in the visual biology.

3. Several major claims need to be tuned and carefully worded to reflect what the data actually suggest.

i) The cre lines label cells that are restricted to a cortical layer ≠ they define a cortical layer. This has to be carefully distinguished when interpreting the results and articulating the claims. In the abstract and throughout the manuscript, it needs to be made clear to the readers that the connections are to layer-specific cell types, not the layers. This is one of the differences between cre-dependent optogenetic mapping vs. glutamate uncage mapping utilizing the endogenous glutamate receptors.

ii) Given that response amplitude is dependent on pre- and post-synaptic properties that may be highly dynamic in vivo, and measurement of IPSP amplitudes almost at the reversal potential for GABA-A receptors has limited utility, it is tricky to use the PSP amplitude to calculate the connection rate in a static slice. It would be far more compelling to report currents (or better, conductance) as a readout of relative response magnitude. This point needs to be discussed and conclusions would be better softened based on such amplitude data.

iii) Title "interlaminar synaptic connectivity" suggests that this would be a broad survey of ascending and descending connections between many cell types across the layers, but this work only contains ascending connections onto L2/3 pyramidal cells. Reviewer suggests amending title to reflect that all inputs are converging onto L2/3 pyramidal cells.

iv) For divergence analysis, Tlx3 has a small data size (5 data points) for statistics and in layer 5, 1 connection determined the conclusion (1 divergent/249 probed). Strong claim was made based on this small data size. The authors need to either increase n number, or show that the current n number is enough for the statistic tests.

4. Data analysis and display:

i) The authors only show half of the graph on the lower two rows in Figure 1 – Supplement 1. The authors should show the activity both above and below the target cell and state the full width at half maximum for the resolution in xy and z of the photostimulation.

ii) The authors make a good point that patch-clamp technique has its flaws when it comes down to qualitatively measuring inputs from distal locations in Panels 4C-D. The authors made the case that the connectivity rates between SST to L23 and SST to L5 are relatively similar when recorded in current or voltage clamp. However the authors did not compare the actual connectivity rate between the current clamp and voltage clamp. Based on the plots, it looks like SST to L2/3 current vs SSt to L2/3 voltage clamp might be significantly different, whereas SST to L4/5 current vs SST toL4/5 voltage clamp may not appear to be different. If it is true, this would suggest that the connectivity rates measured in all other figures might be affected but not in a linear fashion across all connections. Please add statistic comparison of the connectivity using voltage and current and discuss the significance of the results.

iii) In Table 1, the Kolmogorov-Smirnov test is used on data for which the distribution is not fully specified. Rather, the authors state that they fit the data and then tested the goodness of fit, which invalidates the calculation of the critical region. In addition, the p-value obtained with this method seems to suggest that the fit is in fact better for the normal distribution than the lognormal distribution. Perhaps the authors can confirm the calculations or discuss this issue.

iv) Connection rate data display should show individual data points, instead of just a line so readers can easily grasp the bin size, and where they fall into the cortical depth and lateral position. For example, in Figure2B right, there are positive green cells around 300 lines, and there is no data point there. It is hard to see how to get the green line in the graph from the green triangles in the middle.

5. Discussion points:

i) Please add discussion of the rationale for using 3 methods (manual, SVM, and threshold) to threshold the response. Also add description of how objectively the human determined a synaptic input. Related to the response threshold, it is also important to discuss how polysynaptic events were differentiated from monosynaptic events.

ii) Discussion of false-positive rate and false negative rate and potential sources of them. For example, what is the estimated labeling efficiency of labeling presynaptic cells? In order to validate the connectivity rate, it is important to understand how many of neuronal population that was aimed to be sampled was actually labeled with channelrhodopsin. An example where this concern might be reflected is in panel 4B. Although the same SST-Cre driver line was used, the number of sampled cells in L4 is widely different between Ai167 and AAV experiments, resulting in a doubling of the connectivity rate in L4. In general, discussion of the differences of the results from transgenic lines and the AAV viral approach is needed.

iii) Discussion on the distribution of synapses in Figure 5. Does this suggest an inverse sombrero-function of activation where excitation from neurons in neighboring columns distal neurons is suppressed by inhibitory neurons in the same column? Does short-term synaptic plasticity shaping response amplitudes?

iv) In Figure 3C, there is strong asymmetry on L6 inputs at horizontal axis. Explain and discuss the implication.

*Reviewer #1:*

Preliminary comments:

In this manuscript, Hage and colleagues applied channelrhodopsin-mediated 2 photon stimulation to measure the synaptic connections from layers 4, 5 and 6 to layer 2/3 pyramidal neurons in the mouse primary visual cortex (V1). There are a lot of literature about the cortical microcircuits of V1. The main differences between the current work and the existing work are that the mice used are older than most of the previous work (adult vs. young adult) and the stimulation resolution is much higher (2 photon vs. 1 photon). Because of the single-cell resolution and efficiency of the light stimulation, one main advantage of the current study is to be able to examine the convergence of the inputs to L2/3 pyramidal neurons and divergence of L4 *Rorb*+, L5 Tlx3+ and Sst+ neurons to L2/3 pyramidal neurons. Overall, this manuscript showed convincing evidence that 2p photostimulation can be used to map local cortical circuits. Regarding the biological findings, there are not much novel findings using the 2P stimulation. Instead, it mostly validates many previous work using paired-recording and 1p photostimulation. Part of the reason is that although the stimulation is with 2p single cell resolution, in the analyses only the reduced resolution was used (bin size is 150 μm along cortical depth and 100 μm along horizontal axis), which is comparable with 1p mapping resolution.

Concerns:

1). The cre lines label cells that are restricted to a cortical layer ≠ they define a cortical layer. This has to be very carefully distinguished when interpreting the results and articulating the claims. For example, ntst-cre do not define L6 and ntsr-cre only labels deep two thirds of the layer 6. Along the line of shallower L5 neurons are more likely to connect to L2/3 pyramidal neurons, one wonders whether a cre line labeling shallower L6 would have the same results. Similarly, Tlx3 is one of many cre lines that label only a subset of L5 neurons. For exmaple, Efr3a-cre also contains contico-cortical projections (Kim et al., 2014). This issue is further exemplified in the case that L5 *Rorb*+ and Tlx3+ show quantitatively different results, and L4 *Rorb*+ and Scnn 1a+ are quantitatively different (Figure 3B, right). To this point, if L4 *Rorb* and Scnn1a represent independent cell types, should the connection rate of L4 is at least the sum of the L4 *Rorb*+ and Scnn1a+ results? To fix this problem, we should either ask the authors to have at least one other cre line in each layers, or to tone down in the entire paper what they can claim.

2) Connection rate data display should show individual data points, instead of just a line so readers can easily grasp the bin size, and where they fall into the cortical depth and lateral position. For example, in Figure2B right, there are positive green cells around 300 lines, and there is no data point there. It is hard to see how to get the green line in the graph from the green triangles in the middle.

3) The different results from transgenic lines and the AAV approach: first, it is well appreciated that the authors are comparing the results from two gene delivery methods. However, the observed differences were not addressed. E.g., Figure 4B right panel, green and red line look different within L2/3 range which is the main focus of the manuscript. Was the difference has anything to do with the ectopic expression of Sst-cre line to PV+ neurons or other cell types that have been observed in this particular line? Along the same line, in figure 3B, when it stated that *Rorb*-cre data were plotted using both AAV and transgenic, were the positive cells from the two are combined to generated the connection, or it is the average of the two? If it is combined, it created the bias towards L2/3 cells; if it is averaged between the two, given how different they are quantitatively in Figure 2B right panel, this also created potential bias since other layers were not operated the same way.

4) For divergence analysis, Tlx3 has a small data size (5 data points) for statistics. Therefore, for L5, based on 1 connection, it leads to the conclusion that 'the observed rates of divergence do not depend on the intersomatic distance between postsynaptic neurons' (1 divergent/249 probed). Also based these 5 data point, the authors made strong claim that 'broad diverging (Figure 7) local projections from L5 IT neurons could provide a mechanism to distribute this information within V1…..". The authors need to either increase n number, or show that the current n number is enough for the statistic tests.

*Reviewer #2:*

There is a fundamental flaw in this study: the authors did not confirm any the putative connections that they report in this work by performing dual whole cell recordings from stimulated and postsynaptic cells. Photostimulation of off-target cells may lead to false positive errors wherein off-target cells fire concomitantly with the on-target cell. If the on-target cell is not monosynaptically connected to the patched cell but the off-target cell is connected, the authors may incorrectly attribute a synaptic connection to the on-target cell. This is not a theoretical possibility since opsins are expressed in axons, presynaptic terminals and dendrites. The authors should perform dual patch clamp experiments, electrically stimulate the putative presynaptic cells, confirm the connectivity and the connection amplitude, and report the false-positive and false negative rate of their photostimulation. This is critical and, in fact, has been done in similar studies previously for more than a decade. Otherwise, their entire study is up for grabs.

*Reviewer #3:*

The study by Hage et al. uses 2p photostimulation of ChrimsonR-expressing excitatory and inhibitory neurons to measure connectivity with postsynaptic L2/3 pyramidal neurons. Consistent with previous studies, they find the highest connections from layer 4 excitatory and layer 2/3 SST neurons. Overall, they also observe a log-normal distribution of connection strengths. These results build on a relatively large literature of similar work, but the findings do make an important addition to the field. In particular, the data sets for convergence and divergence of inputs are made possible by the outstanding multi-patch/photostimulation combination approach, substantially increasing the throughput of the experiments. Moreover, the older age of the mice versus previous studies presents an important point of comparison. The data are of high quality, and I have only a few concerns that should be addressed.

1. My biggest concern is why connections between layer 2/2 PNs were not investigated here. Even as a comparison point, it seems strange to omit intralaminar connectivity from the study. Given previous findings that local interactions in layer 2/3 may contribute to V1 tuning characteristics, it would be very helpful to get a sense of the relative contribution of intra- versus interlaminar wiring using the present method.

2. A second major concern is with the interpretation of response amplitude. As long as the response is large enough to be statistically above baseline, it seems reasonable to categorically judge that a functional "connection" exists. Moreover, one would expect this anatomical connection observed in the slice to be good evidence that such a connection exists in vivo. However, response amplitude is dependent on presynaptic (release probability) and postsynaptic (receptor complement and biochemical state) properties that may be highly dynamic in vivo. In addition, measuring IPSP amplitudes almost at the reversal potential for GABA-A receptors has limited utility. It would be far more compelling to report currents (or better, conductances) as a readout of relative response magnitude. Overall, it is unclear to me what conclusions to draw about PSP amplitudes in a static slice. I would urge the discussion of this point and perhaps a softening of the conclusions drawn from the amplitude data.

3. The ACSF used in this study is relatively "high divalent" (2 ca^2+^ and 2Mg^2+^). This will raise both spike threshold and release probability, potentially skewing measures of PSP amplitude (in opposite directions) due to excessive axon conduction failure and non-physiologically high release to single spikes. This issue should at least be discussed (and the choice of ACSF explained).

4. On a slightly related note, there is no consideration for short-term synaptic plasticity shaping response amplitudes. in vivo, cellular activity can average 1-5Hz, producing sustained synaptic depression that shapes amplitudes. Given the presence of multi-spike PSPs (as illustrated in Figure 5S1), it might be possible to estimate the amount of depression/facilitation at some of the connections. These data may be modest, but would nevertheless be potentially very interesting to the field.

5. Please note the AAV titer in the methods.

[Editors’ note: further revisions were suggested prior to acceptance, as described below.]

Thank you for submitting your article "Synaptic connectivity to L2/3 of primary visual cortex measured by two-photon optogenetic stimulation" for consideration by *eLife*. Your article has been reviewed by 3 peer reviewers, one of whom is a member of our Board of Reviewing Editors, and the evaluation has been overseen by Ronald Calabrese as the Senior Editor. The reviewers have opted to remain anonymous.

Essential revisions:

1) Present cell density counts for each Cre line and for viral infection and add high quality images of representative expression.

2) Address the comments from reviewer #3 for clarification

3) Scholarly cite the previous work using 2p synaptic mapping in the primary cortices.

4) Add description of PENK-cre line in the Methods.

*Reviewer #1:*

In this manuscript, Hage and colleagues applied channelrhodopsin-mediated 2-photon stimulation to measure the synaptic connections from layers 4, 5 and 6 to layer 2/3 pyramidal neurons in the adult mouse primary visual cortex (V1). With the high throughput of photostimulation and high stimulation resolution, the current study examined the convergence of the inputs to L2/3 pyramidal neurons using genetically defined cell population. The results are also compared with the gold standard of paired recordings. In this resubmission version, the manuscript shows convincing evidence that 2p photostimulation can be used to map local cortical circuits. This work validates many previous work using paired-recording and 1p photostimulation and further provides cell type-specific cortical connectivity.

The authors addressed all of my previous major concerns by including the pair recording data and increased sample size. I now support the publication of the work.

*Reviewer #2:*

After their reticence, in this revised version the authors have finally carried out the critical dual recording control experiments that validate their method. The predictability is very high so their putative presynaptic neurons appear to be a faithful representation of the underlying connectivity. They have also toned down some of their claims and added novel statistics. I am satisfied with their response and the paper has been greatly improved. The only remaining issue is to put this work in the methodological context of previous two-photon mapping of excitatory and inhibitory cortical connections, with caged glutamate and optogenetics. The abstract, introduction and discussion are written as if the authors were the first to use two-photon mapping of synaptic circuits, ignoring any mention to previous studies. While their method may represent the current state-of-the-art in terms of throughput, genetic lines and opsins, this is not the first time that a similar approach has been published. Previous work pioneering two-photon mapping should be properly acknowledged, for the same collegiality reasons than the authors would like their own work to be properly acknowledged by subsequent studies.

*Reviewer #3:*

Hage and colleagues present a semi-comprehensive survey of functional synaptic connectomics onto L2/3 pyramidal cells (PCs) in mouse V1. They go broad – sampling a good variety of Cre lines that sparsely label cell types in different layers and two major classes of GABAergic neurons. They provide an admirable and rigorous analysis of the effective spatial resolution of their photo-stimulation and perform difficult and important ground truth validation of their optogenetic approaches using targeted paired recording. Overall, I find the experiments performed to a high caliber and the conclusions largely sound. One key issue is the density of the opsin expression in these Cre lines is not quantified – neither for the Cre expression per se in each line nor the infectivity ratio, making it hard to assess the denominator in the connectivity frequency estimates. However, the presumed sparsity of expression (since they typically only sampled a relatively small number of Cre+ cells per recording with manual selection) does help mitigate any issues with 'off target' activation since their resolution is not at the single cell level.

The authors analyze their extensive data in a number of logical and expected ways, quantifying connection probability and strength vs cell type and horizontal distance from the patched PC, as well as a nice analysis of convergence, divergence, and reciprocal connections. Nearly all of their findings confirm prior work from pair recordings, helping validate their approach, with the major exception being the stronger input from L5 intratelecenphalic neurons to L2/3 PCs which is certainly interesting and unexpected based on past work.

My criticism is this study is nearly overly broad but somewhat shallow – there are many results but limited new insight gained into the underlying connectivity, and few impactful or entirely new findings. The study establishes their approach as a powerful and valid technique for sampling connectivity in this manner, but the approach has been used by others, including one of the co-authors. Thus, I am uncertain of the true impact of this study. Nevertheless, it is well executed and convincing that this approach is powerful and well-controlled.

1) Please show images of each Cre line in the Ai167 line and with viral infection. Please quantify penetrance and/or infectivity by staining for NeuN and other appropriate strains. My sense is the expression was very sparse in most cases but this is not clearly explicated or quantified.

2) The authors motivate the use of a higher power (85 mW in a small spot!) because this will spike most cells based on a very nice stimulation power analysis. This reduces false negatives but should increase false positives by increasing off target activation. This might be mitigated by very sparse expression (see point 1). Not worth redoing, but I would suggest using less power at the expense of spiking every cell in the future, particularly in denser expressing lines.

3) Based on point 1 and 2 the authors could provide estimates of how many cells on average each photo-stimulus activates in each line. This would be informative.

4) The FWHM numbers in Figure 1 and Figure 1 supplement just don't seem to match up to my eye. Perhaps the FWHM or the average has been computed improperly, or Figure 1 is reporting HWHM? From Figure 1 – Supplement 1 it's clear the 85 mW is driving saturation of spiking (hence the PSFs look 'squared off' and using less power would be preferable). Is is odd the PSFs are saturated in the z axis by not in the x axis, I can't seem to wrap my mind around that. In many of the individual examples (grey lines) the z axis FWHM looks much larger than the average, sometimes almost 100 microns. This is a cause for concern. Figure 1 S1 implies each photo-stimulation may be activiting quite a few cells along the z-axis depending on expression density.

5) Please provide more information on the Penk-Cre line, it's not actually listed in the Methods section and not a super-well established line. From the Allen website it looks very sparse and biased to L2(?). Is it all pyramidal cells, what shows this?

6) Though the study probes ~10,000 connections, this is actually not that much. Lefort et al. probed thousands by patching 6 cells at a time and doing many recordings. In table 1 the data says only 205 Exc to Exc connections were probed in L2/3. This is actually very few, not even double their paired recordings. Some explanation for the lower number is warranted – is this related to the very sparse nature of opsin expression in each line?

7) The paired recording confirmation of optically identified pairs is admirable, but if opsin expression was extremely sparse to begin with, this really isn't that compelling. In Figure 3 there appears to be very few opsin+ neurons, so their high confirmation rate is not surprising. This being said, it doesn't negate the proof their approach works, but it doesn't imply the approach would work for dense expression. If this is a real limitation of this approach is much be stated much more forthrightly in the text so other groups don't adopt this approach without carefully checking their spatial resolution. There is nothing wrong with using sparse expression for what they do in this study, but it needs to be more clear to the reader.

8) Are L5 *Rorb*-Cre cells more similar to L4 neurons or other L5 neurons? This is not really for this study, but would be nice to state since the AI must have the transcriptomic data.

9) What was the motivation for sometimes using the Ai167 line (not soma-targeted) and other times an AAV? Some clarification is warranted.

10) For Figure 6D, can the Exc to L2/3 be split according to presynaptic layer for clarity?

11) For the paired pulse analysis the authors identify high frequency bursts of the presynaptic cell in the postsynaptic potentials, with 5-10 ms ISIs. This is very fast (100-200Hz) and not typical for most paired pulsed analysis, although still interesting. It would have better to deliberately generate paired pulse or pulse train responses of connected neurons at lower, more physiological frequencies for most cell types (say 10-40 Hz). Also, I'm not convinced the deconvolution analysis for estimated PPF is valid, this could be assessed, at least at lower frequencies, by comparing against voltage clamp measurement of the same inputs in the same cells (this would not be compromised by space clamp since PPF measurements are of the same equally poorly clamped synapse).

---

## [Author Response]

[Editors’ note: the authors resubmitted a revised version of the paper for consideration. What follows is the authors’ response to the first round of review.]

In this manuscript, Hage and colleagues applied channelrhodopsin-mediated 2 photon stimulation to measure the synaptic connections from layers 4, 5 and 6 to layer 2/3 pyramidal neurons in the adult mouse primary visual cortex (V1). This work built upon rich literature of the cortical microcircuits of V1. Consistent with previous studies, the authors find the highest connections to layer 2/3 pyramidal neurons are from layer 4 excitatory and layer 2/3 SST neurons. Furthermore, the authors observe a log-normal distribution of connection strengths. The current work is focused on addressing adult mice circuits compared to most of the published work using juvenile mice, and is using two photon to reach single cell stimulation resolution. Because of the single-cell resolution and efficiency of the light stimulation, one main advantage of the current study is to be able to examine the input convergence to L2/3 pyramidal neurons and divergence of L4 Rorb+, L5 Tlx3+ and Sst+ neurons to L2/3 pyramidal neurons. Overall, this manuscript showed that 2p photostimulation can be used to map local cortical circuits. Regarding the biological findings, it validates previous work using paired-recording and 1p photostimulation and emphasize on the SST neurons contribution to the local circuits in V1.Concerns:1. With the combination of the 2photon stimulation and recent developed ChR2 variants, the reviewers think it is crucial to validate that the observed connections using this method are bonafide synaptic connections and to evaluate false positive and false negative rates with the method. This requires pair recordings in the following two aspects: (i) patching the optically identified putative pre and postsynaptic pair; (ii) using pair recordings to test whether the connective rates between populations of neurons is true at the statistically level. To make this feasible in the defined time frame, this will be best and easiest to do in the layer2/3 to layer 2/3 connections and is also connected with the requested experiment point 2.2. It is essential to evaluate connections between layer 2/3 pyramidal neurons. This will serve as a benchmark to be compared to not only young adult results known in laye2/3 but also to the rich literature of layer2/3 connectivity in other primary cortices. Given previous findings that local interactions in layer 2/3 may contribute to V1 tuning characteristics, this is also important to the implication in the visual biology.

In order to satisfy the requirement that we measure excitatory connectivity between L2/3 pyramidal neurons, we performed 2P-optogenetic mapping using the Penk-Cre line with AAVmediated expression of soma-targeted ChrimsonR (Figure 3). In addition to measuring excitatory connections between L2/3 pyramidal neurons, we also measured excitatory connectivity onto subclasses of interneurons within L2/3. In agreement with published findings, we found that recurrent excitatory connectivity among L2/3 pyramidal cells was sparse, while connectivity from L2/3 excitatory neurons to Pvalb and VIP interneurons was dramatically and significantly higher. In all cases, measured connection probabilities were statistically similar to measurements from paired patch-clamp recordings in the Allen Institute Synaptic Physiology dataset (Figure 3C).

We expanded this analysis to include intralaminar inhibition from Pvalb and Sst interneurons (Figure 3D, Figure 3—figure supplement 2). As with excitatory connectivity, we did not observe any statistically significant differences between measurements from 2P optogenetics and paired recordings (Figure 3 & Table 1). We believe the additional data and analyses more than fully satisfy the reviewers’ request for statistical comparisons of our photostimulation data to paired recordings among L2/3 pyramidal neurons. To the best of our knowledge, the breadth of these comparisons (multiple pre and postsynaptic cell classes at multiple intersomatic distances) is unique among photostimulation studies, and we believe this to be a major strength of our revised manuscript.

We tested optically identified excitatory and inhibitory connections by subsequent patching of the putative presynaptic neurons (Figure 4, Figure 4—figure supplement 1). For excitatory connections, we focused on translaminar connections identified using the *Rorb* and Tlx3 Cre lines due to the relative lack of gold standard data to compare against. In total, 11 out of 12 excitatory connections were verified (4/4 with *Rorb*:Ai167, 4/4 with *Rorb*:AAV, 3/4 with Tlx3:AAV). In experiments with inhibitory Cre lines, 22 out of 22 connections were verified (15/15 with Sst:AAV, 7/7 with Pvalb:AAV). Throughout the manuscript, a total of 8 such examples are shown (including the observed false positive; Figure 4, Figure 4 —figure supplement 1, and Figure 5 —figure supplement 3). The fraction of optically identified connections verified is similar to previous two-photon stimulation approaches using either C1V1 or caged glutamate to confer photosensitivity. Furthermore, PSP amplitudes measured in mapping experiments and after patching of the presynaptic cell were very strongly correlated (Figure 4 —figure supplement 1).

3. Several major claims need to be tuned and carefully worded to reflect what the data actually suggest.i) The cre lines label cells that are restricted to a cortical layer ≠ they define a cortical layer. This has to be carefully distinguished when interpreting the results and articulating the claims. In the abstract and throughout the manuscript, it needs to be made clear to the readers that the connections are to layer-specific cell types, not the layers. This is one of the differences between cre-dependent optogenetic mapping vs. glutamate uncage mapping utilizing the endogenous glutamate receptors.

We have made changes throughout the manuscript to establish early and to frequently remind readers that we are using the indicated Cre lines and have adjusted our conclusions to better reflect this. For example, we previously highlighted that we observed more frequent translaminar excitatory connections from L5 to L2/3 than previous studies, and we suggested in Discussion that genetic targeting of L5 IT-type neurons within our study was one possible reason. We’ve extended this discussion and, as the reviewers suggest, we more explicitly made the comparison between optogenetic mapping and glutamate uncaging.

ii) Given that response amplitude is dependent on pre- and post-synaptic properties that may be highly dynamic in vivo, and measurement of IPSP amplitudes almost at the reversal potential for GABA-A receptors has limited utility, it is tricky to use the PSP amplitude to calculate the connection rate in a static slice. It would be far more compelling to report currents (or better, conductance) as a readout of relative response magnitude. This point needs to be discussed and conclusions would be better softened based on such amplitude data.

In the revised Discussion section, we more explicitly discuss how the effect of a synaptic connection on the postsynaptic membrane potential is influenced by multiple complex biological mechanisms including presynaptic transmitter release, postsynaptic receptor activation and the intrinsic electrophysiological properties of the cell. We also state that the features measured in the acute slice preparation may differ in vivo due to differences in ionic gradients and the presence of neuromodulators that dynamically influence synaptic transmission. We discuss our decision to use current-clamp recordings and acknowledge its limitations (also related to *Concern 4ii* regarding somatic recording bias). As part of this discussion, we cite several studies that highlight the challenges of using voltage-clamp to measure synaptic currents in morphologically complex cells, as well as several studies that have measured PSP amplitudes to describe the strengths of synaptic connections (similar to our study).

iii) Title "interlaminar synaptic connectivity" suggests that this would be a broad survey of ascending and descending connections between many cell types across the layers, but this work only contains ascending connections onto L2/3 pyramidal cells. Reviewer suggests amending title to reflect that all inputs are converging onto L2/3 pyramidal cells.

The title of the manuscript has been changed to “Synaptic connectivity to L2/3 of primary visual cortex measured by two-photon optogenetic stimulation”

iv) For divergence analysis, Tlx3 has a small data size (5 data points) for statistics and in layer 5, 1 connection determined the conclusion (1 divergent/249 probed). Strong claim was made based on this small data size. The authors need to either increase n number, or show that the current n number is enough for the statistic tests.

We previously limited our statistical comparison of divergence to two intersomatic distances (0-100 and 100-200 microns) due to a limitation of the SciPy stats package which only supports 2x2 tables for Fisher’s Exact tests. We have exported data to RStudio which can accommodate larger contingency tables for the Fisher’s Exact tests and report an updated p-value that includes all instances of divergence (including intersomatic distances of 200-300 microns where divergence from Tlx3 was most frequent). We have also added bootstrap analyses regarding the abundance of convergence and divergence motifs observed (Figure 9 —figure supplement 1, Figure 10 —figure supplement 1). We no longer speak to the broad divergence of Tlx3 specifically, but instead highlight differences in the patterns of divergence observed across inhibitory and excitatory Cre lines.

4. Data analysis and display:i) The authors only show half of the graph on the lower two rows in Figure 1 – Supplement 1. The authors should show the activity both above and below the target cell and state the full width at half maximum for the resolution in xy and z of the photostimulation.

These changes were made.

ii) The authors make a good point that patch-clamp technique has its flaws when it comes down to qualitatively measuring inputs from distal locations in Panels 4C-D. The authors made the case that the connectivity rates between SST to L23 and SST to L5 are relatively similar when recorded in current or voltage clamp. However the authors did not compare the actual connectivity rate between the current clamp and voltage clamp. Based on the plots, it looks like SST to L2/3 current vs SSt to L2/3 voltage clamp might be significantly different, whereas SST to L4/5 current vs SST toL4/5 voltage clamp may not appear to be different. If it is true, this would suggest that the connectivity rates measured in all other figures might be affected but not in a linear fashion across all connections. Please add statistic comparison of the connectivity using voltage and current and discuss the significance of the results.

All voltage-clamp measurements were made using Sst:Ai167 animals, in large part because these mice were more readily available than mice having received stereotaxic injection of AAV. Comparing Sst:Ai167 connectivity measured by current-clamp and voltage-clamp yields the following results:

**Author response table 1. sa2table1:** 

	I-clamp connected /probed	V-clamp connected /probed	I-clamp connection probability	V-clamp connection probability	Fisher's Exact pvalue (I-clamp vs V-clamp)	Fisher’s Exact odds ratio
L2/3 Sst:Ai167 to L2/3 pyr. cell	24/52	16/25	0.46	0.64	0.15	0.48
Translaminar(L4/L5)Sst:Ai167 toL2/3 pyr. cell	2/71	6/73	0.03	0.08	0.28	0.32
L5 Sst:Ai167 to L5 pyr. cell	25/147	20/68	0.17	0.29	0.05	0.49

*for all probed connections with <100 µm horizontal intersomatic distance, p-values are uncorrected for multiple comparisons.

The statistical significance of the comparisons between current-clamp and voltage-clamp recording was far less conclusive than the differences in Sst to pyramidal cell connectivity according to layer using either recording configuration (i-clamp: L2/3 vs L5, p=6e-05; v-clamp: L2/3 vs L5, p=0.004). While our data do not provide strong statistical support for the nonlinearity suggested by the reviewers, their point is well taken, and we have amended our discussion of proximity bias in somatic recordings to acknowledge that the bias is unlikely to be uniform when considering connections to and from different subclasses of neurons.

For our initial submission, we included Sst to L5 pyramidal cell data, as it demonstrated the utility of two-photon optogenetics to measure differences in connectivity from one Cre line to multiple postsynaptic subclasses (Sst→L5 pyr. cell connectivity was lower than Sst→L2/3 pyr. cell connectivity). In the revision, we now make similar comparisons between L2/3 pyramidal cells and L2/3 fast-spiking interneurons for several presynaptic Cre lines (Penk, *Rorb*, Tlx3, Pvalb, and Sst). With the additional data and analyses in our revision (2 new primary figures and 10 additional figure supplements), we think it is beneficial to remove data measuring L5 Sst→L5 pyramidal connectivity to keep the manuscript focused on L2/3 as a receiving layer and stay closer to the recommended word count. We plan to share data measuring connectivity from Sst interneurons to L5 pyramidal cells along with the data in this manuscript as part of the associated GitHub and Dryad repositories.

iii) In Table 1, the Kolmogorov-Smirnov test is used on data for which the distribution is not fully specified. Rather, the authors state that they fit the data and then tested the goodness of fit, which invalidates the calculation of the critical region. In addition, the p-value obtained with this method seems to suggest that the fit is in fact better for the normal distribution than the lognormal distribution. Perhaps the authors can confirm the calculations or discuss this issue.

We agree that it was a mistake to use a KS test for this analysis and thank the reviewers for bringing this to our attention. The low p-values obtained by comparing the observed amplitude distributions to a normal distribution indicated that we should reject the null hypothesis that the distributions were the same, so we believe our interpretation was correct in that regard. With the additional data added to the manuscript, we have modified the section on PSP amplitudes to focus on differences in synaptic strength and kinetics according to pre- and postsynaptic cell identities. We still highlight that many of the distributions of PSP amplitudes are broad and highly skewed (Figure 7, Table 2), but do not argue for the use of a lognormal distribution specifically to describe the data. We believe the comparisons between pre- and postsynaptic cell-classes will be of greater interest to a broad readership, and we appreciate the opportunity to share the measured distributions as supplemental data files for readers who wish to dig deeper.

iv) Connection rate data display should show individual data points, instead of just a line so readers can easily grasp the bin size, and where they fall into the cortical depth and lateral position. For example, in Figure2B right, there are positive green cells around 300 lines, and there is no data point there. It is hard to see how to get the green line in the graph from the green triangles in the middle.

We have added distinct markers to all individual estimates of connection probability. We have added a statement to each figure legend that estimated connection probabilities and confidence intervals were not drawn if fewer than 20 connections were probed. These cases most often corresponded to the cortical depths that were sparsely labeled by a given Cre line, such as near the L2/3-L4 border for *Rorb*-Cre in the figure mentioned by the reviewer (now Figure 4C). For the revision, we relaxed this minimum number of connections probed and now include additional connectivity estimates with broader confidence intervals. We believe it is beneficial to leave some threshold in place. In extreme cases illustrating 95% confidence intervals will require significantly expanding the corresponding axis range and make it more difficult to visualize and compare regions of the plots where our sampling (and statistical confidence) is higher.

5. Discussion points:i) Please add discussion of the rationale for using 3 methods (manual, SVM, and threshold) to threshold the response. Also add description of how objectively the human determined a synaptic input. Related to the response threshold, it is also important to discuss how polysynaptic events were differentiated from monosynaptic events.

We have more explicitly and narratively described our rationale for using the 3 methods (to strengthen our confidence in our results and avoid influence of unintended bias on our conclusions) in the Results section. Further details were also added to the Methods.

We added multiple lines of evidence that lead us to believe polysynaptic events were uncommon in our dataset. This includes the results of new experiments confirming connections by subsequent patching – which themselves suggest the majority of optically identified connections arise from the activity of individual presynaptic neurons. From these verified connections, we also measured presynaptic AP to EPSP latencies. These measures were combined with measured photostimulus to presynaptic AP latencies (from characterization experiments of Figure 1) to generate a distribution of photostimulus to EPSP latencies that would be consistent with monosynaptic connections (Figure 5 —figure supplement 3). This predicted distribution and the observed distribution of photostimulus to EPSP latencies are strikingly similar. However, we acknowledge here and in the corresponding section of the Results that we cannot completely rule out polysynaptic events for any given connection.

ii) Discussion of false-positive rate and false negative rate and potential sources of them. For example, what is the estimated labeling efficiency of labeling presynaptic cells? In order to validate the connectivity rate, it is important to understand how many of neuronal population that was aimed to be sampled was actually labeled with channelrhodopsin. An example where this concern might be reflected is in panel 4B. Although the same SST-Cre driver line was used, the number of sampled cells in L4 is widely different between Ai167 and AAV experiments, resulting in a doubling of the connectivity rate in L4. In general, discussion of the differences of the results from transgenic lines and the AAV viral approach is needed.

A large portion of the Discussion is now devoted to the limitations of photostimulation compared to paired recordings and to more general limitations of slice physiology. Additionally, potential sources of error associated with photostimulation are now described to motivate our initial characterization of photosensitivity (Figure 1).

Differences in the anatomical locations of presynaptic populations labeled by Ai167 and AAV are more explicitly discussed for both *Rorb* and Sst-Cre lines in the revised manuscript. We directly compared connectivity measured using Ai167 and AAV in Figures 4B, 4C, 6B, 6C, and Figure 3 —figure supplement 3, and provide statistics in the text. Additionally, we compared features of synaptic responses measured using either Ai167 or AAV for both *Rorb* and Sst-Cre (Figure 7 —figure supplement 3). Although none of these comparisons revealed significant differences between connections identified with AAV or Ai167, we highlighted that inhibitory connections identified using Sst:Ai167 did trend towards larger amplitude responses compared to those measured using Sst:AAV. Given that we observed significantly larger IPSPs in response to photostimulation of Pvalb interneurons, we proposed that this trend may be indicative of off-target labeling of Pvalb-interneurons in transgenic animals due to *Sst*-driven expression of Cre in early development.

Regarding the reviewers’ specific point of L4 Sst to L2/3 PC connectivity and differences in the number of cells sampled between Ai167 and AAV, the connectivity rate was calculated as the number of connections found divided by the number of connections probed within each distance bin. To probe a connection, photostimuli were targeted to individual labeled cells. We would expect differences in labeling density to be of much greater concern if stimuli were generated in a standardized grid pattern across all experiments (as is often done in one-photon mapping experiments). While the observed connection rate is lower for Sst:Ai167 compared to Sst:AAV, we highlight the highly overlapping confidence intervals (Figure 6B) and provide the following statistics in the text of the Results: (Sst:Ai167: 10.5%, 2 connections/19 probed; Sst:AAV: 20.6%, 51 connections/248 probed; p=0.38, Fisher’s Exact, all measures within 200 µm horizontal distance).

iii) Discussion on the distribution of synapses in Figure 5. Does this suggest an inverse sombrero-function of activation where excitation from neurons in neighboring columns distal neurons is suppressed by inhibitory neurons in the same column? Does short-term synaptic plasticity shaping response amplitudes?

We’ve added a description of differences in the spatial distributions of excitatory and inhibitory connections to the end of the Results section on connectivity (Figure 6 —figure supplement 2). We reference a recent rabies study that similarly found a broader distribution of excitatory inputs to L2/3 pyramidal cells relative to inhibitory inputs (also in mouse V1). We appreciate the suggestion to highlight and discuss this.

We’ve added a new figure describing short-term plasticity measured from photoresponses with multiple PSPs (collected from over 200 optically identified connections, Figure 8). We observed and highlight some significant differences in paired-pulse ratios measured between presynaptic Cre lines. We agree with Reviewer #3’s statement that “the data may be modest, but would nevertheless be potentially very interesting to the field”, and thank all reviewers for the suggestion.

In further regard to changes in connection strength, we characterized rundown over the course of repetitive photostimulation by performing linear regressions of PSP amplitudes versus photostimulus number. On average, rundown was small (0.2% decrease in PSP amplitude per photostimulus trial). The results (including mean, median, and standard deviation of rundown across cells) are further broken down by Cre line in the Methods section describing measurement of PSP amplitudes.

iv) In Figure 3C, there is strong asymmetry on L6 inputs at horizontal axis. Explain and discuss the implication.

While heatmaps (such as Figure 3C in the initial submission) can provide a useful visualization of connection probability in two-dimensional space, they are limited in the ability to convey confidence intervals (such as those associated with sampling). In our initial submission, the “hot spot” near the L5/L6 border that drove the apparent asymmetry on the anterior/left side of the graph has an overall connection rate of 11%; however, the associated 95% confidence interval ranges from 3-26% (3 connections found/28 probed). After conducting more experiments with *Rorb* and Tlx3-Cre, our sampling of these deep L5 excitatory connections increased and the “hot spot” in this region of the plot is less prominent (Figure 6 —figure supplement 2A). We have added an additional panel that plots connection probability with 95% confidence intervals against anterior and posterior horizontal offset (Figure 5 —figure supplement 2C; also related to Discussion point *iii*) to provide readers with a visual sense for these confidence intervals.

[Editors’ note: what follows is the authors’ response to the second round of review.]

Essential revisions:1) Present cell density counts for each Cre line and for viral infection and add high quality images of representative expression.

We have presented these images and estimates of density as a new figure in the manuscript (Figure 2). Following recommendations of Reviewer #3, we also provide estimates of off-target activation based on these measurements and our photostimulus characterization data .

2) Address the comments from reviewer #3 for clarification

Detailed responses follow.

3) Scholarly cite the previous work using 2p synaptic mapping in the primary cortices.

These additions have been made.

4) Add description of PENK-cre line in the Methods.

This has been added. Further described in response to reviewer #3’s comment #5.

Reviewer #3:Hage and colleagues present a semi-comprehensive survey of functional synaptic connectomics onto L2/3 pyramidal cells (PCs) in mouse V1. They go broad – sampling a good variety of Cre lines that sparsely label cell types in different layers and two major classes of GABAergic neurons. They provide an admirable and rigorous analysis of the effective spatial resolution of their photo-stimulation and perform difficult and important ground truth validation of their optogenetic approaches using targeted paired recording. Overall, I find the experiments performed to a high caliber and the conclusions largely sound. One key issue is the density of the opsin expression in these Cre lines is not quantified – neither for the Cre expression per se in each line nor the infectivity ratio, making it hard to assess the denominator in the connectivity frequency estimates. However, the presumed sparsity of expression (since they typically only sampled a relatively small number of Cre+ cells per recording with manual selection) does help mitigate any issues with 'off target' activation since their resolution is not at the single cell level.The authors analyze their extensive data in a number of logical and expected ways, quantifying connection probability and strength vs cell type and horizontal distance from the patched PC, as well as a nice analysis of convergence, divergence, and reciprocal connections. Nearly all of their findings confirm prior work from pair recordings, helping validate their approach, with the major exception being the stronger input from L5 intratelecenphalic neurons to L2/3 PCs which is certainly interesting and unexpected based on past work.My criticism is this study is nearly overly broad but somewhat shallow – there are many results but limited new insight gained into the underlying connectivity, and few impactful or entirely new findings. The study establishes their approach as a powerful and valid technique for sampling connectivity in this manner, but the approach has been used by others, including one of the co-authors. Thus, I am uncertain of the true impact of this study. Nevertheless, it is well executed and convincing that this approach is powerful and well-controlled.1) Please show images of each Cre line in the Ai167 line and with viral infection. Please quantify penetrance and/or infectivity by staining for NeuN and other appropriate strains. My sense is the expression was very sparse in most cases but this is not clearly explicated or quantified.

We added example images for each Cre line/expression method used in this study (Figure 2A). We measured the density of labeled cells in our experiments and report these values in Figure 2B,C. We are not aware of appropriate and validated antibodies which could be used to estimate penetrance for all Cre lines and/or targeted subclasses examined here. This is particularly challenging for the Cre lines that do not utilize an IRES insertion after the endogenous gene (Ntsr1-Cre_GN220, Scnn1a-Tg3-Cre, Tlx3-Cre_PL56). As an alternative compared the densities of cells measured in our photostimulation experiments to densities measured from same Cre lines crossed to the Ai14 tdTomato reporter (Figure 2B,C; Scnn1a was compared to Ai9 to have data from a matched age). For Pvalb and Sst interneurons, these density estimates of ~4600-4800 cells/mm^3^ in Ai14 are similar to previous density measurements for these subclasses (~4900-5000 cells/mm^3^; Kim et al., 2017; Keller et al., 2018). We could not find reported densities of cells labeled using the other Cre lines to compare our measurements to. The analysis suggests that use of AAV labeled ~85% of the targeted cells, but penetrance was lower with Ai167 (~25%). We cite previous estimates of the density of all neurons in mouse visual cortex and plot the reported average of these estimates alongside our labeled cell densities (Figure 2B,C). We note that based on these estimates, photosensitivity by Cre-mediated opsin expression is imparted to a minority of all neurons.

2) The authors motivate the use of a higher power (85 mW in a small spot!) because this will spike most cells based on a very nice stimulation power analysis. This reduces false negatives but should increase false positives by increasing off target activation. This might be mitigated by very sparse expression (see point 1). Not worth redoing, but I would suggest using less power at the expense of spiking every cell in the future, particularly in denser expressing lines.

We appreciate this suggestion and are exploring use of either lower power stimuli or both low and high power stimuli to all tested connections in future projects.

3) Based on point 1 and 2 the authors could provide estimates of how many cells on average each photo-stimulus activates in each line. This would be informative.

We now provide these estimates for all experimental conditions using characterization data and the relative positions of labeled cells (Figure 2D-H). Additional information on the generation of these estimates is provided in the Material and Methods section and within Figure 2—figure supplement 1.

4) The FWHM numbers in Figure 1 and Figure 1 supplement just don't seem to match up to my eye. Perhaps the FWHM or the average has been computed improperly, or Figure 1 is reporting HWHM? From Figure 1 – Supplement 1 it's clear the 85 mW is driving saturation of spiking (hence the PSFs look 'squared off' and using less power would be preferable). Is is odd the PSFs are saturated in the z axis by not in the x axis, I can't seem to wrap my mind around that. In many of the individual examples (grey lines) the z axis FWHM looks much larger than the average, sometimes almost 100 microns. This is a cause for concern. Figure 1 S1 implies each photo-stimulation may be activiting quite a few cells along the z-axis depending on expression density.

We greatly appreciate the close attention paid to Figure 1 and the corresponding supplement. In our initial submission, we reported values as HWHM throughout the manuscript. When submitting our previously revised manuscript with a requested change to FWHM, an old version of Figure 1 was included. This has been corrected.

Regarding saturation, we have added measures of latency and jitter with lateral and axial target displacement (Figure 1—figure supplement 2 and 3). We observed progressive increases in these values with increasing spatial offset – including at distances where the spike probability is 1. An increase in latency with axial offset can also be observed in the example traces in Figure 1A.

We indeed observe variability in spatial resolution across cells and find a minority of cells that exhibit spiking with 50 µm of axial offset. Specifically, 17% of cells when using *Rorb*:AAV and 0-10% of cells in other experimental conditions. The falloff in activation probability is relatively sharp with lateral offset, so the axial alignment between cells needs to be precise to have the highest likelihood of off-target activation. We hope Figure 2D-H and Figure 2—figure supplement 1 provide qualitative and quantitative context for the relationship between photostimulus resolution and relative cell locations measured in our experiments and predicted for homogenously distributed cells across a range of densities.

5) Please provide more information on the Penk-Cre line, it's not actually listed in the Methods section and not a super-well established line. From the Allen website it looks very sparse and biased to L2(?). Is it all pyramidal cells, what shows this?

We have added more information on the Penk-Cre line to the Methods including the full name and analysis of co-labeling by the interneuron marker gene Gad1. We report the densities of labeled cells in our photostimulation experiments and with Penk-Cre crossed to Ai14. As described in response to Recommendation 1, we report that with the Cre lines used in this study, a minority of all neurons are labeled.

We added that transcriptomic analysis of cells labeled by the Penk-Cre line indicates that a subset of labeled cells are VIP interneurons (Tasic et al. 2018). To estimate the fraction of Penk-Cre labeled cells that are GABAergic interneurons, we examined double fluorescence *in situ* hybridization images staining for tdTomato and Gad1. We observed that ~12% of tdTomato labeled cells in the Penk-Cre line are co-labeled with Gad1 (Figure 3—figure supplement 2). We performed the same analyses on analogous images from interneuron Cre lines (Pvalb-Cre and VIP-Cre) and observe a high degree of tdTomato and Gad1 co-labeling (85-86%). In contrast, we measured a low amount co-labeling with *Rorb*-Cre (3%) – increasing our confidence in the analysis. We describe that off-target labeling of interneurons will result in an underestimate of excitatory connection probability. However, we do not expect the identity of photostimulated cells to depend on the identity of cells patched or their relative locations. Therefore, reducing the number of connections probed by a common factor has little effect on the relative differences in excitatory connectivity observed between postsynaptic cell subclasses and across intersomatic distances.

6) Though the study probes ~10,000 connections, this is actually not that much. Lefort et al. probed thousands by patching 6 cells at a time and doing many recordings. In table 1 the data says only 205 Exc to Exc connections were probed in L2/3. This is actually very few, not even double their paired recordings. Some explanation for the lower number is warranted – is this related to the very sparse nature of opsin expression in each line?

Table 1 compares data from 2P photostimulation experiments to data from paired recordings at matched intersomatic distances. The value of 205 excitatory to excitatory connections probed refers only to cells within 100 µm of each other. On the following row of Table 1, we report analogous data measured at 100-200 µm (312 connections tested by 2P opto and 48 by paired recordings). Additional photostimulation data collected with the Penk Cre line at larger intersomatic distances are shown as a map and a histogram in Figure 3B,C, but are not reported within Table 1, as we do not have sufficient paired data at these distances to make meaningful comparisons. Across all distances, we tested 895 L2/3 excitatory to L2/3 excitatory connections from 25 patched cells using the Penk-Cre line (reported in the legend of Figure 3).

For further explanation of the number of tested connections within 100 µm of intersomatic distance, we note the different sampling strategies of paired recordings and photostimulation. Though it is not a strict limitation, in practice, paired recordings for measuring synaptic connectivity are often made between nearby cells. In photostimulation experiments, we targeted all identified opsin-expressing cells within a 300 µm x 300 µm field of view before adjusting either the focus or stage position to examine additional cells in a new field of view. Even in cases where a patched/postsynaptic cell is centered within the frame, most of the area (~65%) corresponds to intersomatic distances >100 µm away from the postsynaptic cell – leading to greater sampling of more distant cells. We hesitate to call this a bias as it represents the inherent geometry of the system – neurons have a greater number of distal neighbors than proximal neighbors. Notably, within our study, measuring connectivity at extended distances revealed translaminar excitatory connections with large horizontal offsets and a gradual decline in excitatory connectivity with distance relative to inhibitory connectivity (Figures 5, 6; Figure 6—figure supplement 2). Therefore, it was informative to test connections at extended distances.

Regarding the broader merit of the ~10,000 connections probed, the impressive study of Lefort et al. (2009) reports on 8895 possible connections tested by recording from 2550 neurons. Another impressive and widely cited multipatch study (Jiang et al., 2015) describes 11,771 connections probed via 2201 patch-clamp recordings. In our study, we report on 10,720 tested connections from 384 postsynaptic recordings. As the process of patching is widely viewed as a rate limiting step, this study represents a significant improvement in throughput in this regard. The reviewer is correct that relative density of labeling does influence the throughput of a given experiment. For example, from the relatively dense Ntsr1-Cre line, we probed 1216 connections from 23 patched cells. In comparison, mapping with the sparse Scnn1a-Cre line yielded 158 tested connections from 22 patched cells. We report the total number of connections tested and the corresponding number of postsynaptic cells patched in the legends of each figure containing mapping data (Figures 3-6).

The other major factor influencing throughput is the length of the recordings. We likely could have acquired more data from each patched cell had we conducted experiments in conditions that favor longer recordings (e.g. room temperature and brain slices generated from juvenile mice). We also tested connections relatively thoroughly (each cell stimulated at least 20 times) and cautiously (at least 1 second between photostimuli). Depending on the goals of future studies, it may be beneficial to adjust some of these factors in favor examining more connections.

For most readers, we believe the focus on translaminar connectivity within the >10,000 connections tested will be of greater interest than experimental throughput. While large-scale multipatch studies often include recordings from neurons in multiple layers, the sampling is commonly biased to testing intralaminar connections. For example, for inputs to L2/3 pyramidal neurons, Jiang et al. tested a total 651 intralaminar and 442 translaminar connections (from L5, L4 & L6 not examined; data compiled from their Figure S14). Similarly, Lefort et al. report 1828 excitatory connections tested within layers 2 and 3, compared to 1059 possible translaminar excitatory connections to L2/3 from all other layers (L4-L6; compiled from their Table 2). 2P optogenetic stimulation allowed us to deeply characterize synaptic inputs to a single subclass of postsynaptic cell (L2/3 pyramidal cells), with a purposeful focus on translaminar connections. A total of 7824 translaminar connections to these cells were tested. From these, >300 translaminar excitatory connections and >100 translaminar inhibitory connections were measured. In comparison, the studies above describe a total of 65 translaminar excitatory (all from Lefort et al., 2009) and 31 translaminar inhibitory (all from Jiang et al., 2015) connections to L2/3 pyramidal cells. Our intent here to highlight the relative depth of characterization enabled by our approach. Importantly, the multipatch studies mentioned include additional, broader sampling of translaminar connections to other layers, enabled in part by the bidirectional testing of connectivity possible with paired recordings.

We hope to deepen and extend this dataset in the future, but we also believe the depth to which we have characterized translaminar synaptic connections to L2/3 (and compared to intralaminar connections under the same conditions) is unique and will be of interest to many readers.

7) The paired recording confirmation of optically identified pairs is admirable, but if opsin expression was extremely sparse to begin with, this really isn't that compelling. In Figure 3 there appears to be very few opsin+ neurons, so their high confirmation rate is not surprising. This being said, it doesn't negate the proof their approach works, but it doesn't imply the approach would work for dense expression. If this is a real limitation of this approach is much be stated much more forthrightly in the text so other groups don't adopt this approach without carefully checking their spatial resolution. There is nothing wrong with using sparse expression for what they do in this study, but it needs to be more clear to the reader.

We added a plot of the measured positive predictive value versus the corresponding cell densities (Figure 4—figure supplement 1). We also estimate off-target activation across a range of photosensitive cell densities (Figure 2H) and include a previously reported estimate made using two-photon glutamate uncaging in rat cortex (with a high density of endogenously photosensitive cells) that is in close agreement with our own estimates. We highlight that if photosensitivity was imparted to most neurons, with the spatial resolutions measured in our study, we would expect more common and problematic off-target activation. We hope these additions provide appropriate context and caution to readers.

8) Are L5 Rorb-Cre cells more similar to L4 neurons or other L5 neurons? This is not really for this study, but would be nice to state since the AI must have the transcriptomic data.

Based on transcriptomic data, at least some L5 *Rorb* Cre neurons more closely resemble L5 IT neurons than L4 excitatory neurons. Additionally, the long-range axonal projections measured in *Rorb* Cre mice are consistent with labeling of L5 IT neurons. We cite and state these previous observations more deliberately in this version of the manuscript. Given the caveats of sampling by FACS and layer dissections, it is difficult to broadly state if all or most L5-*Rorb* Cre cells are more similar to L4 neurons or other L5 neuron types. Spatial transcriptomics may soon provide additional information, but we agree that this question is outside of the scope of this study.

9) What was the motivation for sometimes using the Ai167 line (not soma-targeted) and other times an AAV? Some clarification is warranted.

Both strategies were developed and tested largely in parallel and found to be suitable for targeted 2P photostimulation. In the added section on density measurements, we describe how the more complete labeling with AAV was motivation to use it as our primary strategy, but also acknowledge that the sparse labeling with Ai167 may limit off-target activation. When motivating our comparison of connectivity and connection properties between Sst:Ai167 and Sst:AAV experiments, we discuss how previous studies of synaptic connectivity have primarily used transgenic labeling and that viral transfection carries caveats of possible tropism and toxicity.

Each presynaptic subclass examined in this study was tested using AAV (or both AAV and Ai167), except for Ntsr1-Cre/L6 corticothalamic neurons. As photostimulation of these cells in the Ai167 line produced little in the way of synaptic responses, we did not pursue additional mapping with somatic restriction using AAV. The lack of somatic restriction in Ai167, combined with the weak response to one-photon stimulation (Figure 5—figure supplement 1) mitigated concern that the near absence of 2P-evoked connections was a result of axonal severing.

10) For Figure 6D, can the Exc to L2/3 be split according to presynaptic layer for clarity?

We’ve added plots of excitatory and inhibitory connection amplitudes versus horizontal distance split by layer (Figure 7F).

11) For the paired pulse analysis the authors identify high frequency bursts of the presynaptic cell in the postsynaptic potentials, with 5-10 ms ISIs. This is very fast (100-200Hz) and not typical for most paired pulsed analysis, although still interesting. It would have better to deliberately generate paired pulse or pulse train responses of connected neurons at lower, more physiological frequencies for most cell types (say 10-40 Hz). Also, I'm not convinced the deconvolution analysis for estimated PPF is valid, this could be assessed, at least at lower frequencies, by comparing against voltage clamp measurement of the same inputs in the same cells (this would not be compromised by space clamp since PPF measurements are of the same equally poorly clamped synapse).

In our previous submission, we acknowledged that the measurements of STP presented here are more limited than one would obtain by paired recordings. We also described how we did not attempt to apply trains of photstimuli due to the variable photostimulus responses observed across ChrimsonR-expressing cells. Finally, we point to the potential for newer, faster opsins to make such assays more consistent and interpretable in the future. We have added references to previous studies that used exponential deconvolution to account for temporal summation of PSPs. While we are sympathetic to the reviewer’s desire for such data in this study, we don’t believe this generation of tools is sufficient. We note here that the figure on STP was added following the suggestion received in our initial reviews. Extraction of STP data from doublets of PSPs has been a frequently asked question when sharing the data in other contexts as well. We hope that we have provided an adequate description of the data and our analyses to satisfy interested readers, but also make the limitations clear.

References

Jiang X, Shen S, Cadwell CR, Berens P, Sinz F, Ecker AS, Patel S, Tolias AS. 2015. Principles of connectivity among morphologically defined cell types in adult neocortex. *Science* 350:aac9462.

Keller D, Erö C, Markram H. 2018. Cell Densities in the Mouse Brain: A Systematic Review. *Front Neuroanat* 12:83.

Kim Y, Yang GR, Pradhan K, Venkataraju KU, Bota M, García DMLC, Fitzgerald G, Ram K, He M, Levine JM, Mitra P, Huang ZJ, Wang XJ, Osten P. 2017. Brain-wide Maps Reveal Stereotyped Cell-Type-Based Cortical Architecture and Subcortical Sexual Dimorphism. *Cell* 171:456–469.e22.

Lefort S, Tomm C, Floyd SJC, Petersen CC. 2009. The excitatory neuronal network of the C2 barrel column in mouse primary somatosensory cortex. *Neuron* 61:301–16.

Tasic B, Yao Z, Graybuck LT, Smith KA, Nguyen TN, Bertagnolli D, Goldy J, Garren E, Economo MN, Viswanathan S, Penn O, Bakken T, Menon V, Miller J, Fong O, Hirokawa KE, Lathia K, Rimorin C, Tieu M, Larsen R, Casper T, Barkan E, Kroll M, Parry S, Shapovalova NV, Hirschstein D, Pendergraft J, Sullivan HA, Kim TK, Szafer A, Dee N, Groblewski P, Wickersham I, Cetin A, Harris JA, Levi BP, Sunkin SM, Madisen L, Daigle TL, Looger L, Bernard A, Phillips J, Lein E, Hawrylycz M, Svoboda K, Jones AR, Koch C, Zeng H. 2018. Shared and distinct transcriptomic cell types across neocortical areas. *Nature* 563:72–78.